# Hybridizing carbonate and ether at molecular scales for high-energy and high-safety lithium metal batteries

Jiawei Chen[1], Daoming Zhang[2], Lei Zhu[1], Mingzhu Liu[3], Tianle Zheng [4], Jie Xu [1], Jun Li[2], Fei Wang [1], Yonggang Wang [1], Xiaoli Dong [1] ✉ & Yongyao Xia [1]✉

Commonly-used ether and carbonate electrolytes show distinct advantages in active lithium-metal anode and high-voltage cathode, respectively. While these complementary characteristics hold promise for energy-dense lithium metal batteries, such synergy cannot be realized solely through physical blending. Herein, a linear functionalized solvent, bis(2-methoxyethyl) carbonate (BMC), is conceived by intramolecularly hybridizing ethers and carbonates. The integration of the electron-donating ether group with the electron-withdrawing carbonate group can rationalizes the charge distribution, imparting BMC with notable oxidative/reductive stability and relatively weak solvation ability. Furthermore, BMC also offers advantages including the ability to slightly dissolve $LiNO_3$, excellent thermostability and nonflamm-ability. Consequently, the optimized BMC-based electrolyte, even with typical concentrations in the single solvent, demonstrates high-voltage tolerance (4.4 V) and impressive Li plating/stripping Coulombic efficiency (99.4%). Moreover, it fulfills practical lithium metal batteries with satisfactory cycling performance and exceptional tolerance towards thermal/mechanical abuse, showcasing its suitability for safe high-energy lithium metal batteries.

The ambitious goal of achieving carbon neutrality has been driving the advancement of energy-dense battery chemistry, particularly in the realm of high-voltage lithium metal batteries (LMBs)[1–4]. However, their practical implementation poses demanding requirements for the electrolytes, which must simultaneously satisfy the aggressive high-voltage cathode, the hyperactive Li anode and safety performance[5–7] As the most successful electrolyte system in commercial lithium-ion batteries (LIBs), carbonate-based electrolytes have demonstrated excellent oxidative stability of ~4.3 V vs. $Li^+/Li$, accommodating the well operation of high-voltage cathode[8]. Nevertheless, the conjugated electron-withdrawing effect of the carbonyl oxygen results in a pronounced positive charge on the carbonyl carbon. The electron-deficient feature of carbonyl carbon makes carbonates tend to gain

electrons and thus exhibit poor reductive stability[9]. Consequently, continuous severe parasitic reactions will occur when coupling carbonate-based electrolytes with hyperactive Li anode, generating organic-rich solid-electrolyte-interphase (SEI) that is non-uniform, unstable and mechanically fragile[5]. Additionally, the formation of nasty Li dendrites is easily triggered by inhomogeneous distributions of $Li^+$ flux and further fuels the parasitic reactions[10,11]. Ultimately, the resultant accumulation of resistive 'dead Li' and decreased Coulombic efficiency (CE) upon repeated Li plating/stripping processes would lead to the poor cycle life of LMBs using carbonate-based electrolytes[5,12].

By contrast, electrolytes based on ether solvents have long stood out for their exceptional reductive stability in supporting Li anode

[1]Department of Chemistry and Shanghai Key Laboratory of Molecular Catalysis and Innovative Materials, Institute of New Energy, iChEM (Collaborative Innovation Center of Chemistry for Energy Materials), Fudan University, Shanghai 200433, China. [2]Sinopec Shanghai Research Institute of Petrochemical Technology Co., Ltd., Shanghai 201208, China. [3]School of Chemistry, South China Normal University, Guangzhou 510006, China. [4]Department of Chemistry, College of Sciences, Shanghai University, Shanghai 200444, China. ✉e-mail: xldong@fudan.edu.cn; yyxia@fudan.edu.cn

among various types of solvents[5,6,9]. Nevertheless, the lone pair electrons on ether oxygen have a strong donating tendency, which renders routine ether-based electrolytes exhibit limited oxidative stability beyond ~4.0 V vs. Li+/Li[13], thereby failing to support state-of-the-art high-voltage LMBs. Although common strategies such as high/localized high concentration electrolytes (HCEs/LHCEs) can be implemented to alter solvation structures and establish a superior cathode-electrolyte-interphase (CEI) to compensate the shortcomings of ethers[14–16], the practical application of these approaches is still hindered by their associated costs and environmental burdens[17]. Recently, several researches also demonstrated the utilization of ether-based electrolytes in high-voltage LMBs through the incorporation of fluorinated functional groups into ethers[18,19]. In fact, the upgraded oxidative stability can be regarded as the benefits from the reduced electron cloud density on ether oxygen owing to the electron-withdrawing fluorinated groups. Regrettably, above strategies rarely take into account that the modified solvent molecules are still volatile and flammable with low flash points, which can easily intertwine with short circuits induced by Li dendrites, thus exposing LMBs to thermal runaway and even catastrophic safety accidents[20,21]. Therefore, how to design a solvent simultaneously possessing high redox stability and high safety through molecular engineering presents a significant challenge.

Herein, we revisit ethers and carbonates from the molecule level, both of which are sort of like Yin and Yang with opposite properties in Chinese culture who can complement each other to compose integral Tai Chi. By hybridizing the electron-withdrawing carbonate bond and the electron-donating ether bond within a single molecule, it is possible to precisely adjust their respective charge distributions, an achievement that cannot be attained through mere physical blending of ethers and carbonates. In this way, an intramolecularly hybridized linear solvent, bis(2-methoxyethyl) carbonate (BMC), was conceived and synthesized successfully, exhibiting considerable redox stability and relatively weak solvation power. Beyond that, the deliberate elongation of the molecular chain for increasing the intermolecular van der Waals forces brings about an elevated thermal stability and a heightened flash point for BMC as expect. Despite the recent report on the use of BMC in safe graphite-based LIBs[22], we initiated a fundamental understanding on the molecular hybridization and a comprehensive investigation of BMC for hyperactive Li metal anode and high-

voltage LMBs. Paired with the general concentration of lithium bis(-fluorosulfonyl)imide (LiFSI), the optimized single-solvent BMC-based electrolyte fulfills a high average dendrite-free Li plating/stripping CE of 99.4%, surpassing those obtained in routine ether-based and carbonate-based electrolytes. Moreover, it adapts to the well operation of LiNi$_{0.8}$Co$_{0.1}$Mn$_{0.1}$O$_2$ (NCM811) under high voltage of 4.4 V. These merits thus enable 92% capacity retention in Li||high-loading-NCM811 (4.8 mAh cm$^{-2}$) full cell after 150 cycles. Respectable cycling performance with BMC-based electrolyte can also be attained even with the more demanding anode-free Cu||NCM811 coin and pouch cells. More importantly, Li metal pouch cells come through the nail penetration test and exhibit improved thermal safety temperature up to 155 °C, strongly manifesting the critical property of BMC as a safe solvent. The molecular design concept put forward in this work can offer an additional train of thought towards the development of promising electrolytes for future commercial LMBs.

## Results
### Molecular design and physicochemical property
In order to bolster the performance and ensure the safety of high-voltage LMBs, it is a pressing necessity to employ an electrolyte showcasing commendable redox stability and intrinsic safety. Nevertheless, the prevailing ether-based and carbonate-based electrolytes, presently predominant in lithium battery chemistry, encounter obstacles in harmoniously integrating these coveted characteristics. Specifically, ether solvents, exemplified by the commonly used linear 1,2-dimethoxyethane (DME, Fig. 1a), typically exhibit favorable reductive stability. Coupled with its merit to dissolve LiNO$_3$—a critical additive for stabilizing the Li interface but scarcely soluble in carbonates—DME demonstrates its applicability for Li anode[10,23,24]. Nevertheless, the lone pair electrons of charge-concentrated ether-oxygen (see electrostatic potential (ESP) map) impart DME with electron-donating characteristics and make it prone to oxidation. Conversely, carbonate solvents like linear dimethyl carbonate (DMC, Fig. 1b) exhibit opposite behavior, highlighting their suitability for high-voltage cathodes with commendable oxidative stability. However, conjugate electron-withdrawing effect of the carbonyl oxygen results in the low electron density at carbonyl carbon, rendering it electropositive (as observed in the ESP map) and susceptible to gain electron (poor reductive stability). Given the complementary merits of ethers and carbonates, it is

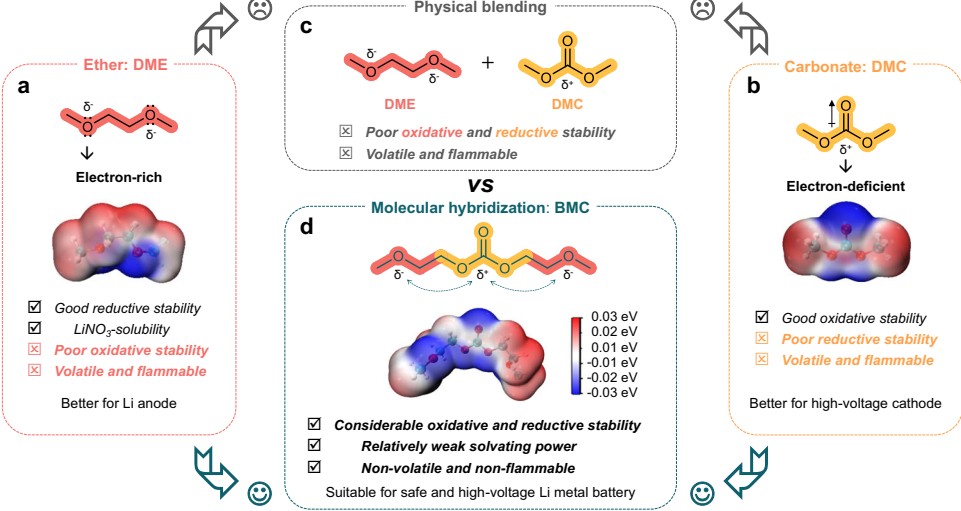

**Fig. 1 | Schematic illustration of BMC molecule design.** Molecular structures, electrostatic potential (ESP) maps and characteristics of (**a**) DME (representative linear ether) and (**b**) DMC (representative linear carbonate). **c** The intuitively physical blending of DME and DMC. **d** Well-designed BMC solvent by molecularly hybridizing ether oxygen and ester carbonyl groups into a linear molecule. The meticulously engineered molecular hybridization results in a solvent capable of satisfying the demands for high-voltage and high-safety LMBs, a feat unattainable through mere physical blending of ethers and carbonates.

easy to intuitively conceive an electrolyte by physically blending them together to support both reactive Li anode and high-voltage cathodes (Fig. 1c). Unfortunately, simple physical blending often leads to counterproductive outcomes with rather poor redox stability because simply mixing them together cannot change their molecular structure and thus would not influence the distribution of electron within each individual molecule. This disappointing recognition indicates physical blending exposes the shortest plank instead of the longest one, which cannot make full use of the advantages of ether and carbonate solvents. Moreover, whether in the form of ether, carbonate, or their mixtures, they usually exhibit volatility and flammability, thereby presenting potential safety risks[21,25]. To fundamentally overcome the limitations of ethers and carbonates, it is crucial to modify their electron distribution and molecular structure. Therefore, we wonder whether it is viable to regulate the electron distribution by grafting the ether group with electron-donating characteristic onto the electron-withdrawing carbonate group. Additionally, grafting can potentially enhance van der Waals forces by extending molecular chains, thereby raising the boiling and flash points of the solvent. Inspired by this, a linear BMC solvent is precisely designed via molecular hybridization of ether and carbonate, with $CH_3OCH_2$- moieties strategically incorporated at both termini of DMC (Fig. 1d). The ESP map of BMC, in comparison with those of DME and DMC, indicates that electrons of oxygen in ether groups on both sides tend to donate to the intermediate carbonate group, increasing the electron density around carbonyl carbon while decreasing that around ether oxygen. This can be further proved with the charge states of the atoms in the solvent molecules obtained from calculations (Supplementary Fig. 1). Such optimization of electron distribution would equip BMC with better oxidative and reductive stability than ether and carbonate, respectively. Moreover, besides the enhanced thermal stability and fire resistance achieved through intentionally lengthening molecular chain, BMC unexpectedly demonstrates other merits like relatively weak solvating power and the ability to slightly dissolve $LiNO_3$. All of these characteristics derived from the molecular hybridization design, which will be systematically validated later, can effectively meet the requirements of safe high-voltage LMBs. Furthermore, the simple synthesis process of BMC does not require expensive precursors (Supplementary Table 1), which makes it suitable for large-scale production and has good commercialization potential.

To ascertain the properties of synthesized BMC, more comprehensive theoretical calculations were conducted. As shown in Fig. 2a, BMC exhibits a more positive electron affinity energy when compared to DMC, signifying its increased difficulty in gaining electrons and thus enhancing its reductive stability. Additionally, BMC shows a higher ionization energy than DME (Fig. 2b), suggesting greater challenges in donating electrons and therefore better oxidative stability. This further proves the successful tame of oxidative/reductive stability through intramolecular hybridization, effectively compensating the opposite weakness of ethers and carbonates themselves. Moreover, the change of electron distribution influences the coordination with Li$^+$, as reflected with the optimized binding energy depicted in Fig. 2c. DME is generally known to form a five-membered ring chelation structure with Li$^+$ through two oxygen atoms and shows a more negative binding energy than Li$^+$-DMC[26], which means a stronger coordination ability of DME to Li$^+$. Benefitting from the synergetic electron distribution, BMC exhibits a minima electronegativity near the carbonyl oxygen but relatively positive compared to DMC (see ESP maps in Fig. 1), thus generating a weaker coordination affinity with Li$^+$ (Li$^+$-BMC complex-1). Interestingly, there exists alternative coordination configurations for BMC with Li$^+$ (Supplementary Fig. 2). Some of them demonstrate more negative binding energy, such as Li$^+$-BMC complex-2 (similar five-membered ring chelation structure to Li$^+$-DME) and Li$^+$-BMC complex-4 (chelation structure coordinating to Li$^+$ with both of carbonyl and ether oxygens). These two chelation configurations with stronger interaction to Li$^+$ potentially facilitate the dissolution of $LiNO_3$ in BMC (Supplementary Fig. 3). However, it seems that the dominant configuration does not lie in these chelation solvation structures but rather in the Li$^+$-BMC complex-1, which solely relies on carbonyl oxygen coordinating to Li$^+$ with relatively weak solvation energy. This aspect is discussed in Supplementary Fig. 2 and will be further validated through molecular dynamics (MD) simulations and Raman spectroscopy. Furthermore, the large molecular size of BMC (Supplementary Fig. 4) introduces significant steric hindrance, presenting challenges in effectively accommodating multiple BMC molecules within the Li$^+$ primary solvation sheaths (PSSs)[27]. Combined with the relatively weak solvation energy of dominant Li$^+$-BMC complex-1, more anions will have the opportunity to enter the Li$^+$ PSSs and construct desired anion-derived SEI.

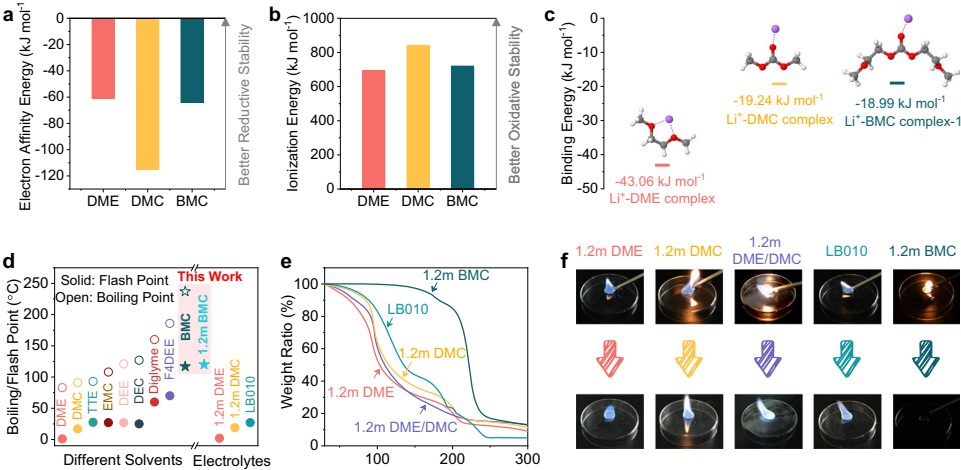

**Fig. 2 | Physicochemical properties of solvents and electrolytes. a** Calculated electron affinity and (**b**) ionization energy of DME, DMC and BMC. **c** Optimized structures and binding energy of Li$^+$-DME, Li$^+$-DMC and Li$^+$-BMC complexes that are depicted by the ball-and-stick model. Purple balls stand for Li ions, while gray, white and red balls stand for C, H and O atoms respectively. (**d**) Comparisons of boiling/ flash points with other liner ether and carbonate solvents/electrolytes (TTE: 1,1,2,2-tetrafluoroethyl-2,2,3,3-tetrafluoropropyl ether, EMC: ethyl methyl carbonate, DEE: 1,2-diethoxyethane, DEC: diethyl carbonate, F4DEE: 1,2-bis(2,2-difluoroethoxy) ethane). **e** TG curves and (**f**) flammability analyses for various electrolytes (Complete flammability tests can be found in Supplementary Movies 1−5).

Safety performance is also an important index to screen the solvent and electrolyte for reliable batteries. As revealed by differential scanning calorimetry (DSC) analysis (Supplementary Fig. 5), the linear BMC exhibits a broad liquid temperature up to 238 °C. Furthermore, its flash point was observed to reach as high as 117 °C, which is close to the tested value (121 °C) on the recently published report focusing on the safety performance of BMC[22]. The superiority over numerous other chain ethers and carbonate solvents in terms of both boiling and flash points significantly highlights the safety characteristics of BMC (Fig. 2d)[6,21,28]. On this basis, an electrolyte with 1.2 mol kg⁻¹ LiFSI salt in the single-solvent BMC (denoted as 1.2 m BMC hereafter) is subsequently formulated, and its thermal stability is evaluated with thermogravimetric (TG) analyses. As shown in Fig. 2e, the weight loss on 1.2 m BMC is negligible (5%) upon heating to 165 °C, demonstrating a remarkable thermal stability. By contrast, physically blended 1.2 m DME/DMC (1/1 by weight), individual 1.2 m DME and 1.2 m DMC, or commercial carbonate-based electrolyte LB010 (1 M LiPF₆ in EC/EMC/DMC 1/1/1 by weight) all exhibit significant mass loss upon heating up to 100 °C, which might cause the rapid rise in internal pressure of the battery and pose a safety risk[25]. Furthermore, these electrolytes generally display remarkably low flash points (≤ 27 °C, Fig. 2d), which are

highly susceptible to ignition upon exposure to an open flame (Fig. 2f, Supplementary Movies 1–4). As a comparison, 1.2 m BMC exhibits a high flash point of 120 °C, ensuring its complete nonflammability (Supplementary Movie 5). These safety features can be attributed to high flash point and enhanced van der Waals force of BMC owing to the intentional elongation of the alkoxy chain (Supplementary Fig. 6). However, the enhanced intermolecular forces also contribute to a higher viscosity of the solvent (Supplementary Fig. 7), consequently leading to a reduction in ionic conductivity of the BMC-based electrolyte (Supplementary Fig. 8). Fortunately, Li⁺ transport can be compensated by a higher Li⁺ transfer number ($t_{Li+}$, 0.47) in 1.2 m BMC compared to other electrolytes (Supplementary Fig. 9).

### Electrolyte structure and compatibility with electrodes

The molecular structure of solvent guides the coordination behavior with Li⁺, leading to distinct solvation structures that subsequently impact interfacial chemistry and further compatibility with electrodes[5,13]. Hence, MD simulations were conducted to demonstrate the changes on solvation structures after the molecular hybridization. It can be detected from Fig. 3a that the Li⁺ primary solvation sheaths (PSSs) of 1.2 m DME are predominantly occupied by DME with few FSI⁻

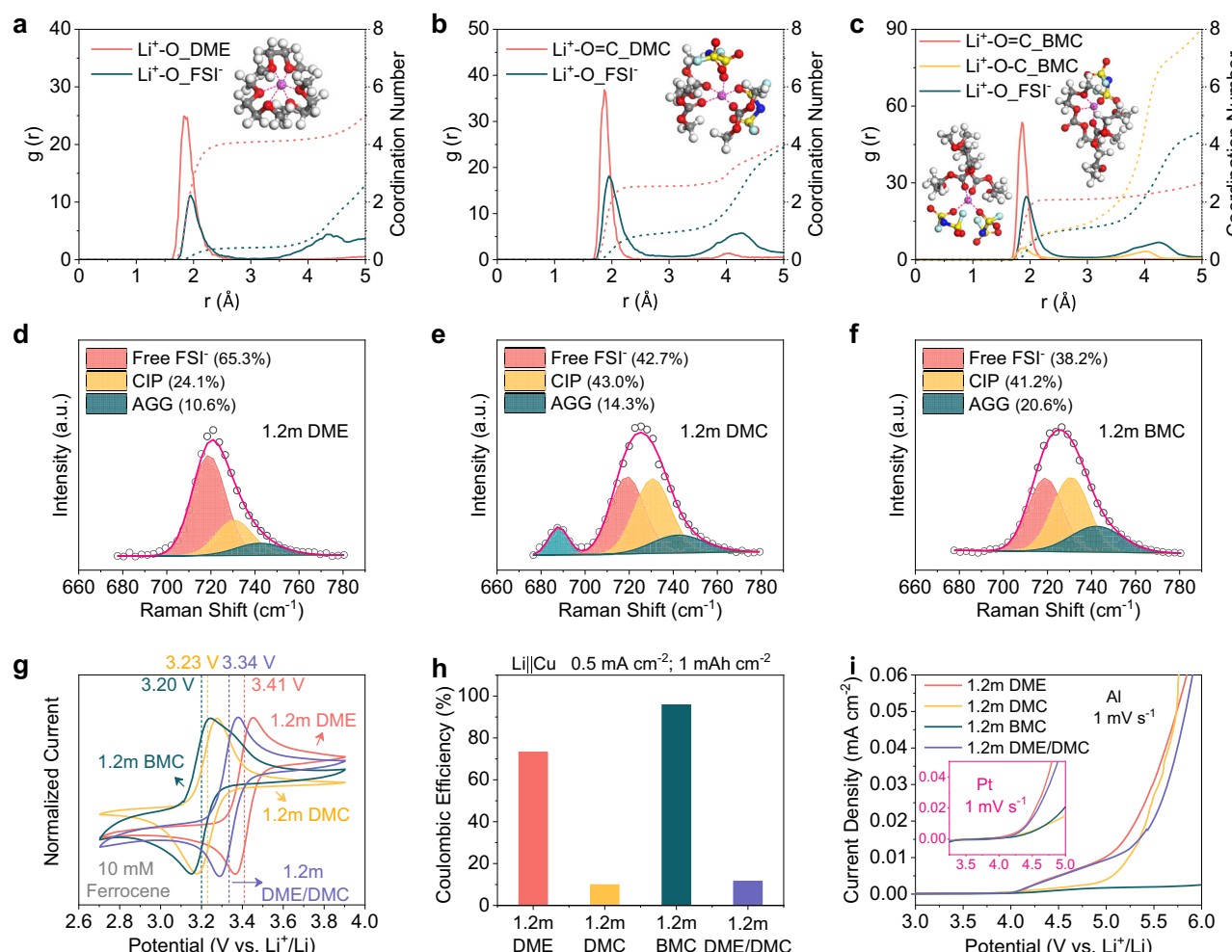

**Fig. 3 | Solvation structures of electrolytes and their electrochemical behaviors with electrodes.** The radial distribution function (g(r), solid lines) and coordination number (dash lines) for (**a**) 1.2 m DME, (**b**) 1.2 m DMC and (**c**) 1.2 m BMC (The insets are representative Li⁺ solvation structures extracted from MD simulations. Purple balls stand for Li ions, while gray, white, red, blue, yellow and cyan balls stand for C, H, O, N, S and F atoms respective.). Raman spectra of (**d**) 1.2 m DME, (**e**) 1.2 m DMC and (**f**) 1.2 m BMC in 680-780 cm⁻¹ (S-N-S bending vibration of FSI) (CIP:

contact ion pair; AGG: ion aggregate). **g** Cyclic voltammetry (CV) curves of 10 mM ferrocene (Fc) dissolved in various electrolytes using Pt as working electrodes at 5 mV s⁻¹. **h** Li plating/stripping CE obtained from Li||Cu cells after 10 cycles tests at 0.5 mA cm⁻², 1 mAh cm⁻² in various electrolytes. **i** Oxidative stability of various electrolytes measured with Al working electrodes at 1 mV s⁻¹ (The inset is results obtained with Pt working electrodes).

due to the strong chelation between $Li^+$ and DME (Supplementary Fig. 10), which is consistent with above-mentioned high binding energy of $Li^+$-DME complex. As a comparison, the relatively weaker coordination propensity of DMC towards $Li^+$ leads to an approximately threefold increase in the ingress of $FSI^-$ into 1.2 m DMC $Li^+$ PSSs compared to that in 1.2 m DME (Fig. 3b and Supplementary Fig. 11). Similar situation can also be detected in 1.2 m BMC, whose $Li^+$ PSSs are found to have the highest number of $FSI^-$ as expected (Fig. 3c and Supplementary Figs. 12 and 13). The simulation results also validate the presence of a diverse range of $Li^+$-BMC coordination configurations, which have also been shown in DFT calculations. It is evident that the coordination number of $Li^+$-C-O_BMC is only half that of $Li^+$-O = C_BMC, implying that carbonyl oxygens serve as the main coordinating sites. Further statistical analysis of $Li^+$-BMC coordination configurations in MD simulations confirms that the dominant coordination configuration is represented by $Li^+$-BMC complex-1 (Supplementary Table 2), where the carbonyl oxygen coordinates solely with $Li^+$ and exhibits relatively weak binding energy.

Meanwhile, spectroscopic characterizations were also performed to further verify the above conclusions about solvation structures. Raman spectra explicitly provide the coordination of $Li^+$ with solvent molecules, which reconfirms carbonyl oxygen as the main solvating site of BMC (Supplementary Fig. 14). The coordination environment of $FSI^-$ in different electrolytes has been explored from the S-N-S bending vibration in 680-780 $cm^{-1}$ (Fig. 3d–f), which can be classified into three distinct bands: free $FSI^-$ (anion without pairing, 719.0 $cm^{-1}$), contact ion pair (CIP, anion paired with one $Li^+$, 730.6 $cm^{-1}$), and ion aggregate (AGG, anion paired with two or more $Li^+$, 742.3 $cm^{-1}$)[29]. Their respective proportions are determined by the area of the Gaussian-Lorentzian peaks obtained by deconvolution. It can be observed that the proportion of free $FSI^-$ in 1.2 m DME (65.3%) decreases to 42.7% in 1.2 m DMC and 38.2% in 1.2 m BMC (Supplementary Table 3), respectively. Synchronously, AGG increases from only 10.6% in 1.2 m DME to 14.3% (1.2 m DMC) and 20.6% (1.2 m BMC), indicating more anions participating in solvation structure. The obtained results are consistent with those from [7]Li NMR spectra (Supplementary Fig. 15) and above MD simulations. Overall, the probability of $FSI^-$ presenting within $Li^+$ PSSs increases sequentially for 1.2 m DME, 1.2 m DMC and 1.2 m BMC. Since the composition of SEI largely depends on the $Li^+$ PSSs, more involvements of $FSI^-$ can facilitate the construction of mechanically and electrochemically stable LiF-rich SEI (Supplementary Fig. 16), contributing to the enhancement of Li plating/stripping CE[14,15]. Beyond that, the extent of $Li^+$-$FSI^-$ ion pairing also significantly affects Li electrode potential ($E_{Li}$), which can be revealed with the potential of ferrocene $(Fc)^+/Fc$ that is constant according to IUPAC recommendations[30,31]. As exhibited in Fig. 3g, the 10 mM Fc provides the highest redox potential ($E_{Fc}$, 3.41 V vs. $Li^+/Li$) in 1.2 m DME and shows the lowest one ($E_{Fc}$, 3.20 V vs. $Li^+/Li$) in 1.2 m BMC, indicating that $E_{Li}$ in 1.2 m BMC has been shifted upwards by 0.21 V compared to that in 1.2 m DME (Supplementary Fig. 17). The upward-shifted $E_{Li}$ can effectively shorten the gap between redox of $Li^+/Li$ and potential window of electrolyte, thereby diminishing the undesired electrolyte decomposition and considerably improving the Li anode CE[30]. All these results can well explain the inferior CE of 1.2 m DME than that of 1.2 m BMC shown in Fig. 3h and Supplementary Fig. 18. As for 1.2 m DMC, weighed down by the poor reductive stability of DMC itself, side reactions still proceed drastically (Supplementary Figs. 16 and 19) despite there is a fair number of $FSI^-$ involved in the solvation structure and the $E_{Li}$ is upshifted by 0.18 V (Supplementary Fig. 17), leading to a terrible Li CE of 9.9% (Fig. 3h) and poor cycling stability with Li||Li symmetric cells (Supplementary Fig. 20). Furthermore, even when physically blended with DME, the 1.2 m DME/DMC mixture only achieves a 0.07 V upshift on the $E_{Li}$ and a CE of 11.5%. Such CE is much lower than that obtained in the molecularly hybridized 1.2 m BMC (reaching 96%).

In the next logic step, linear sweep voltammetry (LSV) measurements on Al and Pt were both performed to evaluate the high-voltage tolerance of electrolytes (Fig. 3i). Whether on Al or Pt, 1.2 m DME begins to experience a gradual increase in current from 4.0 V vs. $Li^+/Li$ as a result of the intrinsic oxidative instability of DME. Despite undergoing physical blending with DMC, this inherent flaw persists as the weakest link, significantly constraining the high-voltage performance of 1.2 m DME/DMC. By contrast, with molecular introduction of carbonate group, the obtained BMC-based electrolyte presents an upgrade anodic limit comparable to that of 1.2 m DMC on Pt. While on Al, 1.2 m BMC even far outperforms the DMC-based electrolyte, which should be attributed to its another feature, excellent resistance to Al corrosion as the chronoamperometry (CA) tests and subsequent morphological investigations revealed in Supplementary Figs. 21 and 22. Hence, by virtue of enhanced oxidative stability and remarkable anti-corrosion to Al, 1.2 m BMC easily supports NCM811 at 4.3 V with a higher average CE over 200 cycles compared to DME and DMC counterparts (Supplementary Fig. 23), highlighting the importance of fundamentally changing the physicochemical properties of solvents at the molecular level.

## Li plating/stripping behavior and interphase chemistry

Generally, the realization of a long-term Li anode cycle with high CE depends on both of the reductively stable electrolyte and high-quality SEI. The conceived BMC solvent exhibits excellent reductive stability and relatively weak solvation affinity with $Li^+$ to permit $FSI^-$ into $Li^+$ PSSs, ensuring its superiority over DME and DMC counterparts. However, the resultant SEI seems insufficient to achieve a long-term stable cycle of Li plating/stripping (Supplementary Fig. 24). To make an optimization, the capability of BMC to dissolve a small amount of $LiNO_3$ has been utilized to produce SEI-favorable N-containing species ($LiN_xO_y$ and $Li_3N$)[32], combined with another film-forming additive lithium difluoro(bisoxalato) phosphate (LiDFBOP) to contribute beneficial P-containing species (P-O and P-F)[33,34] (Supplementary Fig. 25). The optimized electrolyte is denoted as 1.2 m BMC+ hereafter, which exhibits minimal variations in terms of oxidative stability, ionic conductivity, and $Li^+$ transfer number when compared to the additives-free 1.2 m BMC (Supplementary Fig. 26). For comparison, the ether-based 1.2 m DME+ is also formulated with same additives, whereas the commercial carbonate-based LB010+ is prepared with other two commonly-used additives (fluoroethylene carbonate (FEC) and vinylene carbonate (VC))[32,35] owing to its difficulty in dissolving $LiNO_3$. The detailed formulas are described in the Methods Section. The optimized 1.2 m BMC +, as shown in Fig. 4a, exhibits a remarkable Li plating/stripping CE of 99.4% over 10 cycles, surpassing the values obtained in 1.2 m DME+ (98.9%) and LB010+ (97.5%) as well as BMC-based electrolyte with single additive (Supplementary Fig. 27). Even after extending the cycling period to 125 cycles, the optimized BMC-based electrolyte consistently maintains a towering average Li CE of 99.3%, whereas 1.2 m DME+ and LB010+ fail to sustain adequately (Supplementary Fig. 28). The achieved Li plating/stripping CE in 1.2 m BMC+ even tops the list of previously reported ether-based and carbonate-based electrolytes whether they are at normal or high/locally high concentrations (Fig. 4b and Supplementary Table 4). More importantly, BMC-based electrolytes can exhibit excellent stability with Li||Li symmetric cells (Supplementary Fig. 29) and reliability as the calendar aging time (Supplementary Fig. 30) is extended and even the operating temperature (Supplementary Fig. 31) is increased.

Since the CE of Li anode is also closely related to its electrodeposit morphology[36], the Li plating in different electrolytes were recorded and analyzed with the digital photos and scanning electron microscopy (SEM) images. It can be detected that the Li electrodeposits in 1.2 m DME+ exhibit copious holes that can increase the contact area with the electrolyte and promote side reactions (Fig. 3c), while those in LB010+ consist of tiny Li particles that can easily induce the growth of

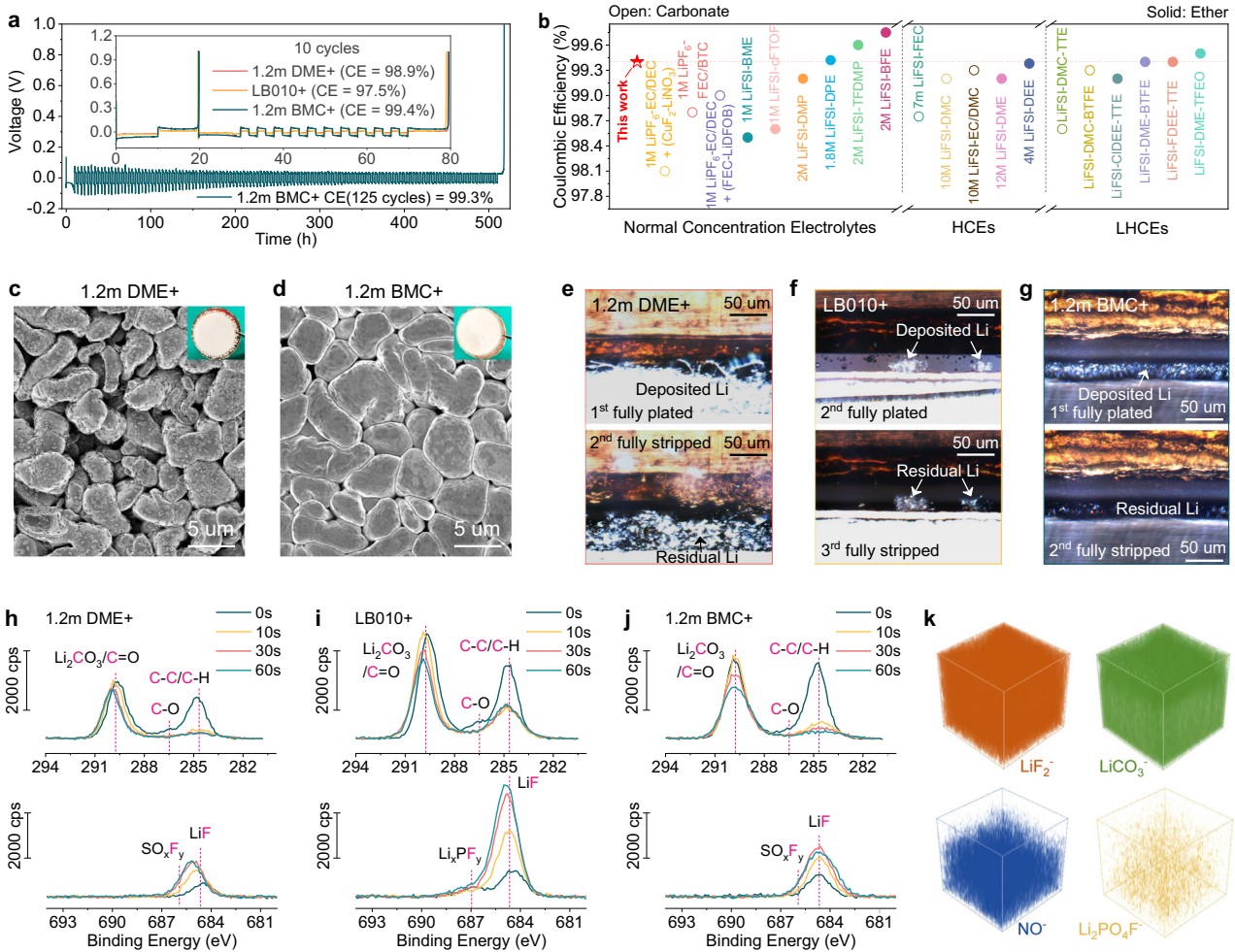

**Fig. 4 | Electrochemical performance of Li||Cu half cells, in situ ECCS observations, and chemical compositions of SEI. a** Long-term Li plating/stripping CE test performed at 0.5 mA cm⁻² and 1 mAh cm⁻² in 1.2 m BMC+ (The inset is results obtained within 10 cycles in various electrolytes). **b** Comparisons of Li plating/stripping CE with previously reported ether-based and carbonate-based electrolytes that are in the form of normal concentration electrolytes, HCEs and LHCEs (Additional details can be referenced in Supplementary Table 4). SEM and optical images for the surface of Li electrodeposits after the first plating on the Cu foil at 0.5 mA cm⁻² and 3 mAh cm⁻² in (**c**) 1.2 m DME + and (**d**) 1.2 m BMC +. In situ ECCS observations for evolution processes of Li plating/stripping in (**e**) 1.2 m DME +, (**f**) LB010+ and (**g**) 1.2 m BMC+ (In the case of LB010 +, since no Li plating occurred in the first cycle within the observed range, the test was specially extended to 3 cycles. Complete evolution processes of Li plating/stripping and more details can be found in Supplementary Fig. 34 and Supplementary Movies 6–8). $C_{1s}$ and $F_{1s}$ XPS patterns of the Cu substrates with various durations of Ar⁺ sputtering after one cycle of Li plating/stripping in (**h**) 1.2 m DME +, (**i**) LB010+ and (**j**) 1.2 m BMC +. **k** TOF-SIMS 3D reconstruction for the sputtered volume of the Cu substrate after one cycle of Li plating/stripping in 1.2 m BMC +.

dendrites (Supplementary Fig. 32). As a contrast, the optimized BMC-based electrolyte provides dense deposit composed of bulky particles and flat surface without dendrites (Fig. 3d and Supplementary Fig. 33). Such desired Li electrodeposit can effectively reduce the opportunity of parasitic reactions and achieve high CE of Li anode. In order to further visualize the evolution process of Li plating/stripping, in situ electrochemical confocal system (ECCS) test[37] was conducted in different electrolytes. As visualized in Fig. 4e, Supplementary Fig. 34a and Supplementary Movie 6, Li plating in 1.2 m DME+ presents a fluffy structure with dendrites and there is a lot of residual Li (made up of 'SEI Li' and 'dead Li') after fully stripped. When it comes to LB010+, it is not until the second cycle that scattered and mosslike Li plating are found, indicating rather uneven deposition (Fig. 4f, Supplementary Fig. 34b and Supplementary Movie 7). And the deposited Li also shows great irreversibility as expected. In sharp contrast, a more even, flatter and denser Li plating is observed in BMC-based electrolyte (Fig. 4g, Supplementary Fig. 34c and Supplementary Movie 8), which well corresponds to the SEM characterizations. Moreover, the amount of residual Li is minimal in 1.2 m BMC+ after complete stripping

compared to those in 1.2 m DME+ and LB010+ electrolytes, corresponding to its highest Li anode CE.

X-ray photoelectron spectroscopy (XPS) and time-of-flight secondary ion mass spectrometry (TOF-SIMS) were then performed to analyze the chemical composition and structure of SEI derived by BMC-based electrolyte. As the XPS depth profiles and corresponding $C_{1s}$, $F_{1s}$ patterns revealed in Fig. 4h–j and Supplementary Fig. 35, the outer layers of the SEI in all electrolytes are predominantly occupied by organic species, whereas inorganic species are more prevalent in the inner layers. Moreover, the presence of $Li_xPO_yF_z$, $LiNO_2$ and $Li_3N$ species confirms the involvement of LiDFBOP and LiNO₃ additives in SEI formations in 1.2 m DME+ and 1.2 m BMC+ (Supplementary Fig. 36). It is also worth noting that the LiF content in 1.2 m BMC+ exceeds that found in 1.2 m DME+ (Fig. 4h and j), which should be attributed to a higher degree of FSI⁻ involvement within Li⁺ solvation structures in BMC-based electrolytes. Regarding LB010 + -derived SEI, although it contains a substantial amount of desired LiF, it also comprises numerous organic constituents (Fig. 4i). Additionally, its TOF-SIMS chemical mapping reveals an uneven distribution of these

constituents (Supplementary Fig. 37), making it challenging to achieve satisfactory CE for the Li anode. By contrast, the SEI constituents are more evenly distributed in 1.2 m DME+ and 1.2 m BMC + . To further investigate the difference between 1.2 m DME + -derived SEI and 1.2 m BMC + -derived SEI, TOF-SIMS depth profiles and 3D reconstructions were also performed (Fig. 4k and Supplementary Figs. 38 and 39). Besides containing a higher concentration of LiF$_2^-$ fragments, the SEI derived from 1.2 m BMC+ exhibits a more uniform and comprehensive distribution of its constituents. These inherent advantages enable the 1.2 m BMC + -derived SEI to better induce uniform Li$^+$ flux for dendrite-free electrodeposit and suppress side reactions, thereby achieving a high CE during Li plating/stripping and excellent stability.

### Electrochemical and safety performance of practical LMBs
To further evaluate the feasibility of BMC in high energy batteries, practical LMBs are fabricated with limited-excess Li anode (100 μm, 20 mAh cm$^{-2}$) and high-loading-NCM811 cathode (24 mg cm$^{-2}$, 4.8 mAh cm$^{-2}$). On the basis of its high Li plating/stripping CE and the excelltent oxidation tolerance of optimized BMC-based electrolyte, the full cell utilizing 1.2 m BMC+ electrolyte presents an excellent performance with a high average CE of 99.8% over 150 cycles and

achieves a 92% capacity retention (Fig. 5a). In sharp contrast, shortened life spans are found for cells conducted in both routine ether-based and carbonate-based electrolytes. Only 51% and 49% of initial capacities are maintained in 1.2 m DME+ after 100 cycles and LB010+ after 55 cycles, respectively. The substantial capacity loss and pronounced increase in cell polarization observed in 1.2 m DME+ and LB010+ over the charge/discharge cycles can be attributed to their respective side reactions with the high-voltage cathode and active Li anode (Supplementary Figs. 40 and 41). In contrast, the full cell equipped with 1.2 m BMC+ achieves a high level of reversibility, and there is no significant increase in polarization observed during cycling (Fig. 5b), indicating minimal side reactions and stable interphases in optimized BMC-based electrolyte. Notably, the full cells based on 1.2 m BMC+ also performes pretty well even when subjected to more harsh conditions such as 4.4 V or 50 °C, showcasing the excellent tolerance of BMC at high voltage and temperature (Supplementary Figs. 42 and 43). Regretfully, due to the limited ionic conductivity of BMC-based electrolyte, the full cell exhibits an inferior rate capability compared to those operated in 1.2 m DME+ and LB010+ (Supplementary Fig. 44), which can be compensated by designing electrolyte formulations with co-solvents or diluents (Supplementary Fig. 45)[15,22].

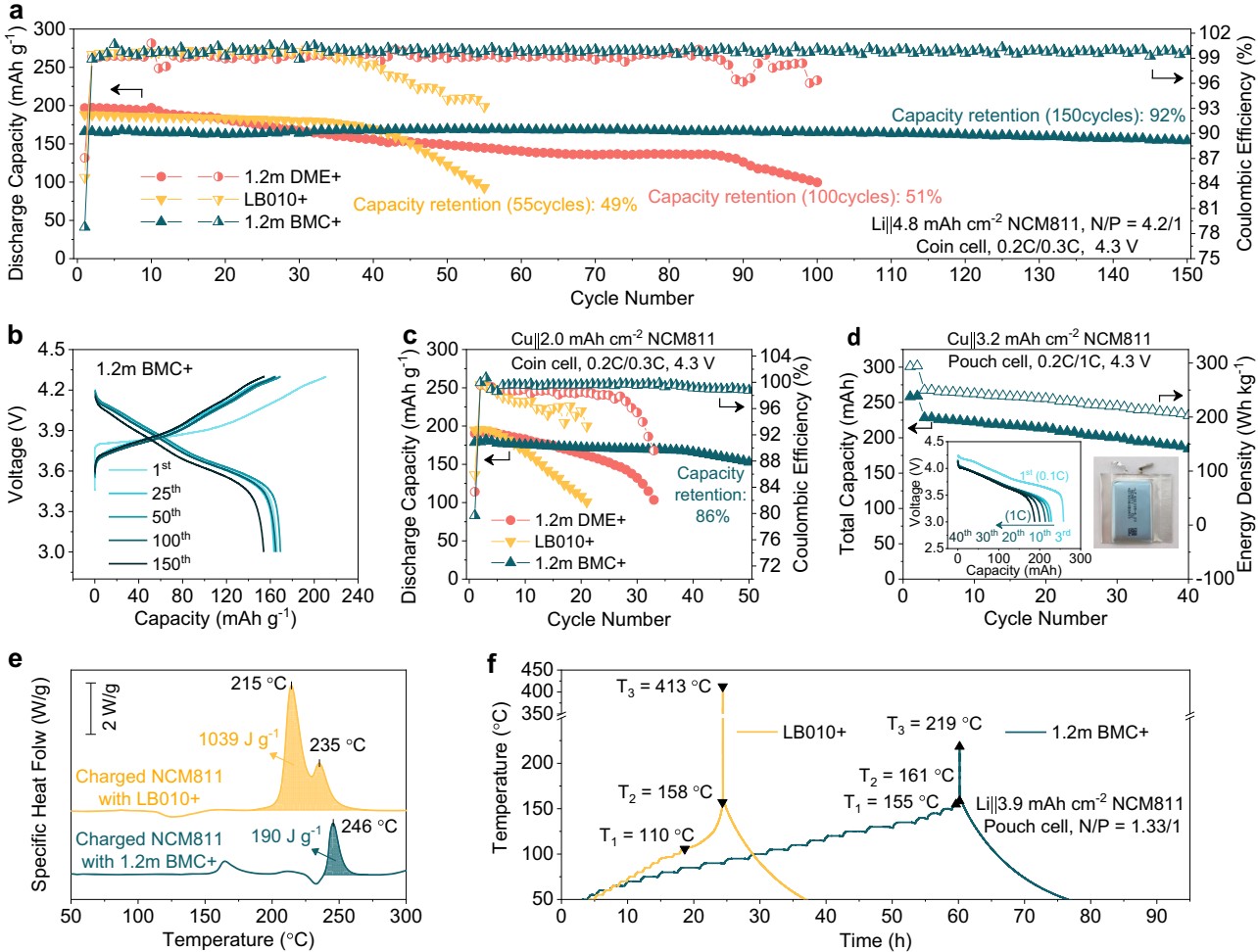

**Fig. 5 | Electrochemical performance and safety assessment of Li metal full cells. a** Cycling performance of 100 μm Li ||NCM811 full cells operated at 0.2 C charge / 0.3 C discharge with various lean electrolytes (7.3 mL Ah$^{-1}$) in the voltage range of 3–4.3 V. **b** Corresponding selected charge/discharge curves in 1.2 m BMC + . **c** Cycling performance of anode-free Cu ||NCM811 coin cells operated at 0.2 C charge / 0.3 C discharge with various lean electrolytes (5 mL Ah$^{-1}$) in the

voltage range of 3–4.3 V. **d** Cycling performance of the industrial anode-free Cu || NCM811 pouch cell operated at 0.2 C charge / 1 C discharge with lean 1.2 m BMC+ (2 g Ah$^{-1}$) in the voltage range of 3–4.3 V. **e** DSC curves of fully charged NCM811 with different electrolytes. **f** ARC results of fully charged Li ||NCM811 pouch cells with different electrolytes.

To make a further challenge, aggressive anode-free Cu||NCM811 coin cells have been fabricated with different electrolytes. As shown in Fig. 5c and Supplementary Fig. 46, the Cu||NCM811 cells with routine ether-based 1.2 m DME+ electrolyte witnesses a fast capacity fade with a capacity retention of only 69% after 30 cycles. The cycling performance even deteriorates in LB010+ electrolyte, which loses 2.25% per cycle and keeps mere 55% after 20 cycles. Regarding to the cell operated in 1.2 m BMC +, it exhibits extraordinarily stable cycling performance with a capacity retention of 86% after 50 cycles (Supplementary Fig. 47), surpassing the cell based on advanced LHCEs (Supplementary Fig. 48). Inspired by the excellent performance of the optimized BMC-based electrolyte, Cu||NCM811 pouch cells with a nominal capacity of 250 mAh were further applied with lean 1.2 m BMC+ (2 g Ah⁻¹) and cycled at a relatively high discharge rate of 1 C (Fig. 5d). Under these harsh conditions, the pouch cell can output a capacity of 230 mAh, and an impressive energy density of 250 Wh kg⁻¹ based on the total weight of the cell (Supplementary Fig. 49), which can be further improved if Ah-level pouch cells are performed. Additionally, a considerable energy retention of 82% is achieved after around 40 cycles.

LMBs are much more challenging in terms of safety than LIBs[38], so they desperately need intrinsically safe electrolytes. Therefore, the safety performance of BMC-based electrolyte and corresponding pouch cells under thermal/mechanical abuses were evaluated. As the DSC results shown in Fig. 5e, the exothermal peak of fully charged NCM811 mixed with 1.2 m BMC+ is delayed to 246 °C compared to commercial carbonate-based LB010 +. Moreover, the released heat in 1.2 m BMC+ is greatly reduced and only accounts for one fifth in LB010 +. The obtained result is similar to the recent report that comprehensively delves into the pivotal role of BMC in enhancing the safety of LIBs[22]. Figure 5f exhibits the accelerating rate calorimetry (ARC) results of fully charged Li||NCM811 pouch cells. It can be detected that the critical temperature $T_1$ for thermal safety is significantly increased to 155 °C from 110 °C, and the maximum temperature $T_3$ is almost halved. Moreover, the Ah-level Li||NCM811 pouch cell after being charged to 3.8 V in 1.2 m BMC+ comes through the nail penetration test without fire explosion, which contrasts with the failure observed in LB010+ (Supplementary Fig. 50, Supplementary Movies 9 and 10). All these results strongly prove that the conceived BMC solvent can significantly improve the intrinsic safety of LMBs. Together with its excellent performance on high-voltage cathode and Li metal anode, the optimized BMC-based electrolyte can realize high energy density and high safety at the same time, further demonstrating its efficacy for promoting advanced LMBs.

## Discussion

In this work, an advanced BMC solvent has been conceived and obtained via molecular hybridization, which regulates the electron distribution by integrating the electron-donating ether group with the electron-withdrawing carbonate group within a single molecule. Benefitting from its rational charge distribution, the linear BMC solvent exhibits considerable oxidative/reductive stability and relatively weak solvation ability. Moreover, the especial molecular structure of BMC also endows it with advantages including the ability to slightly dissolve LiNO₃, excellent thermostability and nonflammability. Optimized BMC-based electrolyte can enable a dendrite-free Li plating with a high CE of 99.4% and excellent high-voltage tolerance up to 4.4 V. As a result, it demonstrates satisfactory cycling performance for practical LMBs including limited-Li||(4.8 mAh cm⁻²) NCM811 full cells (150 cycles with a capacity retention of 92%) and even anode-free Cu|| NCM811 cells (50 cycles with a capacity retention of 86%). Benefitting from the inherent safety of BMC solvent, Li metal pouch cells exhibit impressive tolerance towards thermal/mechanical abuse, indicating that the optimized BMC-based electrolyte can well satisfy the strict requirements of high-energy and high-safety LMBs. This study showcases the potential of intramolecular hybridization between different solvent types, providing innovative design principles for the development of promising electrolytes in future energy-dense and safe LMBs.

## Methods

### Synthesis of BMC

Bis(2-methoxyethyl) carbonate (BMC, $C_7H_{14}O_5$) (Supplementary Figs. 51 and 52) was synthesized according to the method previously described[39]. More specifically, to a mixture of 114 g (1.5 mol) 2-methoxyethanol (99%, Alfa Aesar) and 67.5 g (0.75 mol) dimethyl carbonate (DMC, 99.9%, DodoChem), 0.7 g (13 mmol) sodium methoxide (NaOMe, 97%, Aladdin) was added under N₂ atmosphere. Subsequently, the mixture was stirred at 95 °C overnight. Following this, generated methanol was removed via rotary evaporation and the reaction was terminated by introducing water. The resulting solution was then transferred to a separatory funnel and extracted with dichloromethane (3 × 50 mL). The obtained organic phase was dried using anhydrous MgSO₄ and then vacuum evaporated on rotavapor to remove volatiles. The crude product was vacuum distilled (4 mm Hg, 103 °C) three times to yield ~40 g colorless liquid as the product. Yield: 30%. ¹H NMR (400 MHz, DMSO-d6) δ 4.19 (dd, J = 5.4, 3.7 Hz, 4H), 3.52 (dd, J = 5.4, 3.7 Hz, 4H), 3.26 (s, 6H). ¹³C NMR (100 MHz, CDCl₃) δ 154.99 (s), 69.99 (s), 66.71 (s), 58.70 (s).

### Electrolyte preparations

Cell-grade LiFSI, DME, DMC, FEC, VC, TTE and commercial carbonate electrolyte LB010 (1 M LiPF₆ in EC:DMC:EMC = 1:1:1 wt.%) were procured from DodoChem. LiNO₃ (99.99% metals basis) and ferrocene (Fc, 99%) were purchased from Aladdin. LiDFBOP (98.7%) was provided by Shanghai Rolechem. Prior to preparations of electrolytes, solvents were dried by 4 Å zeolites to make sure the water content was lower than 20 ppm, which was tested by 831 KF Coulometer (Metrohm). 2.244 g LiFSI was dissolved into 10 g of DME, DMC, BMC and DME/DMC (1/1 by weight) to obtained 1.2 m DME, 1.2 m DMC, 1.2 m BMC and 1.2 m DME/DMC electrolytes, respectively. Additives-containing 1.2 m BMC+ electrolyte was prepared by incorporating 0.75 wt.% LiNO₃ and 1 wt.% LiDFBOP into 1.2 m BMC, while additives-containing 1.2 m DME+ electrolyte was obtained by introducing the same amounts of LiNO₃ and LiDFBOP into 1.2 m DME. Additionally, additives-containing LB010+ electrolyte was prepared by adding 2 wt.% FEC and 2 wt.% VC to LB010. All preparations were conducted inside an Ar-filled glove box (MIK-ROUNA) with O₂ and H₂O contents below 0.1 ppm.

### Theoretical calculations

The molecular geometries were optimized using DFT at the B3LYP/6-311 + + G(d) level, and frequent analyses were conducted at the same level to ensure each stationary point was properly optimized. To simulate a liquid environment, the continuous polarization medium (PCM) model with an acetone dielectric constant was employed. All DFT calculations were performed on Gaussian 09 software package. Ionization energy refers to the energy difference between the investigated structures before and after oxidation, while electron affinity energy refers to the energy difference between the investigated structures before and after reduction. Higher ionization energy indicates better oxidative stability, whereas more negative electron affinity energy suggests poorer reductive stability. Binding energy refers to the difference in energy between investigated structures before and after coordination. A more negative binding energy indicates a stronger interaction between the structures. The electrostatic potential (ESP) of various solvent molecules was calculated using Multiwfn 3.5 and VMD 1.9.3 programs.

Molecular dynamics (MD) simulations were conducted with the Forcite module in Material Studio software, utilizing electrolyte molar ratios obtained from experimental results. The COMPASS force field was selected for all molecular dynamic simulations, with a fixed time

step of 1.0 femtosecond (fs). The systems were equilibrated for at least 20 picoseconds (ps) in the NVT ensemble, followed by production runs conducted in an NPT ensemble for 200 ps using the Berendsen barostat to maintain a pressure of 0.1 GPa with a decay constant of 0.1 ps. All steps were conducted under a Nosé thermostat with target temperatures set at 298 K. The simulation time was sufficient to ensure that the electrolyte system reached equilibrium. The reliability of the MD simulation results was confirmed by comparing the calculated densities of electrolytes from simulation with their experimentally measured counterparts as follows: 1.2 m $DME_{(MD)} = 0.964$ g cm$^{-3}$ compared to 1.2 m $DME_{(Exp)} = 0.956$ g cm$^{-3}$, 1.2 m $DMC_{(MD)} = 1.148$ g cm$^{-3}$ compared to 1.2 m $DMC_{(Exp)} = 1.137$ g cm$^{-3}$, and 1.2 m $BMC_{(MD)} = 1.210$ g cm$^{-3}$ compared to 1.2 m $BMC_{(Exp)} = 1.202$ g cm$^{-3}$, respectively, with discrepancies remaining within a margin of less than 1%.

## Material characterizations

$^1$H and $^{13}$C NMR spectra were acquired using a 400 MHz NMR spectrometer (AVANCE NEO, Bruker) and $^7$Li NMR spectra were obtained on a 600 MHz NMR spectrometer (AVANCE NEO, Bruker) at 25 °C. Raman spectra were carried out on an automated confocal Raman microscopy system (WITec Apyron, WITec) with an excitation wavelength of 532 nm and a laser power of 50 mW. The spectral data were analyzed using the WITec Project Plus software. To investigate the morphology of deposited Li, cycled NCM811 cathode and Al collectors, SEM images were obtained through Hitachi S4800 (Hitachi) with the accelerating voltage of 1 kV. In order to investigate the behavior divergence of Li plating, in-situ optical observation was conducted using an electro-chemical reaction visualizing confocal system (ECCS B310, Lasertec). The in-situ cells were assembled using Cu foil (working electrode), Li foil (counter and reference electrode) and Celgard 2400 separator in the clamp with a certain pressure. The cells were then cut into cross-sections with special cutting tools for observation. The current of Li plating/stripping was fixed at 0.5 mA, while the Li plating capacity was fixed at 0.25 mAh. For XPS and TOF-SIMS measurements, Cu electrodes retrieved from Li||Cu cells after one cycle of plating/stripping at 0.5 mA cm$^{-2}$ and 5 mA cm$^{-2}$ were transferred to instrumental chambers from an Ar-filled glove box without exposure to air. XPS depth profiles were collected on a Nexsa instrument (Thermo Fisher Scientific) using a monochromatic Al Ka X-ray source (excitation energy = 1468.6 eV) and Ar$^+$ sputtering at 1 kV for 0, 10, 30 and 60 s. TOF-SIMS analyses were carried out on an IONTOF M6 (IONTOF) instrument in high mass resolution mode, utilizing a 30 keV Bi$^{3+}$ ion beam for the acquisition phase and a 2 keV Cs$^+$ ion beam for the sputter phase. The typical areas analyzed and sputtered were 100 μm × 100 μm and 300 μm × 300 μm, respectively. Prior to conducting morphology and composition characterizations, all retrieved electrodes underwent a triple rinse with DME in order to eliminate any residual electrolyte. Subsequently, the electrodes were dried for 12 h at room temperature. The viscosity tests were conducted on a rotational rheometer (HAAKE MARS III). TG curves of the electrolytes (-10 mg per sample) were recorded using a heating rate of 10 °C min$^{-1}$ from 30 to 130 °C on TG 201 F1 Libra (Netzsch). DSC measurements were conducted using a DSC 200 F3 Maia (Netzsch) instrument with N$_2$ as the method gas. The DSC curve of pure BMC was recorded at a heating rate of 5 °C min$^{-1}$ from -50 to 250 °C. Prior to testing, -10 mg of the sample was sealed in a stainless-steel sample pan and weighed, followed by cooling down to -50 °C and maintaining for 5 min with liquid nitrogen as coolant. DSC curves of cycled NCM811 powders (around 10 mg) mixed with different electrolytes (around 10 mg) were recorded at a heating rate of 5 °C min$^{-1}$ from 25 to 300 °C. Before test, the NCM811 powders were retrieved from 100 μm Li||NCM811 full coin cells that were fully charged to 4.3 V after 2 formation cycles and then sealed in a stainless-steel sample pan with the electrolyte. The flash points of electrolytes were investigated on flash point analyzers (MINIFLASH

FPH VISION (10 - 400 °C) and MINIFLASH FP VISION (−45 - 120 °C), Grabner Instruments) according to ASTM D6450 standard. All measurements were conducted using a sample volume of 1 mL and repeated for three times to ensure the reliability. The initial temperature was typically set at 18 °C below the anticipated flash point, and the heating rate was maintained at 5.5 ± 0.5 °C per minute. An electric arc with high voltage was applied as the ignition source, operating at a frequency of 1 °C per minute. Following each flash test, a short air pulse from a small membrane compressor introduced -1.5 ± 0.5 mL of air into the test chamber. Flash detection in the chamber relied on detecting an increase in pressure after ignition occurs. The limit for successful flash point detection was set at a pressure increase of 20 kPa, which corresponded to approximately a flame volume of 1.5 mL above barometric pressure. Further detailed testing process can be referred in previous literature[40]. Accelerating rate calorimetry (ARC, THT) was utilized to monitor the real-time temperature changes of pouch cells. The test temperature range was set at 30–300 °C. Before ARC test, industrial Li||NCM811 (500 mAh, N/P = 1.33/1, Electrolyte/Cathode = 2.5 g Ah$^{-1}$) pouch cells were charged to 4.3 V after 2 formation cycles. The nail penetration test for pouch cells was conducted at room temperature in an open-air environment using a M01-005 nail penetration testing system (Dongguan Saice). Before nail penetration test, 5 Ah Li||NCM811 (4 mAh cm$^{-2}$, N/P = 2.5/1, Electrolyte/Cathode = 2 g Ah$^{-1}$) pouch cells were charged to 3.8 V after 2 formation cycles. The cell was vertically penetrated by a steel nail with a diameter of 3 mm at a speed of 25 mm s$^{-1}$ during the test. The ionic conductivity of electrolytes at different temperatures was measured and calculated via electrochemical impedance spectroscopy (EIS) using two stainless-steel sheets (1 cm$^2$) symmetrically placed in the electrolytes.

## Electrochemical measurements

The anodic stability of electrolytes was determined by analyzing the LSV curves of Al and Pt in a three-electrode device with lithium metal serving as both counter and reference electrodes, using a scan rate of 1 mV s$^{-1}$. To acquire the Li$^+$ transference number of electrolytes, a constant voltage bias of 10 mV was applied to Li||Li symmetric cells. For Al corrosion testing, Li||Al half-cells were subjected to various voltages for 22 h, with each voltage being maintained for 2 h. The Li electrode potentials ($E_{Li}$) in various electrolytes were revealed with the CV curves of ferrocene (Fc), which were obtained using three-electrode devices consisting of Pt as a working electrode and lithium metal as both counter and reference electrodes, at a scan rate of 5 mV s$^{-1}$. It was assumed that the potential of Fc$^+$/Fc remained constant according to IUPAC recommendations[30,31]. The LSV, CV, EIS and CA measurements were all conducted on VMP3 potentiostats (BioLogic). For CE tests of Li plating/stripping, Li||Cu (CIVEN Metal, 10 μm) cells were assembled, and tested following a similar protocol as previously reported[5,6]. Specifically, an initial formation cycle was conducted with a plating/stripping capacity of 5 mAh cm$^{-2}$ at 0.5 mA cm$^{-2}$, followed by the plating of 5 mAh cm$^{-2}$ Li ($Q_p$) on Cu at 0.5 mA cm$^{-2}$ as a reservoir. A fixed capacity ($Q_c = 1.0$ mAh cm$^{-2}$) of Li was then repeatedly stripped/plated at 0.5 mA cm$^{-2}$ for n cycles, and the residual Li ($Q_r$) was subsequently stripped to 1 V at the same current density. The average CE can be calculated using the following equation:

$$CE = \frac{nQ_c + Q_r}{nQ_c + Q_p}$$

For calendar ageing test, after 10 cycles of normal Li plating/stripping in Li||Cu cells at 0.5 mA cm$^{-2}$ and 1 mAh cm$^{-2}$, different ageing intervals (10 h, 24 h, 48 h and a week) were introduced in every alternate cycle when 1 mAh cm$^{-2}$ Li had been plated. For Li||NCM811 (thickness: 24 μm, areal capacity: 1.2 - 1.3 mAh cm$^{-2}$) half cells, 100 μm Li||NCM811 (thickness: 82 μm, areal capacity: 4.8 mAh cm$^{-2}$) full coin cells, and anode-free Cu||NCM811 (thickness: 36 μm, areal capacity:

2.0 mAh cm$^{-2}$) coin cells, the cathode (diameter: 12 mm, area: 1.13 cm$^2$) was prepared by mixing NCM811 powder (Shenzhen Kejing Star Technology Co. Ltd.), acetylene black, and polyvinylidene difluoride (PVDF) in a ratio of 96:2:2 using N-methyl-2-pyrrolidone (NMP) as the solvent. All NCM811 coin cells were cycled at a charge rate of 0.2 C and discharge rate of 0.3 C (1 C = 200 mAh g$^{-1}$) between voltage ranges of 3 V to 4.3 V or up to 4.4 V. For the coin cell configurations, each CR-2032 type coin cell was fabricated using Celgard 2400 (polypropylene, thickness: 25 μm, diameter: 18 mm) as the separator and 50 μL electrolyte (unless otherwise specified). To suppress corrosion from the electrolyte, Al-Clad coin cell case (Hohsen) was used for NCM811 cathode. Both thick (1000 μm) and thin (100 μm) Li foils were purchased from China Energy Lithium Co., Ltd. Industrial Li||NCM811 and anode-free Cu||NCM811 pouch cells were purchased from Li-Fun Technology Co. Ltd. Following two formation cycles at a 0.1 C charge/discharge, pouch cells underwent cycling at a 0.2 C charge/1 C discharge between 3 and 4.3 V. Additionally, pouch cell cycling was conducted using a constant-current-constant-voltage protocol (cells were charged to the maximum voltage and held until current dropped below 0.1 C) together with a pressure of 200 kPa. The CE measurements of Li plating/stripping and cycling performance of cells were carried out on a standard battery tester (CT4008, Neware).

## Reporting summary

Further information on research design is available in the Nature Portfolio Reporting Summary linked to this article.

## Data availability

The data generated in this study are provided in the Supplementary Information/Source data file. Source data are provided with this paper.

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

## Acknowledgements
This work is supported by the National Key Research and Development Program of China (2022YFB2402200 X.D.), National Natural Science Foundation of China (21935003 Y.X., 22109028 X.D., 22379028 X.D.), Natural Science Foundation of Shanghai (22ZR1404400 X.D.), Chenguang Program sponsored by Shanghai Education Development Foundation and Shanghai Municipal Education Commission (19CG01 X.D.).

## Author contributions
X.D. and Y.X. conceived the idea and designed the experiments. J.C. and J.X. synthesized the molecule. J.C. carried out material characterizations and electrochemical measurements. D.Z. performed in situ ECCS optical observations. L.Z. conducted ARC and nail penetration tests. M.L. conducted DFT calculations. T.Z. performed MD simulations. All authors engaged in result discussions. J.C., J.L., F.W., Y.W., X.D. and Y.X. co-wrote the manuscript with input from all authors.

## Competing interests
A patent related to the work has been submitted (application number CN202210168155.0) by Fudan University. The inventors are X.D., J.C., J.X. and Y.X. The patent refers to the methodology described in this paper, but it offers a greater variety of analogous solvents compared to this work. Other authors declare no competing interests.
