## [Peer Review File · Nature Communications]

Hybridizing Carbonate and Ether at Molecular Scales for High-energy and High-safety Lithium Metal BatteriesREVIEWER COMMENTS

Reviewer #1 (Remarks to the Author):

The author showed that a BMC-based electrolyte can facilitate better cycling stability than DME and DMC. Although the manuscript showed a clear benefit of using BMC for Li metal cycling with the aid of additives, the results on solvation structures are not conclusive enough to support the author's claim. Also, the novelty of the current work is overstated in the manuscript without adequately acknowledging the previous work using the same solvent. Therefore, I cannot recommend the acceptance of the present paper in this form for publication in this prime journal unless the following issues can be settled.

1. For constructive progress of the common knowledge in the research field, it is better to acknowledge what has already been disclosed, and the author's description of BMC as a "novel solvent" is inappropriate. The previous paper in EES (published in May 2023) already reported BMC-based electrolytes using the same synthetic method, in which its physical properties, solvation structures, ionic conductivities, electrochemical performances, and thermal stability have been comprehensively included. Thus, the novelty of this work should be carefully claimed in the manuscript, particularly in the Abstract and Introduction. Although BMC has already been reported as an electrolyte solvent, its utilization in Li metal batteries is novel, which is not clearly stated in the current manuscript.
2. "Microsolubility" is not a conventional term.
3. Figure 2c only shows two possible conformations of Li-bound BMC. The mutual coordination of carbonyl and ether oxygens with Li should be considered in this comparison. The marginal solubility of LiNO₃ only reflects the limited population of five-membered ring chelation but does not directly imply that the carbonyl oxygen is the primary coordinating site when considering other possible coordination structures.
4. The measuring limit of the instrument is lower than the reported flash point of BMC (121C) in the literature, which would give an inaccurate value in this study.
5. Provide values for the calculated and measured densities for the electrolyte to confirm the reliability of the MD simulation.
6. The deconvoluted peaks in the Raman spectra for the same coordination species do not seem to have the same FWHM and peak center, which cannot give a reasonable population estimate. Although the peaks for DMC and BMC seem nearly identical when directly overlapping each other, the calculated proportion is largely different. It is recommended to provide fitting details.
7. Also, the results in supplementary Fig 10, which identifies the dominant solvation site, should be compared with other solvent systems. The dramatic change of the peak intensities for C-O stretching vibrations in supplementary Fig 10a needs to be explained.
8. The oxidative stability test should be conducted with BMC with LiNO₃ and LiBF₄BOP additives.
9. It is hard to estimate the thickness of Li electrodeposits from BMC+ from Supplementary Fig 25C.
10. The nail penetration test result using BMC should be compared with that of other electrolytes to see a clear improvement. In fact, 1-Ah pouch cells usually do not explode even with conventional electrolytes.
11. The comparison of Li plating/stripping CE with carbonate-based and ether-based electrolytes in Fig. 4b needs to be updated. The recent reports (Ali Coskun, Nat. Commun., 2023, 14, 299; Vilas G. Pol, Nat. Commun, 2023, 14, 868; Jun Lu, Nat. Commun., 2023, 14, 1081) have introduced the ether-based electrolytes with a comparable CE.

Reviewer #2 (Remarks to the Author):

In this manuscript, the authors reported a molecular hybridization strategy aimed at developing a comprehensive solvent suitable for high-energy and high-safety lithium metal batteries. By ingeniously

incorporating carbonate groups and ether groups with contrasting properties into a single molecule, the synthesized BMC solvent reasonably modulates the electron distribution and thereby demonstrates considerable oxidation and reduction stability. Moreover, the unique molecular structure also imparts the BMC with many other merits such as relatively weak solvating power, LiNO₃-microsolubility, excellent thermostability, and nonflammability. Paired with 1.2m LiFSI, the optimized single-solvent BMC-based electrolyte exhibits both impressive Li plating/stripping CE and fairish high-voltage tolerance. Consequently, satisfactory cycling performance is fulfilled for Li||high-loading-NCM811 (4.8 mAh cm⁻²) full cells as well as anode-free Cu||NCM811 coin/pouch cells. More importantly, in the case of thermal/mechanical abuses, Li metal pouch cells using BMC also demonstrate reliable safety. The manuscript is well-structured, and the proposed strategy demonstrates brilliance and ease of implementation. Undoubtedly, this work will have a positive impact on innovative research and development in the field of safe and high-energy Li metal batteries, as well as other rechargeable alkali metal systems. Therefore, I highly recommend accepting it for publication in this journal. However, some minor revisions are necessary to further enhance the quality of this manuscript.

Question 1: In Fig. 1, the authors described the physicochemical properties of a representative ether solvent (DME) and a representative carbonate solvent (DMC), as well as the alterations in their properties after physical blending and molecular hybridization. What perplexes me is why the oxidation and reduction stability of the mixture are both poor after physical blending.

Question 2: The authors introduced CH₃OCH₂- moieties at both ends of DMC to extend the oxyalkyl chain and enhance intermolecular forces, thereby achieving a high boiling point and flash point for BMC (Fig. 2d) to ensure battery safety. However, it is important to note that the increase in intermolecular forces may also impact the solvent's viscosity, thus I recommend conducting viscosity testing.

Question 3: As the authors pointed out: "The upward-shifted ELi can effectively shorten the gap between redox of Li⁺/Li and potential window of electrolyte, thereby diminishing the undesired electrolyte decomposition and considerably improving the Li anode CE". Therefore, they utilized CV tests (Fig. 3g) of 10 mM ferrocene (Fc) to compare the ELi in 1.2m DME, 1.2m DMC and 1.2m BMC electrolytes. To further highlight the difference between molecular hybridization and physical blending, an evaluation of ELi in 1.2m DME/DMC should also be conducted.

Question 4: Whether the physicochemical properties of electrolytes (such as ionic conductivity and Li transport number) will be affected by the addition of LiNO₃ and LiDFBOP?

Question 5: Upon further optimization with both LiNO₃ and LiDFBOP additives, 1.2m BMC+ electrolyte fulfills an impressive Li plating/stripping CE of 99.4% (Fig. 4a). Can the same effect be achieved by adding only one of these additives?

Question 6: Fig. 4e-4g present in-situ ECCS observations for evolution processes of Li plating/stripping to prove the advancement of optimized BMC-based electrolyte. However, the number of process snapshots is insufficient for comprehensive and intuitive comparisons, as only the initial plating and final stripping stages are captured. Therefore, it would be beneficial to include more detailed process snapshots in Supplementary Information.

Reviewer #3 (Remarks to the Author):

Conventional ether and carbonate-based electrolytes have significant limitations in high-voltage Li metal batteries. In this work, Xia et al. reported the hybridization of the ether and carbonate functional moieties to form bis(2-methoxyethyl) carbonate (BMC). With the additives, BMC-based electrolyte showed the improved performance. However, the same solvent structure had been

published in the journal of EES (10.1039/D3EE00157A), thus significantly reducing the novelty of this work. Therefore, I do not recommend to publish this paper in Nature Communications.

1. The structure after hybridization showed better performance compared to DME and DMC, but the electrolyte in the full cell test still needs LiNO₃ and LiDFBOP additive, which implies the BMC solvent still has obvious side reactions.
2. The authors claimed the electron distribution change after hybridization, the steric effect of BMC should also be considered.
3. Compared to DME and DMC, BMC shows extremely low ionic conductivity which leads to the higher polarization and lower specific capacity.
4. Why does BMC show non-flammability?
5. Symmetrical cell with each electrolyte should be measured to compare the polarization during long cycling.
6. Considering the low ionic conductivity of BMC electrolyte, the rate performance of full cells with these electrolytes need be evaluated.
7. I suggest authors add N/P ratio of all full cells in figure.

Response Letter to Reviewers

Dear Reviewers,

We sincerely appreciate your evaluation of our manuscript and the valuable suggestions you provided. These comments have been instrumental in guiding us to improve the overall quality of our work. In response to your valuable advice, we meticulously addressed each point in the response letter and implemented revisions in the revised manuscript and supplementary information. These modifications are highlighted in different colors for your convenience. We trust that our detailed responses adequately address your concerns and hope that the revised versions meet your expectations. Once more, we are grateful for your efforts and constructive advice, which have significantly contributed to enhancing our manuscript.

Your sincerely,

The Author Team

Response to Reviewer #1

Overall comment: The author showed that a BMC-based electrolyte can facilitate better cycling stability than DME and DMC. Although the manuscript showed a clear benefit of using BMC for Li metal cycling with the aid of additives, the results on solvation structures are not conclusive enough to support the author's claim. Also, the novelty of the current work is overstated in the manuscript without adequately acknowledging the previous work using the same solvent. Therefore, I cannot recommend the acceptance of the present paper in this form for publication in this prime journal unless the following issues can be settled.

Response: We express our sincere gratitude for your generous allocation of time to meticulously review our manuscript. Your insightful and profound comments have significantly enhanced the quality of our manuscript. According to your valuable suggestions, we have prepared a comprehensive point-by-point response to address these questions as follows. Corresponding revisions have also been implemented in the manuscript and supplementary information, highlighted with **green** background for easy identification. We truly appreciate your constructive advices and hope these detailed responses can well address your concerns.

Question 1: For constructive progress of the common knowledge in the research field, it is better to acknowledge what has already been disclosed, and the author's description of BMC as a "novel solvent" is inappropriate. The previous paper in EES (published in May 2023) already reported BMC-based electrolytes using the same synthetic method, in which its physical properties, solvation structures, ionic conductivities, electrochemical performances, and thermal stability have been comprehensively included. Thus, the novelty of this work should be carefully claimed in the manuscript, particularly in the Abstract and Introduction. Although BMC has already been reported as an electrolyte solvent, its utilization in Li metal batteries is novel, which is not clearly stated in the current manuscript.

Response: Great thanks for your kind recognition of the novelty about our BMC-based electrolyte in Li metal batteries and meanwhile we apologize for the misleading description. The recent report of BMC is significant in realizing safe Li-ion batteries. It offers a viable way to balance the flammability and ionic conductivity of linear organic carbonates by combining alkyl-chain extension and alkoxy substitution, which modified the commonly-used carbonate solvent DEC (low flash point and flammable) to nonflammable BMC with enhanced flash point. Moreover, it also helps us better understand the as-fabricated high-safety lithium metal batteries based on the BMC solvent in our manuscript. Exactly as what you kindly said, the utilization of BMC in Li metal batteries is novel, which we should clearly state in the manuscript. Therefore, we have accordingly modified the descriptions to avoid the misleading "novel solvent" and emphasize the novel use of BMC in Li-metal batteries in the Abstract and Introduction, which are listed as follows and

highlighted with green background in the revised manuscript.

Updates in the Revised Manuscript:

(1) *Line 21 in the Abstract of the revised manuscript*

The description: “In this context, a novel solvent, bis(2-methoxyethyl) carbonate (BMC), is conceived via innovative molecular hybridization of the ether and carbonate functional groups.” has been modified as: “Herein, a linear functionalized solvent, bis(2-methoxyethyl) carbonate (BMC), is conceived by intramolecularly hybridizing ethers and carbonates.”

(2) *Line 76 in the Introduction of the revised manuscript*

The description: “In this way, a novel linear solvent, bis(2-methoxyethyl) carbonate (BMC), was conceived and synthesized successfully, exhibiting considerable redox stability and relatively weak solvation power. Beyond that, the deliberate elongation of the molecular chain for increasing the intermolecular van der Waals forces brings about an elevated thermal stability and a heightened flash point for BMC as expect.” has been modified as: “In this way, an intramolecularly hybridized linear solvent, bis(2-methoxyethyl) carbonate (BMC), was conceived and synthesized successfully, exhibiting considerable redox stability and relatively weak solvation power. Beyond that, the deliberate elongation of the molecular chain for increasing the intermolecular van der Waals forces brings about an elevated thermal stability and a heightened flash point for BMC as expect. Despite the recent report on the use of BMC in safe graphite-based LIBs²², we initiated a fundamental understanding on the molecular hybridization and a comprehensive investigation of BMC for hyperactive Li metal anode and high-voltage LMBs.” to acknowledge the authors of EES (*Energy Environ. Sci.*, 2023, 16, 2924) and highlight the novelty of our work in LMBs.

Question 2: "Microsolubility" is not a conventional term.

Response: Thanks for your kind remind. We sincerely apologize for the utilization of unconventional terms. According to your suggestion, the phrase "LiNO₃-microsolubility" has been revised to "the ability to slightly dissolve LiNO₃" in the revised version highlighted with green background.

Updates in the Revised Manuscript:

(1) *Line 25 in the Abstract of the revised manuscript*

The description: “Furthermore, BMC also offers advantages including LiNO₃-microsolubility, excellent thermostability and nonflammability.” has been revised as “Furthermore, BMC also offers advantages including the ability to slightly dissolve LiNO₃, excellent thermostability and nonflammability.”

(2) *Line 142 in the Results of the revised manuscript*

The description: “Moreover, besides the enhanced thermal stability and fire resistance achieved through intentionally lengthening molecular chain, BMC unexpectedly demonstrates other merits like relatively weak solvating power and LiNO₃-microsolubility.” has been modified as “Moreover, besides the enhanced thermal stability and fire resistance achieved through intentionally lengthening molecular chain, BMC unexpectedly demonstrates other merits like relatively weak solvating power and the ability to slightly dissolve LiNO₃.”

(3) Line 442 in the Discussion of the revised manuscript

The description: “Moreover, the unique molecular structure of BMC also endows it with advantages including LiNO₃-microsolubility, excellent thermostability and nonflammability.” has been revised as “Moreover, the unique molecular structure of BMC also endows it with advantages including the ability to slightly dissolve LiNO₃, excellent thermostability and nonflammability.”

Question 3: Figure 2c only shows two possible conformations of Li-bound BMC. The mutual coordination of carbonyl and ether oxygens with Li should be considered in this comparison. The marginal solubility of LiNO₃ only reflects the limited population of five-membered ring chelation but does not directly imply that the carbonyl oxygen is the primary coordinating site when considering other possible coordination structures.

Response: Thank you sincerely for your valuable advice, and we deeply apologize for our inadvertent oversight. According to your suggestion, we have taken into consideration some other possible main configurations of Li-bound BMC. Four possible structures of different coordination configurations are summarized in **Fig.(answer) 1**, including the coordination sites of carbonyl oxygens with Li⁺ (Li⁺-BMC complex-1, **Fig.(answer) 1a**), the five-membered ring chelation with Li⁺ through two oxygen atoms (Li⁺-BMC complex-2, **Fig.(answer) 1b**), the scenario where Li⁺ is coordinated with two single bond oxygens in the carbonate group (Li⁺-BMC complex-3, **Fig.(answer) 1c**), as well as the mutual coordination of carbonyl and ether oxygens with Li⁺ as you proposed (Li⁺-BMC complex-4, **Fig.(answer) 1d**). The binding energy of different configurations has also been calculated and compared to exhibit their stability. It can be detected that the mutual coordination of two single bond oxygens in the carbonate group with Li⁺ (Li⁺-BMC complex-3, **Fig.(answer) 1c**) only exhibits a weak binding energy of -7.48 kJ mol⁻¹. By contrast, the mutual coordination of carbonyl and ether oxygens with Li⁺ (Li⁺-BMC complex-4, **Fig.(answer) 1d**) demonstrates a stronger binding energy of -34.48 kJ mol⁻¹, even surpassing that of the five-membered ring chelation structure coordination configuration (-23.82 kJ mol⁻¹, Li⁺-BMC complex-2, **Fig.(answer) 1b**). In spite of its strong binding energy with Li⁺, one important thing to be considered is that achieving such coordination configuration of Li⁺-BMC complex-4 requires BMC molecule to be a distorted conformation. As displayed in **Fig.(answer) 2**, there are two conformations for BMC structure, which are the commonly-recognized linear conformation (BMC-structure-1) and the distorted one (BMC-structure-2). The latter is obviously unfavorable owing to its relatively positive electronic energy value, 6.85 kJ mol⁻¹.

¹ higher than the BMC-structure-1. This indicates that the population of BMC-structure-2 is a minority in the whole electrolyte. Consequently, it can be reasonably inferred that the mutual coordination of carbonyl and ether oxygens with Li⁺ (Li⁺-BMC complex-4), which is based on BMC-structure-2, may not be the most dominant coordination configuration between Li⁺ and BMC.

Fig.(answer) 1 Optimized structures and binding energy of (a) Li⁺-BMC complex-1 (Li⁺ coordinates with carbonyl O atom), (b) Li⁺-BMC complex-2 (Li⁺ coordinates with one single bond O atom in the carbonate group and one ether O atom), (c) Li⁺-BMC complex-3 (Li⁺ coordinates with two single bond O atoms in the carbonate group) and (d) Li⁺-BMC complex-4 (Li⁺ coordinates with one carbonyl O atom and one ether O atom) that are depicted by the ball-and-stick model. Purple balls stand for Li⁺, while gray, white and red balls stand for C, H and O atoms, respectively.

Fig.(answer) 2 Different kinds of optimized BMC structures and their respective electronic energy. Gray, white and red balls stand for C, H and O atoms, respectively.

Besides, the limited solubility of LiNO_3 also implies the conformation of Li^+ -BMC complex. It is commonly known that LiNO_3 is widely used as the additive in ether-based electrolytes (typically 5 wt % in dimethoxyethane and 1,3-dioxolane, *Angew. Chem. Int. Ed.*, 2018, 57, 14055-14059), while it has very low solubility in carbonate electrolytes ($< 10^{-5} \text{ g mL}^{-1}$, corresponding to $< 0.1 \text{ wt } \%$, *Nano Energy*, 2022, 104, 16, 107881; *Proc. Natl. Acad. Sci. USA*, 2018, 115, 5676-5680). The clamp-like structure of Li^+ -BMC complex-2 and Li^+ -BMC complex-4, with the participation of ether oxygen, should contribute to the dissolution of LiNO_3 by BMC to some extent. However, the reality of limited LiNO_3 -solubility (only about 0.75 wt %) is not consistent with this speculation, which indicates that the both of these two conformations are not the most dominant coordination configuration between Li^+ and BMC.

Furthermore, according to the MD results of 1.2m BMC electrolyte shown in **Fig. 2c**, the coordination number of C-O (including single bond oxygens in the carbonate group and ether oxygens) with Li^+ is less than half of that for C=O (carbonyl oxygens). This implies that carbonyl oxygens serve as the primary coordinating site while the coordination configurations depending on C-O (Li^+ -BMC complex-2, Li^+ -BMC complex-3 and Li^+ -BMC complex-4) are in a minority. These conclusions can also be confirmed by the Raman results shown in **Fig.^(answer) 3**, where negligible shifts are observed for the carbonate C-O and ether C-O-C stretching vibrations with the introduction of Li salt. By contrast, a new peak locating at 1729 cm^{-1} arises near the carbonate C=O stretching vibration (1748 cm^{-1}) owing to the binding with Li^+ . This can effectively demonstrate that carbonyl oxygen is the main solvating site of BMC (Please refer to the detailed response to Question 7 you kindly raised below).

According to your warm advice, **Fig.^(answer) 1 and 2** have been integrated and offered as Supplementary Fig. 2 in the revised manuscript and supplementary information to elucidate the possible conformations of Li^+ -BMC complex and BMC structure. **Fig.^(answer) 3** has been provided as Supplementary Fig. 14 in the revised manuscript and supplementary information to support the solvation structure.

Fig.(answer) 3 (a) Raman spectra of 1.2m DME compared with pure DME in 800-925 cm⁻¹ (ether C-O-C stretching vibration). (b) Fitted Raman spectra of 1.2m DMC compared with pure DMC in 1680-1800 cm⁻¹ (carbonate C=O stretching vibration). Raman spectra of 1.2m BMC compared with pure BMC in (c) 800-925 cm⁻¹ (ether C-O-C and carbonate C-O stretching vibrations) and (d) 1680-1800 cm⁻¹ (carbonate C=O stretching vibration). (e) Raman spectra of LiFSI salt.

Updates in the Revised Manuscript:

(1) Line 170 in the Results of the revised manuscript

The description “Benefitting from the synergetic electron distribution, BMC exhibits a minima electronegativity near the carbonyl oxygen but relatively positive compared to DMC (see ESP maps

in Fig. 1), thus generating a weaker coordination affinity with Li^+ . Interestingly, there exists an alternative coordination configuration for BMC with Li^+ , displaying a similar five-membered ring chelation structure to Li^+ -DME and legitimately yielding a more negative binding energy. It is the feature that enables BMC to dissolve LiNO_3 (Supplementary Fig. 2). However, only a limited amount can be dissolved, suggesting that this specific solvation structure may not dominate and the carbonyl oxygen still remains as the primary coordinating site for Li^+ with relatively weak solvation energy.” has been revised as “Benefitting from the synergetic electron distribution, BMC exhibits a minima electronegativity near the carbonyl oxygen but relatively positive compared to DMC (see ESP maps in Fig. 1), thus generating a weaker coordination affinity with Li^+ (Li^+ -BMC complex-1). Interestingly, there exists alternative coordination configurations for BMC with Li^+ (Supplementary Fig. 2). Some of them demonstrate more negative binding energy, such as Li^+ -BMC complex-2 (similar five-membered ring chelation structure to Li^+ -DME) and Li^+ -BMC complex-4 (clamp-like structure coordinating to Li^+ with both of carbonyl and ether oxygens). These chelation configurations potentially facilitate the dissolution of LiNO_3 in BMC (Supplementary Fig. 3). However, it seems that the dominant configuration does not lie in these chelation solvation structures but rather in the Li^+ -BMC complex-1, which solely relies on carbonyl oxygen coordinating to Li^+ with relatively weak solvation energy. This aspect is discussed in Supplementary Fig. 2 and will be further validated through molecular dynamics (MD) simulations and Raman spectroscopy.”

(2) Note for Supplementary Fig. 2 in the revised supplementary information

The description “Note: There are several alternative major coordination configurations for Li^+ -BMC complex when considering the diverse potential coordinating sites (carbonyl oxygen and ether oxygen). Four possible coordination configurations and corresponding molecular structures of BMC are summarized in Supplementary Fig. 2, including the coordination sites of carbonyl oxygens with Li^+ (Li^+ -BMC complex-1 with a binding energy of $-18.99 \text{ kJ mol}^{-1}$, Supplementary Fig. 2a), the five-membered ring chelation with Li^+ through two oxygen atoms (Li^+ -BMC complex-2 with a binding energy of $-23.82 \text{ kJ mol}^{-1}$, Supplementary Fig. 2b), the scenario where Li^+ is coordinated with two single bond oxygens in the carbonate group (Li^+ -BMC complex-3 with a binding energy of $-7.48 \text{ kJ mol}^{-1}$, Supplementary Fig. 2c), as well as the mutual coordination of carbonyl and ether oxygens with Li^+ (Li^+ -BMC complex-4 with a binding energy of $-34.48 \text{ kJ mol}^{-1}$, Supplementary Fig. 2d). It can be detected that Li^+ -BMC complex-3 shows a relatively positive binding energy, indicating its less possibility to exist in the electrolyte. By contrast, both Li^+ -BMC complex-2 and Li^+ -BMC complex-4 show a chelation structure with stronger binding energy compared to Li^+ -BMC complex-1. However, it should be noted that achieving the coordination configuration of Li^+ -BMC complex-4 requires BMC to adopt a distorted conformational arrangement (BMC-structure-2, Supplementary Fig. 2f). Such structure exhibits a relatively positive electronic energy value, being 6.85 kJ mol^{-1} higher than the common linear BMC-structure-1 (Supplementary Fig. 2e). This suggests that BMC-structure-2 is unfavorable in the whole electrolyte. Consequently, it can be reasonably inferred that the mutual coordination of carbonyl and ether oxygens with Li^+ (Li^+ -BMC complex-

4), which is based on BMC-structure-2, may represent a minority coordination configuration between Li^+ and BMC. Moreover, given that only slight increase of LiNO_3 dissolution in BMC (Supplementary Fig. 3) facilitated by the clamp-like structure involving ether oxygen, it can be inferred that the chelation configurations (Li^+ -BMC complex-2 and Li^+ -BMC complex-4) do not dominate the 1.2m BMC electrolyte. Subsequent MD and Raman results will further confirm this conclusion and demonstrate that the most prevalent coordination configuration is represented by Li^+ -BMC complex-1, where carbonyl oxygen solely coordinates to Li^+ with relatively weak solvation energy.” has been added below Supplementary Fig. 2 in the revised supplementary information.

(3) Note for Supplementary Fig. 3 in the revised supplementary information

The description “Note: LiNO_3 can hardly be dissolved in DMC-based electrolyte. For BMC electrolyte, although it can dissolve LiNO_3 , the dissolution limit is approximately 0.75%.” has been enriched as “Note: LiNO_3 can hardly be dissolved in DMC-based electrolyte, which is consistent with its widely-reported low solubility in carbonate electrolytes ($< 10^{-5} \text{ g mL}^{-1}$, corresponding to $< 0.1 \text{ wt } \%$)^{1,2}. In comparison, BMC-based electrolyte with LiNO_3 is a transparent solution, indicating that it can dissolve more LiNO_3 than in DMC. However, the upper limit of the LiNO_3 -solubility is approximately 0.75 wt %, which is still much lower than that of the commonly-used ether solvents (typically 5 wt % in DME and 1,3-dioxolane)³”

Question 4: The measuring limit of the instrument is lower than the reported flash point of BMC (121C) in the literature, which would give an inaccurate value in this study.

Response: Great thanks for your insightful suggestion. The instrument used for flash points investigation is ABA4 automatic closed-cup flash point tester (Anton Paar), which does have a relatively narrow testing temperature range of -30 to 110 °C. Its up-limit temperature is lower than the reported flash point of BMC (121 °C), thereby yielding an inaccurate value in our previous manuscript. Guiding with your kind remind, we have applied alternative flash point analyzers (MINIFLASH FPH VISION, 10 ~ 400 °C and MINIFLASH FP VISION, -45 ~ 120 °C, Grabner Instruments). The reevaluation of the flash points complied with the professional ASTM D6450 (Continuously closed cup flash point method) standard developed by the American Society Testing and Materials (*J. Loss Prevent. Proc.*, 2015, 33, 183-187). All measurements were conducted using a sample volume of 1 mL and repeated for three times to ensure the reliability. The initial temperature was typically set at 18 °C below the anticipated flash point, and the heating rate was maintained at $5.5 \pm 0.5 \text{ }^\circ\text{C per minute}$. The apparatus automatically regulated the oven temperature to the initial set point, and subsequently lifts the sample cup into the oven for heating at a specified rate. An electric arc with high voltage was applied as the ignition source, operating at a frequency of 1 °C per minute. Following each flash test, a short air pulse from a small membrane compressor introduced approximately $1.5 \pm 0.5 \text{ mL}$ of air into the test chamber. Flash detection in the chamber

relied on detecting an increase in pressure after ignition occurs. A pressure transducer with an operational range of 80 to 177 kPa, resolution of 0.1 kPa, and minimum accuracy of ± 0.5 kPa was utilized for this purpose. The limit for successful flash point detection was set at a pressure increase of 20 kPa, which corresponded to approximately a flame volume of 1.5 mL above barometric pressure.

Fig.(answer) 4 Details of flash point results for different liquid samples exported from the software coupled with the instrument. The pressure increase of 20 kPa, which corresponds to a flame volume of approximately 1.5 mL above the barometric pressure, is established as the threshold for successful flash point detection.

The re-measured flash point results are presented in **Fig.(answer) 4** with original pictures exported from the instrument. It can be observed that the flash points of 1.2m DME, 1.2m DMC, and LB010 show minimal deviation from previous values (**Table(answer) 1**). In contrast, BMC exhibits a significantly higher flash point of 117 °C, close to the literature-reported value (121 °C , *Energy*

Environ. Sci., 2023, 16, 2924), where the slight discrepancy may be attributed to differences in the instrument and the testing standard. Additionally, the flash point of 1.2m BMC has also been determined as 120 °C. The results are also consistent with your kind remind that flash point instruments with a limited temperature range exhibit significant inaccuracy when non-standard methods are used to measure the flash points of samples near or beyond the measuring limit of the instruments, which should be paid attention for similar researches. The newly obtained flash points shown in **Fig.(answer) 4** has been re-figured and compared with various linear ether and carbonate solvents/electrolytes, as shown in **Fig.(answer) 5**. It can be detected that the flash points of pure BMC solvent (117 °C), together with the 1.2m BMC electrolyte (120 °C), are superior to numerous other linear ethers and carbonates.

According to your warm remind, **Fig.(answer) 5** has been provided as **Fig. 2d** in the revised manuscript to update the comparisons of boiling/flash points among various liner ether and carbonate solvents/electrolytes.

Table(answer) 1 Flash points for different liquid samples tested on the MINIFLASH FPH VISION compared with those obtained in our previous manuscript. *The flash points of 1.2m DME, 1.2m DMC and LB010 are tested on the MINIFLASH FP VISION (with a temperature range from -45 to 120 °C).

Sample	Flash point (previous version)	Flash point (revised version)
1.2m DME	6 °C	2 °C*
1.2m DMC	19 °C	19 °C*
LB010	26 °C	27 °C*
BMC	85 °C	117 °C
1.2m BMC	>110 °C	120 °C

Fig.(answer) 5 Comparisons of boiling/flash points among various liner ether and carbonate solvents/electrolytes.

Updates in the Revised Manuscript:

(1) *Line 191 in the Results of the revised manuscript*

The description “Furthermore, its flash point was observed to reach as high as 85 °C, offering strong confirmation of the superior characteristics of BMC solvent compared to numerous other linear ethers and carbonates (Fig. 2d).” has been revised as “Furthermore, its flash point was observed to reach as high as 117 °C, offering strong confirmation of the superior characteristics of BMC solvent compared to numerous other linear ethers and carbonates (Fig. 2d).”

(2) *Line 201 in the Results of the revised manuscript*

The description “Furthermore, these electrolytes generally display remarkably low flash points (< 26 °C, Fig. 2d), which are highly susceptible to ignition upon exposure to an open flame (Fig. 2f, Supplementary Videos 1-4). As a comparison, 1.2m BMC exhibits a flash point higher than 110 °C (the measuring limit of the instrument), ensuring its complete nonflammability (Supplementary Video 5).” has been revised as “Furthermore, these electrolytes generally display remarkably low flash points (≤ 27 °C, Fig. 2d), which are highly susceptible to ignition upon exposure to an open flame (Fig. 2f, Supplementary Videos 1-4). As a comparison, 1.2m BMC exhibits a high flash point of 120 °C, ensuring its complete nonflammability (Supplementary Video 5).”

(3) *Line 551 in the Methods of the revised manuscript*

The description “The flash points of electrolytes were investigated on the ABA4 automatic closed-cup flash point tester (Anton Paar).” has been revised as “The flash points of electrolytes were investigated on flash point analyzers (MINIFLASH FPH VISION (10 ~ 400 °C) and MINIFLASH FP VISION (-45 ~ 120 °C), Grabner Instruments) according to ASTM D6450 standard. The detailed testing process can be referred in previous literature⁴⁰.”

Question 5: Provide values for the calculated and measured densities for the electrolyte to confirm the reliability of the MD simulation.

Response: Thank you very much for your valuable suggestion. In accordance with your kind recommendation, we have conducted meticulous measurements of the densities for various electrolytes. The measured densities for 1.2m DME, 1.2m DMC, and 1.2m BMC were determined as 0.956 g cm⁻³, 1.137 g cm⁻³, and 1.202 g cm⁻³, respectively, through weighing the mass of 1 mL solution for three times. The calculated densities obtained from MD simulation results are found as follows: 0.964 g cm⁻³ for 1.2m DME, 1.148 g cm⁻³ for 1.2m DMC, and 1.210 g cm⁻³ for 1.2m BMC. It is noteworthy that all calculated values from MD exhibit slight overestimations compared to their corresponding measured counterparts with an acceptable deviation (within a margin of less than 1%), thereby indicating the reliability of our MD simulation. To make it clear, the calculated and measured densities for the electrolyte are supplemented in Methods section of the revised manuscript highlighted with green background.

Updates in the Revised Manuscript:

Line 507 in the Methods of the revised manuscript

The description “The reliability of the MD simulation results was confirmed by comparing the calculated densities of electrolytes from simulation with their experimentally measured counterparts as follows: 1.2m DME_(MD) = 0.964 g cm⁻³ compared to 1.2m DME_(Exp) = 0.956 g cm⁻³, 1.2m DMC_(MD) = 1.148 g cm⁻³ compared to 1.2m DMC_(Exp) = 1.137 g cm⁻³, and 1.2m BMC_(MD) = 1.210 g cm⁻³ compared to 1.2m BMC_(Exp) = 1.202 g cm⁻³, respectively, with discrepancies remaining within a margin of less than 1%.” has been added in Methods section in the revised manuscript highlighted with green background.

Question 6: The deconvoluted peaks in the Raman spectra for the same coordination species do not seem to have the same FWHM and peak center, which cannot give a reasonable population estimate. Although the peaks for DMC and BMC seem nearly identical when directly overlapping each other, the calculated proportion is largely different. It is recommended to provide fitting details.

Response: We sincerely appreciate your precious suggestion and thoughtful remind, which has prompted us to realize the potential misestimation based on our Raman fitting results. In accordance with your insightful recommendation, we have reviewed the fitting details for Raman spectra in the previous version of **Fig. 3d-3f**, and found that some deconvoluted peaks for the same coordination species in different electrolytes dose exhibit divergences in FWHM and peak center. In order to rectify these differences, we have carefully repeated the peak-differentiating and imitating for these Raman spectra. Throughout this process, we ensured that the FWHM and peak center remained fixed for identical coordination species across different electrolytes to ensure the reliability of the results. The redone peak-differentiating and imitating results for the Raman spectra are depicted in **Fig.(answer) 6**, where corresponding values are also summarized in **Table(answer) 2**. It can be detected that the newly acquired proportions for various coordination environments of FSI⁻ show slight deviations from the previous ones, while they still maintain consistent rankings: 1.2m DME demonstrates a higher percentage of free FSI⁻ and a lower percentage of AGG compared to other solvent systems, whereas 1.2m BMC exhibits a lower percentage of free FSI⁻ and a higher percentage of AGG. To be rigorous, the replotted Raman spectra of various electrolytes in 680-780 cm⁻¹ (S-N-S bending vibration of FSI⁻) shown in **Fig.(answer) 6a-6c** have replaced the previous version of **Fig. 3d-3f** in the revised manuscript. Meanwhile, **Table(answer) 2** has also been provided as Supplementary Table 2 in the revised supplementary information. All these changes have been highlighted with green background.

Table(answer) **2** Fitting details for the Raman spectra of 1.2m DME, 1.2m DMC and 1.2m BMC in 680-780 cm^{-1} (S-N-S bending vibration of FSI).

	Free FSI			CIP			AGG		
	Peak center	FWHM	Area (proportion)	Peak center	FWHM	Area (proportion)	Peak center	FWHM	Area (proportion)
1.2m DME	719.0 (cm^{-1})	18.1 (cm^{-1})	35675.0 (65.3%)	730.6 (cm^{-1})	19.2 (cm^{-1})	13161.9 (24.1%)	742.3 (cm^{-1})	27.8 (cm^{-1})	5803.2 (10.6%)
1.2m DMC	719.0 (cm^{-1})	18.1 (cm^{-1})	10437.6 (42.7%)	730.6 (cm^{-1})	19.2 (cm^{-1})	10517.9 (43.0%)	742.3 (cm^{-1})	27.8 (cm^{-1})	3490.6 (14.3%)
1.2m BMC	719.0 (cm^{-1})	18.1 (cm^{-1})	21212.9 (38.2%)	730.6 (cm^{-1})	19.2 (cm^{-1})	22886.8 (41.2%)	742.3 (cm^{-1})	27.8 (cm^{-1})	11472.6 (20.6%)

Fig.(answer) **6** Raman spectra of (a) 1.2m DME, (b) 1.2m DMC and (c) 1.2m BMC in 680-780 cm^{-1} (S-N-S bending vibration of FSI).

Updates in the Revised Manuscript:

Line 245 in the Results of the revised manuscript

The description “It can be observed that the proportion of free FSI in 1.2m DME (71.4%) decreases to 44.7% and 35.2% in 1.2m DMC and 1.2m BMC, respectively. Synchronously, AGG increases from only 7.8% in 1.2m DME to 16.5% and 20.1% in 1.2m DMC and 1.2m BMC, indicating more anions participating in solvation structure.” has been revised as “It can be observed that the proportion of free FSI in 1.2m DME (65.3%) decreases to 42.7% in 1.2m DMC and 38.2% in 1.2m BMC (Supplementary Table 2), respectively. Synchronously, AGG increases from only 10.6% in 1.2m DME to 14.3% (1.2m DMC) and 20.6% (1.2m BMC), indicating more anions participating in solvation structure.”

Question 7: Also, the results in supplementary Fig 10, which identifies the dominant solvation site, should be compared with other solvent systems. The dramatic change of the peak intensities for C-O stretching vibrations in supplementary Fig 10a needs to be explained.

Response: Thanks a lot for your insightful advice. According to your kind suggestion, we have compared the Raman results of 1.2m BMC with other solvent systems (pure DME, 1.2m DME, pure DMC and 1.2m DMC) to identify the dominant solvation site of BMC. Here a consistent vertical scale has been applied for each pure solvent with its corresponding electrolyte in Raman spectra to make a direct comparison (i.e., pure DME vs. 1.2m DME; pure DMC vs. 1.2m DMC; pure BMC vs. 1.2m BMC). As depicted in **Fig.(answer) 3a**, certain stretching vibrations of C-O-C (851 cm^{-1}) undergo a significant blue shift and give rise to a new peak (875 cm^{-1}) corresponding to the solvated DME after the introduction of LiFSI in pure DME. This well indicates the coordination of some DME molecules with Li^+ through their sole solvation site, the ether oxygen. **Fig.(answer) 3b** presents the Raman spectra of pure DMC and 1.2m DMC. The carbonate C=O stretching vibration peak of free (uncoordinated) DMC locates at 1757 cm^{-1} . Upon the addition of LiFSI in DMC, a new peak locating at 1735 cm^{-1} arises, corresponding to the coordination of DMC to Li^+ with its carbonyl oxygen. As for the case with BMC, negligible shifts are observed for the carbonate C-O and ether C-O-C stretching vibrations after the introduction of LiFSI (**Fig.(answer) 3c**), which is quite different from that observed in 1.2m DME. In contrast, a new peak locating at 1729 cm^{-1} arises near the carbonate C=O stretching vibration (1748 cm^{-1} , **Fig.(answer) 3d**), which is similar to 1.2m DMC and should correspond to the solvated BMC. Such results can effectively demonstrate that carbonyl oxygen is the main solvating site of BMC, which echoes the discussion in Question 3 you kindly raised.

Additionally, we appreciate your insightful notice on the dramatic change of the peak intensities for C-O stretching vibrations after introducing LiFSI (Supplementary Fig. 10 in the previous version of the manuscript, namely Supplementary Fig. 14 of the revised manuscript). Prior to the answer, we would like to apologize for the lack of precision in our plotting methodology. In fact, the C=O stretching vibration intensity shown in previous Supplementary Fig. 10b also decreases with the addition of LiFSI in previous Supplementary Fig. 10c, which is ignored owing to the absence of a consistent vertical scale. For a fair comparison, we have incorporated the Raman spectra of the pure solvent and its corresponding electrolyte into a single figure, ensuring they share the same vertical scale bar (**Fig.(answer) 3a-3d**). It can be detected that regardless of the solvent systems, there will always be a certain reduction in peak intensities for solvents after incorporating LiFSI due to Raman's response being contingent on the amount of vibrating groups. With the introduction of LiFSI, there is a decrease in BMC content which consequently leads to a reduction in peak intensities for carbonate C-O and ether C-O-C stretching vibrations. Moreover, the antisymmetric stretching vibration of S-N-S from LiFSI at around 850 cm^{-1} slightly overlaps with that of ether C-O-C stretching vibrations (**Fig.(answer) 3e**) (*RSC Adv.* 2016, 6, 23327-23334), which might interfere the relative change between carbonate C-O and ether C-O-C stretching vibrations shown in **Fig.(answer) 3c**.

According to your warm advice, Raman spectra shown in **Fig.(answer) 3** has been provided as Supplementary Fig. 14 in the revised manuscript and supplementary information highlighted with

green background.

Fig.(answer) 3 (a) Raman spectra of 1.2m DME compared with pure DME in 800-925 cm⁻¹ (ether C-O-C stretching vibration). (b) Fitted Raman spectra of 1.2m DMC compared with pure DMC in 1680-1800 cm⁻¹ (carbonate C=O stretching vibration). Raman spectra of 1.2m BMC compared with pure BMC in (c) 800-925 cm⁻¹ (ether C-O-C and carbonate C-O stretching vibrations) and (d) 1680-1800 cm⁻¹ (carbonate C=O stretching vibration). (e) Raman spectra of LiFSI salt.

Updates in the Revised Manuscript:

Note for Supplementary Fig. 14 in the revised supplementary information

The description “Note: The carbonate C-O and ether C-O-C stretching vibrations only experience slight blue shift after introducing LiFSI in pure BMC. In contrast, a new peak arises near the carbonate C=O stretching vibration, corresponding to the solvated BMC. The obtained results indicate that carbonyl oxygen is the main solvating site of BMC.” has been revised as “**Note: As depicted in Supplementary Fig. 14a, certain stretching vibrations of C-O-C (851 cm^{-1}) undergo a significant blue shift and give rise to a new peak (875 cm^{-1}) corresponding to the solvated DME after the introduction of LiFSI in pure DME. This well indicates the coordination of some DME molecules with Li^+ through their sole solvation site, the ether oxygen. Supplementary Fig. 14b presents the Raman spectra of pure DMC and 1.2m DMC. The carbonate C=O stretching vibration peak of free (uncoordinated) DMC locates at 1757 cm^{-1} . Upon the addition of LiFSI in DMC, a new peak locating at 1735 cm^{-1} arises, corresponding to the coordination of DMC to Li^+ with its carbonyl oxygen. As for the case with BMC, negligible shifts are observed for the carbonate C-O and ether C-O-C stretching vibrations after the introduction of LiFSI (Supplementary Fig. 14c), which is quite different from that observed in 1.2m DME. In contrast, a new peak locating at 1729 cm^{-1} arises near the carbonate C=O stretching vibration (1748 cm^{-1} , Supplementary Fig. 14d), which is similar to 1.2m DMC and should correspond to the solvated BMC. Such results can effectively demonstrate that carbonyl oxygen is the main solvating site of BMC, and Li^+ -BMC complex-1 is the most prevalent coordination configuration. Moreover, it can be detected that regardless of the solvent systems, there will always be a certain reduction in peak intensities for solvents after incorporating LiFSI. Noteworthy, the relative change between carbonate C-O and ether C-O-C stretching vibrations shown in Supplementary Fig. 14c might be interfered by the antisymmetric stretching vibration of S-N-S from LiFSI at around 850 cm^{-1} , which slightly overlaps with that of ether C-O-C stretching vibrations (Supplementary Fig. 14e)⁵.” below Supplementary Fig. 14 in the revised version of supplementary information highlighted with green background.**

Question 8: The oxidative stability test should be conducted with BMC with LiNO_3 and LiDFBOP additives.

Response: Great thanks for your precious suggestion. In accordance with your recommendation, we conducted the oxidative stability tests of additives-containing 1.2m BMC+ electrolyte on both Al and Pt working electrodes at 1 mV s^{-1} . According to the LSV results exhibited in **Fig.(answer) 7**, the introduction of LiNO_3 and LiDFBOP additives does not show a great impact on the oxidative stability of the BMC-based electrolyte. According to your kind remind, **Fig.(answer) 7** and corresponding discussions have been offered as Supplementary Fig. 26a in the revised manuscript and supplementary information highlighted with green background.

Fig. (answer) 7 Oxidative stability of 1.2m BMC and 1.2m BMC+ electrolytes measured with Al working electrodes at 1 mV s^{-1} (The inset is results obtained with Pt working electrodes).

Updates in the Revised Manuscript:

(1) *Line 298 in the Results of the revised manuscript*

The description “The optimized electrolyte is denoted as 1.2m BMC+ hereafter.” has been enriched as “The optimized electrolyte is denoted as 1.2m BMC+ hereafter, which exhibits minimal variations in terms of oxidative stability, ionic conductivity, and Li^+ transference number when compared to the additives-free 1.2m BMC (Supplementary Fig. 26).”

(2) *Note for Supplementary Fig. 26 in the revised supplementary information*

The description “Note: As demonstrated in Supplementary Fig. 26a, regardless of whether Al or Pt is used as the working electrode, the LSV results imply that the addition of LiNO_3 and LiDFBOP additives has a minor impact on the oxidative stability of the BMC-based electrolyte.” has been added below Supplementary Fig. 26 in the revised supplementary information highlighted with green background.

Question 9: It is hard to estimate the thickness of Li electrodeposits from BMC+ from Supplementary Fig 25C.

Response: Thanks a lot for your kind suggestion. We are sorry that the boundary between the Li electrodeposits and Cu substrate is not clearly visible in 1.2m BMC+, making it challenging to accurately determine the thickness of Li electrodeposits. According to your kind advice, we have carefully prepared the sample again and reconduted SEM characterizations after the first plating on the Cu foil at 0.5 mA cm^{-2} and 3 mAh cm^{-2} using 1.2m BMC+. After conducting multiple measurements at various positions, it can be concluded from the findings depicted in **Fig. (answer) 8**

that the thickness of Li electrodeposits plated in the 1.2m BMC+ electrolyte is approximately 16 μm . To make the thickness clear, Fig.(answer) 8d has been offered as Supplementary Fig. 33c in the revised manuscript and supplementary information highlighted with green background.

Fig.(answer) 8 SEM images for the cross-section of Li electrodeposits in different places after the first plating on the Cu foil at 0.5 mA cm^{-2} and 3 mAh cm^{-2} using 1.2m BMC+.

Question 10: The nail penetration test result using BMC should be compared with that of other electrolytes to see a clear improvement. In fact, 1-Ah pouch cells usually do not explode even with conventional electrolytes.

Response: Thank you for your valuable and expert advice. The nail penetration test result shown in the previous manuscript was obtained with a fully charged 500 mAh Li||NCM811 (3.9 mAh cm^{-2} , N/P = 1.33/1, Electrolyte/Cathode = 2.5 g Ah^{-1}) pouch cell using BMC-based electrolyte. To make a clear comparison, we accordingly utilized the same type of pouch cell with the commercial carbonate (LB010+) electrolyte to assess its performance during the nail penetration test. The obtained results are exhibited in Fig.(answer) 9. After undergoing two formation cycles and being fully charged to 4.3 V, the 500 mAh pouch cell with LB010+ successfully passed the nail penetration test without any explosions. The results well aligned with your point that 1-Ah pouch cells usually do not explode even with conventional electrolytes, possibly because the batteries with low capacity often fail to generate sufficient heat during mechanical abuse to ignite the electrolyte and cause battery explosions. Therefore, in accordance with your recommendation, we conducted the nail penetration tests on pouch cells with a significantly higher capacity of 5 Ah to demonstrate the safety enhancement achieved by BMC. The 5 Ah Li||NCM811 (4 mAh cm^{-2} , N/P = 2.5/1, Electrolyte/Cathode = 2 g Ah^{-1}) pouch cells with LB010+ and 1.2m BMC+ were both charged to a voltage of 3.8 V (equivalent to approximately 40% SOC) for nail penetration tests after two formation cycles, as depicted in Fig.(answer) 10. A steel nail with a diameter of 3 mm was moving at a speed of 25 mm s^{-1} to vertically penetrate through the cell. Unfortunately, the cell with LB010+

Fig.(answer) 9 Charge/discharge curves of 500 mAh Li||NCM811 (3.9 mAh cm^{-2} , $N/P = 1.33/1$, Electrolyte/Cathode = 2.5 g Ah^{-1}) pouch cells operated in (a) LB010+ and (b) 1.2m BMC+ at a rate of 0.1 C, and corresponding photos after nail penetration tests (see insets).

Fig.(answer) 10 Charge/discharge curves of 5 Ah Li||NCM811 (4 mAh cm^{-2} , $N/P = 2.5/1$, Electrolyte/Cathode = 2 g Ah^{-1}) pouch cells operated in (a) LB010+ and (b) 1.2m BMC+ at a rate of 0.1 C, and corresponding photos before and after nail penetration tests (The nail penetration tests are conducted on the pouch cells after they have been charged to the voltage of 3.8 V).

electrolyte failed in the nail penetration and it triggered thermal runaway resulting in violent explosive combustion (**Fig.(answer)10a** and Supplementary Video 10). As a comparison, the cell with 1.2m BMC+ electrolyte successfully passed the nail penetration test (**Fig.(answer)10b** and Supplementary Video 9), which can be inferred with the lower heat release, exceptional thermal stability and non-flammability characteristics of BMC (**Fig. 2f** and **Fig. 5e, f**). This can well support that BMC solvent equips batteries with high safety performance, which is consistent with its outstanding safety for graphite-based LIBs in previous report (*Energy Environ. Sci.*, 2023, 16, 2924). Noteworthy, preventing thermal runaway caused by mechanical abuse will become increasingly challenging with a further rise in the SOC and battery capacity. To totally address the safety concerns of aggressive lithium metal batteries based on nickel-rich cathodes requires more than relying solely on electrolyte engineering. It necessitates comprehensive cooperation across all aspects, including various modifications to electrodes, separators, current collectors, packaging and cell design. According to your expert advice, **Fig.(answer)10** and corresponding discussions have been provided as Supplementary Fig. 49 in the revised manuscript and supplementary information. Meanwhile, the corresponding videos have been offered as Supplementary Videos 9 and 10.

Updates in the Revised Manuscript:

(1) *Line 428 in the Results of the revised manuscript*

The description “Moreover, the Li||NCM811 pouch cell that has been cycled and fully charged in 1.2m BMC+ comes through the nail penetration test without fire explosion (see the inset in Fig. 5f).” has been revised as “Moreover, the Ah-level Li||NCM811 pouch cell after being charged to 3.8 V in 1.2m BMC+ comes through the nail penetration test without fire explosion, which contrasts with the failure observed in LB010+ (Supplementary Fig. 49, Supplementary Videos 9 and 10).”

(2) *Line 558 in the Methods of the revised manuscript*

The description “The nail penetration test for pouch cells was conducted at room temperature in an open-air environment using a BE-9002 nail penetration testing system (Dongguan Bell).” has been revised as “The nail penetration test for pouch cells was conducted at room temperature in an open-air environment using a M01-005 nail penetration testing system (Dongguan Saice). Before nail penetration test, 5 Ah Li||NCM811 (4 mAh cm⁻², N/P = 2.5/1, Electrolyte/Cathode = 2 g Ah⁻¹) pouch cells were charged to 3.8 V after 2 formation cycles. The cell was vertically penetrated by a steel nail with a diameter of 3 mm at a speed of 25 mm s⁻¹ during the test.”

(3) *Note for Supplementary Fig. 49 in the revised supplementary information*

The description “Note: The nail penetration tests were conducted on Ah-level pouch cells to explore the safety performance. 5 Ah Li||NCM811 pouch cells (4 mAh cm⁻², N/P = 2.5/1, Electrolyte/Cathode = 2 g Ah⁻¹) are prepared with LB010+ and 1.2m BMC+ electrolytes, respectively. After two formation cycles, both cells were charged to a voltage of 3.8 V (equivalent to approximately 40% SOC) for nail penetration tests. A steel nail with a diameter of 3 mm was

moving at a speed of 25 mm s^{-1} to vertically penetrate through the cell. As depicted in Supplementary Fig. 49, the cell with LB010+ electrolyte unfortunately failed in the nail penetration and it triggered thermal runaway resulting in violent explosive combustion (Supplementary Fig. 49a and Supplementary Video 10). As a comparison, the cell with 1.2m BMC+ electrolyte successfully passed the nail penetration test (Supplementary Fig. 49b and Supplementary Video 9), which can be inferred with the lower heat release, exceptional thermal stability and non-flammability characteristics of BMC. The results well indicated that BMC solvent exhibited better safety performance.” has been added below Supplementary Fig. 49 in the revised supplementary information highlighted with green background.

Question 11: The comparison of Li plating/stripping CE with carbonate-based and ether-based electrolytes in Fig. 4b needs to be updated. The recent reports (Ali Coskun, *Nat. Commun.*, 2023, 14, 299; Vilas G. Pol, *Nat. Commun.*, 2023, 14, 868; Jun Lu, *Nat. Commun.*, 2023, 14, 1081) have introduced the ether-based electrolytes with a comparable CE.

Response: Great thanks for your precious suggestion. We have carefully read these recent articles focusing on the development of advanced ether-based electrolytes for lithium metal batteries, all of which demonstrate exceptional Li plating/stripping CEs and are compared with our work. According to your kind advice, the comparison of Li plating/stripping CE with carbonate-based and ether-based electrolytes has been updated as **Fig.(answer) 11** with the introduction of these recent reported electrolytes (1.8M LiFSI in DPE (Vilas G. Pol, *Nat. Commun.*, 2023, 14, 868); 2M LiFSI in TFDMP (Ali Coskun, *Nat. Commun.*, 2023, 14, 299); 2M LiFSI in BFE (Jun Lu, *Nat. Commun.*, 2023, 14, 1081)). To make it clear, **Fig.(answer) 11** has been offered as **Fig. 4b** to update the comparison of Li plating/stripping CE with carbonate-based and ether-based electrolytes, which is highlighted with green background in the revised manuscript. Corresponding details have also been added in Supplementary Table 3 of the revised supplementary information highlighted with green background.

Fig.(answer) 11 Comparisons of Li plating/stripping CE with previously reported ether-based and

carbonate-based electrolytes that are in the form of normal concentration electrolytes, HCEs and LHCEs (The newly added contents are framed with the red dash line).

Once again, we would like to express our sincere gratitude for your time and efforts to review our manuscript. Your expert advice and kind remind are of great importance to improve our manuscript.

Response to Reviewer #2

Overall comment: In this manuscript, the authors reported a molecular hybridization strategy aimed at developing a comprehensive solvent suitable for high-energy and high-safety lithium metal batteries. By ingeniously incorporating carbonate groups and ether groups with contrasting properties into a single molecule, the synthesized BMC solvent reasonably modulates the electron distribution and thereby demonstrates considerable oxidation and reduction stability. Moreover, the unique molecular structure also imparts the BMC with many other merits such as relatively weak solvating power, LiNO₃-microsolubility, excellent thermostability, and nonflammability. Paired with 1.2m LiFSI, the optimized single-solvent BMC-based electrolyte exhibits both impressive Li plating/stripping CE and fairish high-voltage tolerance. Consequently, satisfactory cycling performance is fulfilled for Li||high-loading-NCM811 (4.8 mAh cm⁻²) full cells as well as anode-free Cu||NCM811 coin/pouch cells. More importantly, in the case of thermal/mechanical abuses, Li metal pouch cells using BMC also demonstrate reliable safety. The manuscript is well-structured, and the proposed strategy demonstrates brilliance and ease of implementation. Undoubtedly, this work will have a positive impact on innovative research and development in the field of safe and high-energy Li metal batteries, as well as other rechargeable alkali metal systems. Therefore, I highly recommend accepting it for publication in this journal. However, some minor revisions are necessary to further enhance the quality of this manuscript.

Response: We sincerely appreciate your positive acclaim and kind recommendation of our work. Your constructive suggestions would help us further improve the quality of our manuscript. According to your warm instructions, we have made a detailed response as follows and carefully revised the manuscript as well as the supplementary information, which are highlighted with **turquoise** background. We are truly thankful for your valuable comments and we also hope these revisions can well relieve your confusions.

Question 1: In Fig. 1, the authors described the physicochemical properties of a representative ether solvent (DME) and a representative carbonate solvent (DMC), as well as the alterations in their properties after physical blending and molecular hybridization. What perplexes me is why the oxidation and reduction stability of the mixture are both poor after physical blending.

Response: Thanks a lot for your kind comment. It is known that ether solvents generally exhibit excellent reductive stability while most of them face challenges from their poor oxidation stability. This can be attributed to their lone pair electrons on the ether oxygen, which shows a strong donating tendency and readily to be oxidized. As another representative solvent, carbonate solvents are popular for their oxidative stability while they are typically excluded for Li metal anode owing to their poor reductive stability. These features are related to the molecular structure, where the conjugated electron-withdrawing effect of the carbonyl oxygen leads to a pronounced positive

charge on the carbonyl carbon, which shows electron-deficient feature and tends to gain electrons to get reduced. In a word, the linear ether solvent DME can well support the operation of Li metal anode while fail in the high-voltage cathode, just as opposite of linear carbonate solvent DMC. Therefore, physical blending is expected to be an easily conceived way to exert their separate advantages. Unfortunately, simply mixing them together cannot change their molecular structure and thus would not influence the distribution of electron within each individual molecule. This implies that the DME solvent is still the one tends to donate electron and being oxidized during the charging process. Similarly, the electron-deficient feature of DMC solvent is not changed and would be preferentially reduced at low potential. This disappointing recognition indicates physical blending exposes the shortest plank instead of the longest one, which cannot make full use of the advantages of ether and carbonate solvents. Therefore, efforts should be made at the molecular level to re-distribute and regulate the charge distribution, which ignites the conceived BMC solvent via molecularly hybridizing ether oxygen and carbonate groups into a linear molecule. According to your kind comment, we have made revisions in the revised manuscript highlighted with turquoise background to make it clear.

Updates in the Revised Manuscript:

Line 115 in the Results of the revised manuscript

The description: “Unfortunately, simple physical blending often leads to counterproductive outcomes with rather poor redox stability, as the water capacity of a barrel depends on the length of its shortest plank. This can be attributed to the unchanged electron distribution within each individual molecule when mixing with the other.” has been revised as “Unfortunately, simple physical blending often leads to counterproductive outcomes with rather poor redox stability because simply mixing them together cannot change their molecular structure and thus would not influence the distribution of electron within each individual molecule. This disappointing recognition indicates physical blending exposes the shortest plank instead of the longest one, which cannot make full use of the advantages of ether and carbonate solvents.”

Question 2: The authors introduced CH_3OCH_2 - moieties at both ends of DMC to extend the oxyalkyl chain and enhance intermolecular forces, thereby achieving a high boiling point and flash point for BMC (Fig. 2d) to ensure battery safety. However, it is important to note that the increase in intermolecular forces may also impact the solvent's viscosity, thus I recommend conducting viscosity testing.

Response: We greatly appreciate your valuable advice. Exactly as what you point out, intentionally extending molecular chains to enhance intermolecular forces will not only confer advantages such as increased boiling point and flash points for BMC, but also inevitably lead to a noticeable elevation in its viscosity. According to your kind suggestion, the viscosity of BMC solvent has been measured via a rotational rheometer (HAAKE MARS III), which is compared with those of DME and DMC

solvents. As depicted in **Fig.(answer) 12**, the viscosity values are 0.3 mPa s for DME, 0.44 mPa s for DMC, and 2.96 mPa for BMC respectively. Notably, the viscosity of BMC is significantly higher than those of both DME and DMC owing to the extended chain and enhanced intermolecular forces. In order to obtain a more comprehensive understanding of BMC's physicochemical properties, the viscosity of these solvents and their comparison shown in **Fig.(answer) 12** have been provided as Supplementary Fig. 7 in the revised manuscript and supplementary information highlighted with turquoise background.

Fig.(answer) 12 Viscosity tests for different solvents.

Updates in the Revised Manuscript:

(1) *Line 206 in the Results of the revised manuscript*

The description “However, the enhanced intermolecular forces also contribute to a higher viscosity of the electrolyte, consequently leading to a reduction in ionic conductivity (Supplementary Fig. 4).” has been revised as “However, the enhanced intermolecular forces also contribute to a higher viscosity of the solvent (Supplementary Fig. 7), consequently leading to a reduction in ionic conductivity of the BMC-based electrolyte (Supplementary Fig. 8).”.

(2) *Line 540 in the Methods of the revised manuscript*

The description “The viscosity tests were conducted on a rotational rheometer (HAAKE MARS III).” has been added in the Methods of the revised manuscript highlighted with turquoise background.

(3) *Notes for Supplementary Fig. 7 of the revised supplementary information*

The description “Note: The viscosity values are 0.3 mPa s for DME, 0.44 mPa s for DMC, and 2.96 mPa for BMC respectively.” has been added below Supplementary Fig. 7 in the revised supplementary information highlighted with turquoise background.

Question 3: As the authors pointed out: “The upward-shifted E_{Li} can effectively shorten the gap between redox of Li^+/Li and potential window of electrolyte, thereby diminishing the undesired electrolyte decomposition and considerably improving the Li anode CE”. Therefore, they utilized CV tests (Fig. 3g) of 10 mM ferrocene (Fc) to compare the E_{Li} in 1.2m DME, 1.2m DMC and 1.2m BMC electrolytes. To further highlight the difference between molecular hybridization and physical blending, an evaluation of E_{Li} in 1.2m DME/DMC should also be conducted.

Response: We greatly appreciate your valuable suggestion. According to your warm advice, the evaluation of E_{Li} in 1.2m DME/DMC has been conducted with CV test of 10 mM Fc in 1.2m DME/DMC with the Pt electrode. Moreover, the results have been compared with the molecular hybridization and shown in **Fig.(answer) 13**. It can be detected that the redox potential of Fc^+/Fc obtained in 1.2m DME/DMC is around at 3.34 V vs. Li^+/Li , which lies between those obtained in 1.2m DME (3.41 V vs. Li^+/Li) and 1.2m DMC (3.23 V vs. Li^+/Li) electrolytes. A more visually intuitive comparison chart of E_{Li} in different electrolytes is provided after aligning the CV curves based on the redox potential of Fc^+/Fc in **Fig.(answer) 14**. Compared to that obtained in 1.2m DME, 1.2m DME/DMC only shows an upshift of E_{Li} by 0.07 V, which is much lower than that obtained in 1.2m BMC (0.21 V upshift). Considering that the E_{Li} value is influenced by the degree of Li^+ -FSI⁻ ion pairing, it can be inferred that the solvation structure of the electrolyte is just slightly affected by mere physical blending of ether and carbonate. By contrast, the intramolecularly hybridized BMC enables significantly enhanced incorporation of FSI⁻ into the Li^+ primary solvation sheaths (PSSs) for the formation of Li^+ -FSI⁻ ion pairing due to its relatively weak solvating power and steric effect. Therefore, the E_{Li} tested in 1.2m BMC experiences the greatest upshift, effectively shortening the gap between redox of Li^+/Li and potential window of electrolyte, considerably diminishing the undesired electrolyte decomposition and improving the Li anode CE. This also well highlights the advantages of molecular hybridization over merely physical blending. According to your kind advice, **Fig.(answer) 13** and **14** have been provided as **Fig. 3g** and Supplementary Fig. 17, respectively, in the revised manuscript and supplementary information highlighted with turquoise background.

Fig.(answer) 13 Cyclic voltammetry (CV) curves of 10 mM ferrocene (Fc) dissolved in various electrolytes using Pt as working electrodes at 5 mV s^{-1} .

Fig.(answer) 14 CV curves of 10 mM Fc in various electrolytes with Pt as working electrode at 5 mV s^{-1} . The profiles are aligned according to the redox of Fc^+/Fc to make it easier to see the upshifts of E_{Li} in 1.2m DME/DMC, 1.2m DMC and 1.2m BMC. The E_{Li} in 1.2m DME is set as reference.

Updates in the Revised Manuscript:

(1) Line 264 in the Results of the revised manuscript

The description “As for 1.2m DMC, weighed down by the extremely poor reductive stability of DMC itself, side reactions still proceed drastically (Supplementary Figs. 12 and 15) despite there is a fair number of FSI involved in the solvation structure and E_{Li} is upshifted, leading to a terrible Li CE of 9.9% (Fig. 3h). Furthermore, even when physically blended with DME, the 1.2m DME/DMC mixture achieves a CE of only 11.5%, which is much lower than the molecularly hybridized 1.2m BMC (reaching 96%).” has been revised as “As for 1.2m DMC, weighed down by the extremely poor reductive stability of DMC itself, side reactions still proceed drastically (Supplementary Figs. 16 and 19) despite there is a fair number of FSI involved in the solvation structure and the E_{Li} is upshifted by 0.18 V (Supplementary Figs. 17), leading to a terrible Li CE of 9.9% (Fig. 3h) and poor cycling stability with Li||Li symmetric cells (Supplementary Fig. 20). Furthermore, even when

physically blended with DME, the 1.2m DME/DMC mixture only achieves a 0.07 V upshift on the E_{Li} and a CE of 11.5%. Such CE is much lower than that obtained in the molecularly hybridized 1.2m BMC (reaching 96%).”

(2) Note for Supplementary Fig. 17 of the revised supplementary information

The description “Note: After aligning the CV curves based on the redox potential of Fc^+/Fc , distinct differences in E_{Li} among various electrolytes are observed. Specifically, 1.2m DMC and 1.2m BMC show an upshift of E_{Li} by 0.18 V and 0.21 V, respectively, compared to that in 1.2m DME.” has been revised as “Note: After aligning the CV curves based on the redox potential of Fc^+/Fc , a more visually intuitive comparison of E_{Li} in different electrolytes can be observed. Specifically, 1.2m DME/DMC, 1.2m DMC and 1.2m BMC show an upshift of E_{Li} by 0.07 V, 0.18 V and 0.21 V, respectively, compared to that in 1.2m DME. Considering that the E_{Li} value is influenced by the degree of Li^+ -FSI $^-$ ion pairing, it can be inferred that the solvation structure of the electrolyte is lightly affected by mere physical blending of ether and carbonate (1.2m DME/DMC). By contrast, the intramolecularly hybridized BMC enables significantly enhanced incorporation of FSI $^-$ into the Li^+ primary solvation sheaths (PSSs) for the formation of Li^+ -FSI $^-$ ion pairing due to its relatively weak solvating power and steric effect. Obviously, the E_{Li} tested in 1.2m BMC experiences the greatest upshift, effectively shortening the gap between redox of Li^+/Li and potential window of electrolyte, considerably diminishing the undesired electrolyte decomposition and improving the Li anode CE. These results underscore the utmost importance of thoroughly altering physicochemical properties of solvents at the molecular level.” below Supplementary Fig. 17 in the revised supplementary information highlighted with turquoise background.

Question 4: Whether the physicochemical properties of electrolytes (such as ionic conductivity and Li transport number) will be affected by the addition of $LiNO_3$ and $LiDFBOP$?

Response: Thanks a lot for your insightful questions. According to your kind suggestion, the temperature-dependent ionic conductivity and Li^+ transfer number of additives-containing 1.2m BMC+ electrolyte have been measured and shown in **Fig.(answer) 15**. It can be found that the introduction of a small amount of $LiNO_3$ and $LiDFBOP$ has negligible effect on both the ionic conductivity (25°C: 2.48 $mS\ cm^{-1}$ (1.2m BMC+) vs. 2.56 $mS\ cm^{-1}$ (1.2m BMC)) and Li^+ transfer number (0.48 (1.2m BMC+) vs. 0.47 (1.2m BMC)) of electrolyte. These results indicate that the physicochemical properties (such as ionic conductivity and Li transport number) of electrolytes are seldomly affected by the addition of $LiNO_3$ and $LiDFBOP$. According to your warm advice, **Fig.(answer) 15a** and **15b** have been offered as Supplementary Figs. 26b and 26c, respectively, in the revised manuscript and supplementary information highlighted with turquoise background.

Fig.(answer) 15 (a) Ionic conductivity of 1.2m BMC+ at various temperatures compared with that of 1.2m BMC. (b) Chronoamperometry (CA, under a voltage of 10 mV) curves of Li||Li symmetric cells operated in 1.2m BMC+, together with the corresponding EIS profiles (see insets) before and after the CA test and the calculated Li⁺ transfer number.

Updates in the Revised Manuscript:

(1) *Line 298 in the Results of the revised manuscript*

The description “The optimized electrolyte is denoted as 1.2m BMC+ hereafter.” has been revised as “The optimized electrolyte is denoted as 1.2m BMC+ hereafter, and it exhibits minimal variations in terms of oxidative stability, ionic conductivity, and Li⁺ transfer number when compared to the additives-free 1.2m BMC (Supplementary Fig. 26).”

(2) *Note for Supplementary Fig. 26 of the revised supplementary information*

The description “Moreover, as presented in Supplementary Figs. 26b and 26c, it can be found that the effect is also negligible on both the ionic conductivity (25°C: 2.48 mS cm⁻¹ (1.2m BMC+) vs. 2.56 mS cm⁻¹ (1.2m BMC)) and Li⁺ transfer number (0.48 (1.2m BMC+) vs. 0.47 (1.2m BMC)) of electrolyte upon the introduction of additives.” has been added below Supplementary Fig. 26 in the revised supplementary information highlighted with turquoise background.

Question 5: Upon further optimization with both LiNO₃ and LiDFBOP additives, 1.2m BMC+ electrolyte fulfills an impressive Li plating/stripping CE of 99.4% (Fig. 4a). Can the same effect be achieved by adding only one of these additives?

Response: Great thanks for your thoughtful questions. In response, we have conducted the Li plating/stripping CE test using 0.75% LiNO₃-containing 1.2m BMC and 1% LiDFBOP-containing 1.2m BMC electrolytes. The results obtained are presented in **Fig.(answer) 16**. When solely incorporating LiNO₃ or LiDFBOP into the BMC-based electrolyte, the achieved Li plating/stripping CE is 98.3% or 97.2%, respectively. These are higher than that without additives (1.2m BMC, 96%), while are inferior compared to the value attained through the synergistic effect of both LiNO₃ and

LiDFBOP additives (1.2m BMC+, 99.4%). It seems that the presence of LiNO_3 -derived N-containing species (LiN_xO_y and Li_3N) and LiDFBOP-derived P-containing species (P-O and P-F) appears to be crucial for the formation of the optimal SEI, with neither being dispensable. According to your kind question, these results shown in Fig.(answer) 16 and corresponding discussions have been provided as Supplementary Fig. 27 in the revised manuscript and supplementary information highlighted with turquoise background.

Fig.(answer) 16 Li plating/stripping CE test performed at 0.5 mA cm^{-2} and 1 mAh cm^{-2} in BMC-based electrolytes with different additives: (a) LiNO_3 and (b) LiDFBOP.

Updates in the Revised Manuscript:

(1) Line 304 in the Results of the revised manuscript

The description: “The optimized 1.2m BMC+, as shown in Fig. 4a, exhibits a remarkable Li plating/stripping CE of 99.4% over 10 cycles, surpassing the values obtained in 1.2m DME+ (98.9%) and LB010+ (97.5%).” has been revised as “**The optimized 1.2m BMC+, as shown in Fig. 4a, exhibits a remarkable Li plating/stripping CE of 99.4% over 10 cycles, surpassing the values obtained in 1.2m DME+ (98.9%) and LB010+ (97.5%) as well as BMC-based electrolyte with single additive (Supplementary Fig. 27).**”

(2) Note for Supplementary Fig. 27 of the revised supplementary information

The description “**Note: When solely incorporating LiNO_3 or LiDFBOP into the BMC-based electrolyte, the achieved Li plating/stripping CE is 98.3% or 97.2%, respectively, which is lower compared to the value attained through the synergistic effect of both LiNO_3 and LiDFBOP additives. It seems that the presence of LiNO_3 -derived N-containing species (LiN_xO_y and Li_3N) and LiDFBOP-derived P-containing species (P-O and P-F) appears to be crucial for the formation of the optimal SEI, with neither being dispensable. Therefore, 1.2m BMC containing both LiNO_3 and LiDFBOP (1.2m BMC+) is considered as the optimal electrolyte for further research.**” has been added below Supplementary Fig. 27 in the revised supplementary information highlighted with turquoise background.

Question 6: Fig. 4e-4g present in-situ ECCS observations for evolution processes of Li plating/stripping to prove the advancement of optimized BMC-based electrolyte. However, the number of process snapshots is insufficient for comprehensive and intuitive comparisons, as only the initial plating and final stripping stages are captured. Therefore, it would be beneficial to include more detailed process snapshots in Supplementary Information.

Response: Thank you for your thoughtful consideration. According to your warm suggestion, we have captured more snapshots for evolution processes of Li plating/stripping in different electrolytes during in-situ ECCS observations. As shown in **Fig.(answer) 17**, the moments before the test and after each Li-plating and Li-stripping were captured in the different electrolytes. Guided with your warm advice, **Fig.(answer) 17** has also been offered as Supplementary Fig. 34 in the revised manuscript and supplementary information to provide more comprehensive and intuitive comparisons on the behaviors of Li plating/stripping in different electrolytes.

Fig.(answer) 17 In situ ECCS observations with more details and snapshots for evolution processes of Li plating/stripping in (a) 1.2m DME+, (b) LB010+ and (c) 1.2m BMC+.

Once again, we would like to express our sincere gratitude for your time and efforts to review our manuscript. Your warm feedback and thoughtful advice are meaningful for us to improve our manuscript.

Response to Reviewer #3

Overall comment: Conventional ether and carbonate-based electrolytes have significant limitations in high-voltage Li metal batteries. In this work, Xia et al. reported the hybridization of the ether and carbonate functional moieties to form bis(2-methoxyethyl) carbonate (BMC). With the additives, BMC-based electrolyte showed the improved performance. However, the same solvent structure had been published in the journal of EES (10.1039/D3EE00157A), thus significantly reducing the novelty of this work. Therefore, I do not recommend to publish this paper in Nature Communications.

Response: We would like to express our sincere gratitude to the reviewer for your kind evaluation of our manuscript and valuable suggestions, which really play a crucial role in guiding us to enhance the overall quality of our work. To relieve your concerns about the novelty, we would like to make a response in the following two aspects. On the one hand, we agree with you about the novelty of BMC solvent, which has been disclosed and used as nonflammable electrolytes for safe Li-ion batteries in the journal of EES. We herein acknowledge the authors for their systematic investigation about BMC solvent for Li-ion batteries, especially its safety performance and achieved safe Ah-level Ni-rich layered oxide/graphite pouch cells. This guides us to further understand the as-fabricated high-safety Li metal batteries based on the BMC solvent in our manuscript. On the other hand, it is worth noting that electrolytes face different requirements from Li-ion batteries based on graphite anode and Li metal batteries based on Li metal anode, besides the similar demand of safety feature. Specifically, the latter batteries use the hyperactive Li anode, facing issues like parasitic reactions, dendritic Li growth and decreased Columbic efficiency, thereby imposing higher requirements on the electrolyte. Considering the absence of systematic study of BMC solvent in Li metal batteries, we would like to integrate the kind suggestions from Reviewer #1 and you by deleting the inaccurate words “novel solvent” and modifying the descriptions to avoid misleading. In addition, according to your warm advices, we have carefully responded point-by-point in the response letter and made revisions in the revised manuscript as well as supplementary information, which are highlighted with prominent yellow background for easy identification. We sincerely appreciate your constructive advices and hope these detailed responses well addressed your concerns.

Question 1: The structure after hybridization showed better performance compared to DME and DMC, but the electrolyte in the full cell test still needs LiNO₃ and LiDFBOP additive, which implies the BMC solvent still has obvious side reactions.

Response: Thank you for your positive comments about the better performance of the hybridization BMC and your valuable question about its stability. As for your viewpoint that the need of additives implies the BMC solvent still has obvious side reactions, we would like to make a detailed response

from the following two aspects at both cathode side and anode side:

(1) The stability of BMC at the cathode side

On the one hand, considerable oxidative stability of BMC, without any additives, has been comprehensively demonstrated via LSV investigations (**Fig.(answer) 18**). The results obtained on the Pt working electrode (inset of **Fig.(answer) 18**) indicate that the incorporation of a carbonate group at the molecular level enhances the anodic limit of BMC, surpassing that of DME and comparable to that of DMC. It exhibits even better performance against oxidation than DMC-based electrolyte when applying Al, the widely-used current collector for cathode, as the working electrode. On the other hand, it can be detected that neglectable change of the oxidative stability occurs in 1.2m BMC+ electrolyte with additives (turquoise line) compared to the 1.2m BMC without additives (dark cyan line). These comparisons well indicate the good oxidative stability of BMC itself, independent of additives. Further chronoamperometry (CA) tests (Supplementary Fig. 21, shown as **Fig.(answer) 19** here) and subsequent morphological investigations focusing on Al (Supplementary Fig. 22, shown as **Fig.(answer) 20** here) effectively demonstrate the remarkable anti-corrosion properties of BMC-based electrolyte towards Al. Finally, due to its enhanced oxidative stability and remarkable anti-corrosion properties towards Al, 1.2m BMC easily supports NCM811 at 4.3 V with a higher average CE over 200 cycles compared to its DME and DMC counterparts (Supplementary Fig. 23, shown as **Fig.(answer) 21** here). All these findings collectively indicate that BMC solvent shows considerable oxidative stability without significant side reactions between BMC and NCM811 cathode at 4.3 V.

Fig.(answer) 18 Oxidative stability of various electrolytes measured with Al working electrodes at 1 mV s⁻¹ (The inset picture shows the comparison of the results obtained with Pt working electrodes).

Fig.(answer) 19 (a) CA curves and (b) their zoomed-in plots of Al foils performed in 1.2m DMC and 1.2m BMC. The higher residual current indicates the more serious Al corrosion. In 1.2m BMC, during gradually raising the voltage from 4.0 to 5.0 V, the increase on the current is negligible, a clear indication for remarkable anti-corrosion to Al of the BMC-based electrolyte.

Fig.(answer) 20 SEM images of Al foils after the CA tests conducted in (a-c) 1.2m DMC and (d-f) 1.2m BMC. After the CA tests in Fig.(answer) 19, significant holes and extensive pitting are shown on the Al foil measured in 1.2m DMC, revealing severe Al corruptions. On the contrary, smooth surface with no corruptions is found on the Al foil after tested in 1.2m BMC.

Fig.(answer) 21 Cycling performance of Li||NCM811 half-cells operated in various electrolytes.

(2) The stability of BMC at the anode side

It is known that Li metal is hyperactive and readily engages in chemical reactions with solvents lacking sufficient stability. Therefore, the chemical stability of different solvents towards Li metal was firstly investigated via a direct immersion experiment. As presented in **Fig.(answer) 22**, a polished Li disc was immersed into 1.5 mL solvents to evaluate the chemical stability of BMC towards Li metal. Even after a week, the Li disc remains its lustrous appearance and the liquid remains transparent, demonstrating that BMC has sufficient chemical stability towards Li metal without obvious side reactions.

Fig.(answer) 22 Stability of different solvents towards Li metal (The Li foils are meticulously polished and punched into discs with a diameter of 12 mm, which are subsequently immersed in 1.5 mL of different solvents).

In the next logic step, the electrochemical stability of different electrolytes towards Li metal has been explored. Upon the dissolution of LiFSI with a typical concentration, the single-salt-single-solvent electrolyte system (1.2m BMC) also demonstrates a considerable Li plating/stripping Coulombic efficiency (CE) reaching 96%, surpassing that achieved with carbonate-based 1.2m DMC and even ether-based 1.2m DME (Fig. 3h, shown as Fig.(answer) 23 here). Furthermore, Li||Li symmetric cells are applied to show the stability of different electrolytes without additives. As shown in Fig.(answer) 24, the utilization of 1.2m BMC leads to an extended cycle life of 600 hours for Li||Li symmetric cells without significant increase in polarization during cycling, surpassing both 1.2m DMC and 1.2m DME (Please refer to the detailed discussions in Question 5 you kindly raised). These findings substantiate that the molecular hybridization of ether and carbonate provides BMC-based electrolyte with reasonable compatibility for Li plating/stripping, resulting in much fewer side reactions compared to routine DMC and DME counterparts.

Fig.(answer) 23 Li plating/stripping CE obtained from Li||Cu cells after 10 cycles at 0.5 mA cm^{-2} , 1 mAh cm^{-2} in various electrolytes without additives.

Fig.(answer) 24 Cycling performance of Li||Li symmetric cells operated in various electrolytes without additives.

Above results show that BMC solvent exhibits good compatibility and oxidative/reductive stability with both NCM811 cathode and Li metal anode even without additives. The as-achieved progress is undoubtedly encouraging, although its performance still falls somewhat short for Li metal anode. However, it is indeed unrealistic to expect a single salt and solvent electrolyte system

to meet all the requirements of practical lithium metal batteries. In fact, there are few reports of single-solvent electrolyte systems showing excellent lithium metal reversibility at typical salt concentrations, and almost none if the safety of the battery is also considered. Consequently, the majority of current electrolytes for lithium metal batteries are composed of multiple solvents and often require additives to achieve satisfactory performance. To make a compensation, LiNO₃ and LiDFBOP additives were added in the 1.2m BMC electrolyte to further upgrade the electrochemical performance of lithium metal batteries to meet the practical application as much as possible. In the presence of additives, BMC-based electrolyte still maintains its superiority to its ether and carbonate counterparts (inset in **Fig. 4a**, shown as **Fig.(answer) 25** here). According to your kind remind, the oxidative stability of electrolytes with and without additives in **Fig.(answer) 18** has been offered as Supplementary Fig. 26a in the revised manuscript and supplementary information highlighted to make a clear comparison.

Fig.(answer) 25 10 cycles Li plating/stripping CE test performed at 0.5 mA cm⁻² and 1 mAh cm⁻² in various electrolytes containing additives.

Updates in the Revised Manuscript:

(1) *Line 298 in the Results of the revised manuscript*

The description “The optimized electrolyte is denoted as 1.2m BMC+ hereafter.” has been enriched as “The optimized electrolyte is denoted as 1.2m BMC+ hereafter, which exhibits minimal variations in terms of oxidative stability, ionic conductivity, and Li⁺ transfer number when compared to the additives-free 1.2m BMC (Supplementary Fig. 26).”

(2) *Note for Supplementary Fig. 26 in the revised supplementary information*

The description “Note: As demonstrated in Supplementary Fig. 26a, regardless of whether Al or Pt is used as the working electrode, the LSV results imply that the addition of LiNO₃ and LiDFBOP additives has a minor impact on the oxidative stability of the BMC-based electrolyte.” has been added below Supplementary Fig. 26 in the revised supplementary information.

Question 2: The authors claimed the electron distribution change after hybridization, the steric

effect of BMC should also be considered.

Response: Great thanks for your thoughtful remind. Exactly as what you said, hybridization of the ether and carbonate would not only change the electron distribution but also increase the steric hindrance. To estimate such steric effect, we measure the molecular length according to the optimized DFT results, which are presented in **Fig.(answer) 26**. It can be detected that introducing CH_3OCH_2 - moieties at both ends of DMC to hybridize ether and carbonate within one molecule significantly extended the length of BMC to 11.2 Å, much longer than those of DME (6.5 Å) and DMC (6.2 Å). The large molecular size of BMC introduces significant steric hindrance, presenting challenges in effectively accommodating multiple BMC molecules within the Li^+ primary solvation sheaths. Consequently, this increased steric effect could control and weaken the solvation capability of BMC and facilitates the ingress of more FSI into the inner solvation shell (*J. Am. Chem. Soc.* 2021, 143, 18703-18713; *ACS Appl. Mater. Interfaces* 2022, 14, 44470-44478; *Adv. Mater.* 2023, 35, 2303347). According to your kind remind, **Fig.(answer) 26** and corresponding discussion about the steric effect have been provided as Supplementary Fig. 4 in the revised manuscript and supplementary information highlighted with yellow background.

Fig.(answer) 26 Molecular dimensions of different solvents obtained from DFT calculations.

Updates in the Revised Manuscript:

(1) *Line 182 in the Results of the revised manuscript*

The description: “Furthermore, the large molecular size of BMC (Supplementary Fig. 4) introduces significant steric hindrance, presenting challenges in effectively accommodating multiple BMC molecules within the Li^+ primary solvation sheaths (PSSs)²⁷. Combined with the relatively weak solvation energy of dominant Li^+ -BMC complex-1, more anions will have the opportunity to enter the Li^+ PSSs and construct desired anion-derived SEI.” has been added in the revised manuscript highlighted with yellow background.

(2) *Note for Supplementary Fig. 4 in the revised supplementary information*

The description “Note: It can be detected that introducing CH₃OCH₂- moieties at both ends of DMC to hybridize ether and carbonate within one molecule significantly extended the length of BMC to 11.2 Å, much longer than those of DME (6.5 Å) and DMC (6.2 Å). The large molecular size of BMC introduces significant steric hindrance, presenting challenges in effectively accommodating multiple BMC molecules within the Li⁺ primary solvation sheaths. Consequently, this increased steric effect could control and weaken the solvation capability of BMC and facilitates the ingress of more FSI into the inner solvation shell.” has been added below Supplementary Fig. 4 in the revised supplementary information highlighted with yellow background.

Question 3: Compared to DME and DMC, BMC shows extremely low ionic conductivity which leads to the higher polarization and lower specific capacity.

Response: Great thanks for your insightful question and we agree with you about the low ionic conductivity in the electrolyte based on single BMC solvent. When pursuing a stable and safe solvent molecule for LMBs, we intentionally increased the chain length of the molecule to endow the designed solvent with higher thermal stability and safety. However, the resultant BMC also unavoidably exhibits a higher viscosity (2.96 mPa s) than those of DME (0.30 mPa s) and DMC (0.44 mPa s), which can be observed in **Fig.(answer) 12**. As a result, the BMC-based electrolyte exhibits a low ionic conductivity of approximately 2.5 mS cm⁻¹, which is inferior to that obtained in conventional ether and carbonate counterparts, leading to higher polarization and reduced specific capacity in the Li||NCM811 full cell. It is worth noting that, in order to effectively explore the advantages of BMC in terms of redox stability, reversibility of Li anode, regulation of solvation structure, thermal stability and safety, our BMC-based electrolyte adopts a single solvent system to avoid potential interference from co-solvents. However, such single solvent system rarely exists in practical electrolyte formulations. Instead, practical electrolyte systems utilized in lithium-ion batteries or other advanced batteries typically comprise binary, ternary or even quaternary solvent components in order to fulfill the diverse and often contradicting requirements of battery applications as much as possible, which can hardly be met by any individual solvent (*Xu K. Electrolytes, Interfaces and Interphases: Fundamentals and Applications in Batteries. Royal Society of Chemistry, 2023*). As a result, when it comes to practical applications, it is feasible to incorporate co-solvents or diluents into the BMC-based electrolyte to make up for its shortcomings (low ionic conductivity), thereby achieving considerable rate performance of batteries while preserving the function of BMC in terms of cycling and safety performance. Just as reported in the literature (*Energy Environ. Sci., 2023, 16, 2924*), when using BMC as a sole solvent, the ionic conductivity of the 1M LiPF₆-BMC electrolyte is found to be only 1.45 mS cm⁻¹. However, upon the addition of EC as a co-solvent, the resulting mixture of 1M LiPF₆-BMC/EC (7/3 by vol.) exhibits an enhanced ionic conductivity of 3.02 mS cm⁻¹. Consequently, the Li||NMC811 half-cell demonstrates a capacity retention of around 70% at 4 C in 1M LiPF₆-BMC/EC, which is comparable to that (around 76% of its capacity at a rate of 4 C) observed in conventional 1M LiPF₆-EC/DEC (1/1 by vol.). Therefore,

the insufficient ionic conductivity can be compensated via optimized electrolyte formulations with proper cosolvents to make full use of BMC solvent for practical applications.

Fig.(answer) 12 Viscosity tests for different solvents.

Question 4: Why does BMC show non-flammability?

Response: We appreciate your valuable question. The nonflammability indicates the quality of not being flammable or easily burned. The flash point is widely utilized as an index to assess the fire and explosion risks of liquids, including organic electrolytes. As defined by the American Society for Testing and Materials (ASTM), the flash point refers to the minimum temperature at which a sample's vapor ignites upon application of an ignition source, under specified test conditions with pressure correction at 101.3 kPa (*J. Chem. Eng. Data*, 2010, 55, 2943-2950). According to the experimental results, BMC solvent has a high flash point of 117 °C while DME and DMC solvents only display a much lower flash point of 1 °C and 17 °C (see Fig.(answer) 5), respectively. This implies that the BMC solvent is difficult to ignite compared to DME and DMC solvents when released into the atmosphere by evaporation.

Fig.(answer) 5 Comparisons of boiling/flash points among various liner ether and carbonate solvents/electrolytes.

BMC-BMC complex-1
Binding energy: $-13.99 \text{ kJ mol}^{-1}$

BMC-BMC complex-2
Binding energy: $-11.96 \text{ kJ mol}^{-1}$

DMC-DMC complex-1
Binding energy: $-6.00 \text{ kJ mol}^{-1}$

DMC-DMC complex-2
Binding energy: $-7.57 \text{ kJ mol}^{-1}$

DME-DME complex-1
Binding energy: $-6.40 \text{ kJ mol}^{-1}$

DME-DME complex-2
Binding energy: $-7.79 \text{ kJ mol}^{-1}$

Fig. (answer) 27 Optimized structures and binding energy of BMC-BMC, DMC-DMC and DME-DME complexes that are depicted by the ball-and-stick model. The gray, white, and red balls represent carbon (C), hydrogen (H), and oxygen (O) atoms, respectively. All complexes were directly extracted from the corresponding molecular dynamics (MD) simulation results and optimized using the Gaussian 09 software package at the 3LYP/6-311++G(d) level.

Besides the flash point, the degree of flammability is determined by their volatility, i.e. the amount of time it takes for them to evaporate, which is highly related with the intermolecular force between same solvent molecules. To detect whether BMC has advantages of increasing intermolecular force, we carried out some DFT calculations focusing on the binding energy between two same solvent molecules, where the results are provided in **Fig. (answer) 27**. The complexes shown were all directly extracted from the MD simulation results and optimized using the Gaussian 09 software package at the 3LYP/6-311++G(d) level. According to our results, the binding energy between two BMC

molecules ($-13.99 \text{ kJ mol}^{-1}$; $-11.96 \text{ kJ mol}^{-1}$) is almost twice that of two DMC molecules ($-6.00 \text{ kJ mol}^{-1}$; $-7.57 \text{ kJ mol}^{-1}$) or two DME molecules ($-6.40 \text{ kJ mol}^{-1}$; $-7.79 \text{ kJ mol}^{-1}$). This well indicates that the intermolecular force of BMC solvent is indeed much higher than those of DME and DMC solvents, which can be attributed to the intentionally increased mass and chain length of the BMC solvent molecule during the step of molecular design. The stronger intermolecular force of BMC requires a higher vaporization enthalpy, thereby decreasing the saturated vapor pressure as well as the degree of its flammability. As a result, the difficult evaporation of BMC solvent (strong intermolecular force) together with the difficult ignition of BMC (high flash point) successfully equip the BMC with a non-flammable feature compared to DMC and DME.

According to your kind remind, **Fig.(answer) 27** and corresponding discussion have been offered as Supplementary Fig. 6 in the revised manuscript and supplementary information highlighted with yellow background to highlight the advantages of BMC in view of safety.

Updates in the Revised Manuscript:

(1) *Line 204 in the Results of the revised manuscript*

The description “The safety assessments sufficiently verify the functionality resulting from the increased van der Waals force via intentional elongation of the alkoxy chain.” has been revised as “These safety features can be attributed to high flash point and enhanced van der Waals force of BMC owing to the intentional elongation of the alkoxy chain (Supplementary Fig. 6).”

(2) *Notes for Supplementary Fig. 6 of the revised supplementary information*

The description “Note: According to the DFT calculation results shown in Supplementary Fig. 6, the binding energy between two BMC molecules ($-13.99 \text{ kJ mol}^{-1}$; $-11.96 \text{ kJ mol}^{-1}$) is almost twice than that of two DMC molecules ($-6.00 \text{ kJ mol}^{-1}$; $-7.57 \text{ kJ mol}^{-1}$) or two DME molecules ($-6.40 \text{ kJ mol}^{-1}$; $-7.79 \text{ kJ mol}^{-1}$), indicating that the van der Waals force between BMC solvent is significantly enhanced by extending the molecular chain. The stronger intermolecular force of BMC requires a higher vaporization enthalpy, thereby increasing the boiling point ($238 \text{ }^\circ\text{C}$) of BMC. Moreover, the strong intermolecular force of BMC decreases the saturated vapor pressure, increasing the flash point to $117 \text{ }^\circ\text{C}$. Such high flash point equips the electrolyte based on BMC solvent with advantages in view of safety compared to DMC and DME.” has been added below Supplementary Fig. 6 in the revised supplementary information highlighted with yellow background.

Question 5: Symmetrical cell with each electrolyte should be measured to compare the polarization during long cycling.

Response: We appreciate your valuable advice. According to your warm suggestion, we have assembled Li|Li symmetric cells to investigate their evolutions of polarization during long cycling in different electrolytes with/without additives at 0.5 mA cm^{-2} and 1 mAh cm^{-2} . As illustrated in **Fig.(answer) 24**, the symmetric cell with the 1.2m DMC experienced a short circuit after 220 hours

due to the dendrite growth. For the cell based on 1.2m DME, a significant increase in polarization was observed at 280 hours, indicating the accumulation of resistive 'dead Li' and electrolyte depletion. However, despite initially exhibiting higher voltage polarization due to low ionic conductivity, the cell cycled in 1.2m BMC demonstrated superior stability by not significantly increasing its polarization even after 600 hours. As for the Li||Li symmetric cells with different electrolytes in the presence of additives (**Fig. (answer) 28**), there are no significant differences observed in their electrochemical performance over 900 hours, except that the cell operated in LB010+ exhibits a gradual increase in voltage polarization and surpasses that of the cell using 1.2m BMC+. Given the difficulty in distinguishing the performance disparities among electrolytes containing additives concerning Li plating/stripping reversibility and cycling stability in the Li||Li symmetric systems, a more pronounced comparison can be found to show the advantages of BMC-based electrolytes from the results obtained using Li||Cu cells (Fig. 4a and Supplementary Fig. 28, shown as **Fig. (answer) 29** here). According to your warm advice, symmetrical cells shown in **Fig. (answer) 24** and **28** and corresponding discussions have been offered as Supplementary Figs. 20 and 29 in the revised manuscript and supplementary information highlighted with yellow background

Fig. (answer) 24 Cycling performance of Li||Li symmetric cells operated in various electrolytes without additives.

Fig. (answer) 28 Cycling performance of Li||Li symmetric cells operated in various electrolytes with additives.

Fig. (answer) 29 Long-term Li plating/stripping CE test performed at 0.5 mA cm^{-2} and 1 mAh cm^{-2} in various electrolytes with additives.

Updates in the Revised Manuscript:

(1) *Line 264 in the Results of the revised manuscript*

The description “As for 1.2m DMC, weighed down by the extremely poor reductive stability of DMC itself, side reactions still proceed drastically (Supplementary Figs. 12 and 15) despite there is a fair number of FSI involved in the solvation structure and E_{Li} is upshifted, leading to a terrible Li CE of 9.9% (Fig. 3h).” has been enriched as “As for 1.2m DMC, weighed down by the extremely poor reductive stability of DMC itself, side reactions still proceed drastically (Supplementary Figs. 16 and 19) despite there is a fair number of FSI involved in the solvation structure and the E_{Li} is upshifted by 0.18 V (Supplementary Figs. 17), leading to a terrible Li CE of 9.9% (Fig. 3h) and poor cycling stability with Li||Li symmetric cells (Supplementary Fig. 20).”

(2) *Line 326 in the Results of the revised manuscript*

The description “More importantly, BMC-based electrolytes can exhibit excellent stability and reliability as the calendar aging time (Supplementary Fig. 22) is extended and even the operating temperature (Supplementary Fig. 23) is increased.” has been enriched as “More importantly, BMC-based electrolytes can exhibit excellent stability with Li||Li symmetric cells (Supplementary Fig. 29) and reliability as the calendar aging time (Supplementary Fig. 30) is extended and even the operating temperature (Supplementary Fig. 31) is increased.”

(3) *Notes for Supplementary Fig. 20 of the revised supplementary information*

The description “Note: During cycling at 0.5 mA cm^{-2} and 1 mAh cm^{-2} , the Li||Li symmetric cell operated in the 1.2m DMC experiences a short circuit after 220 h due to the dendrite growth. For the cell operated in 1.2m DME, a significant increase in polarization is observed at 280 h, indicating the accumulation of resistive ‘dead Li’ and electrolyte depletion. However, despite initially exhibiting higher voltage polarization due to low ionic conductivity, the cell cycled in 1.2m BMC demonstrates superior stability by not significantly increasing its polarization even after 600 h.” has been added below Supplementary Fig. 20 in the revised version of supplementary information highlighted with yellow background.

(4) *Notes for Supplementary Fig. 29 of the revised supplementary information*

The description “Note: During cycling at 0.5 mA cm⁻² and 1 mAh cm⁻², the Li||Li symmetric cells operated in 1.2m BMC+ and 1.2m DME+ exhibit exceptional cycling stability over a period of 900 h, with no observed increase in voltage polarization. However, the cell using LB010+ shows a gradual and slight increase in voltage polarization and surpasses that of the cell using 1.2m BMC+, although it does not fail within 900 h.” has been added below Supplementary Fig. 29 in the revised version of supplementary information highlighted with yellow background.

Question 6: Considering the low ionic conductivity of BMC electrolyte, the rate performance of full cells with these electrolytes need be evaluated.

Response: Great thanks for your kind advice. In accordance with your recommendation, we have conducted the rate performance of the Li||NCM811 full cell operated in 1.2m BMC+, compared with those operated in 1.2m DME+ and LB010+ counterparts (Fig.(answer) 30). While increasing the charge/discharge rate from 0.2 C to 0.5 C, 1 C and 2 C, the full cell with 1.2m DME+ delivers discharge capacities of 187 mAh g⁻¹, 173 mAh g⁻¹ and 150 mAh g⁻¹ respectively (capacity retention: 95%, 87% and 76%). Similarly, the full cell with LB010+ demonstrates discharge capacities of 183

Fig.(answer) 30 (a) Rate capability of 100 μm Li||NCM811 full cells operated in various lean electrolytes (7.3 mL Ah⁻¹) in the voltage range of 3-4.3 V, and corresponding selected charge/discharge curves in (b) 1.2m DME+, (c) LB010+ and (d) 1.2m BMC+.

mAh g⁻¹, 166 mAh g⁻¹ and 124 mAh g⁻¹ (capacity retention: 95%, 86% and 64%), while the full cell with 1.2m BMC+ displays discharge capacities of 152 mAh g⁻¹, 121 mAh g⁻¹ and 30 mAh g⁻¹ (capacity retention: 86%, 69% and 17%). It can be obviously observed that the low ionic conductivity of the BMC-based electrolyte, resulting from the high viscosity of BMC, impedes the rate performance of the Li||NCM811 full cell. Therefore, in practical applications, it is imperative to incorporate low-viscosity co-solvents or diluents into the BMC-based electrolyte to enhance its conductivity and achieve considerable rate performance of batteries while preserving the function of BMC in terms of cycling and safety performance. According to your kind suggestion, the rate performance shown in **Fig.(answer) 30** and corresponding discussions have been provided as Supplementary Fig. 44 in the revised manuscript and supplementary information highlighted with yellow background.

Updates in the Revised Manuscript:

(1) *Line 390 in the Results of the revised manuscript*

The description “Regretfully, due to the limited ionic conductivity of BMC-based electrolyte, the full cell exhibits an inferior rate capability compared to those operated in 1.2m DME+ and LB010+ (Supplementary Fig. 44), which can be compensated by designing electrolyte formulations with co-solvents or diluents^{15,22},” has been added in Results of the revised manuscript highlighted with yellow background.

(2) *Notes for Supplementary Fig. 44 of the revised supplementary information*

The description “Note: Supplementary Fig. 44 presents the rate performance of the Li||NCM811 full cell operated in 1.2m BMC+, compared with those operated in 1.2m DME+ and LB010+ counterparts. While increasing the charge/discharge rate from 0.2 C to 0.5 C, 1 C and 2 C, the full cell with 1.2m DME+ delivers discharge capacities of 187 mAh g⁻¹, 173 mAh g⁻¹ and 150 mAh g⁻¹ respectively (capacity retention: 95%, 87% and 76%). Similarly, the full cell with LB010+ demonstrates discharge capacities of 183 mAh g⁻¹, 166 mAh g⁻¹ and 124 mAh g⁻¹ (capacity retention: 95%, 86% and 64%), while the full cell with 1.2m BMC+ displays discharge capacities of 152 mAh g⁻¹, 121 mAh g⁻¹ and 30 mAh g⁻¹ (capacity retention: 86%, 69% and 17%). It can be obviously observed that the low ionic conductivity of the BMC-based electrolyte, resulting from the high viscosity of BMC, impedes the rate performance of the Li||NCM811 full cell. In practical applications, it is feasible to incorporate low-viscosity co-solvents or diluents into the BMC-based electrolyte to enhance its conductivity and achieve improvements in the rate performance of batteries while preserving the function of BMC in terms of cycling and safety performance.” has been added below Supplementary Fig. 44 in the revised supplementary information highlighted with yellow background.

Question 7: I suggest authors add N/P ratio of all full cells in figure.

Response: We appreciate your kind advice. According to your warm suggestion, we have added N/P ratios to all figures that involve full cells. **Fig.(answer) 31-33** have been offered as **Fig. 5** and Supplementary Figs. 42 and 43 in the revised manuscript and supplementary information highlighted with yellow background.

Fig.(answer) 31 (a) Cycling performance of 100 μm Li||NCM811 full cells operated at 0.2 C charge / 0.3 C discharge with various lean electrolytes (7.3 mL Ah⁻¹) in the voltage range of 3-4.3 V. (b) Corresponding selected charge/discharge curves in 1.2m BMC+. (c) Cycling performance of anode-free Cu||NCM811 coin cells operated at 0.2 C charge / 0.3 C discharge with various lean electrolytes (5 mL Ah⁻¹) in the voltage range of 3-4.3 V. (d) Cycling performance of the industrial anode-free Cu||NCM811 pouch cell operated at 0.2 C charge / 1 C discharge with lean 1.2m BMC+ (2g Ah⁻¹) in the voltage range of 3-4.3 V. (e) DSC curves of fully charged NMC811 with different electrolytes. (f) ARC results of fully charged Li||NCM811 pouch cells with different electrolytes.

Fig.(answer) 32 (a) Cycling performance of 100 μm Li||4.8 mAh cm^{-2} NCM811 full cell operated at 0.2 C charge / 0.3 C discharge with lean 1.2m BMC+ (7.3 mL Ah^{-1}) in the voltage range of 3-4.4 V, and (b) corresponding selected charge/discharge curves.

Fig.(answer) 33 (a) Cycling performance of 100 μm Li||4.8 mAh cm^{-2} NCM811 full cells operated with 0.2 C charge / 0.3 C discharge and lean electrolytes (7.3 mL Ah^{-1}) in the voltage range of 3-4.3 V at 50 °C, and corresponding selected charge/discharge curves in (b) 1.2m DME+, (c) LB010+ and (d) 1.2m BMC+.

Once again, we would like to express our sincere gratitude for your time and efforts to review our manuscript. Your kind feedback and valuable suggestions are significant for improving our manuscript.

Reviewers' comments:

Reviewer #1 (Remarks to the Author):

The author partially resolved my concerns, and I do not recommend publishing the current manuscript unless the following issues can be settled.

1. In Figure S2, the Li⁺-BMC complex-4 shows the strongest binding energy among the optimized structures. It is incorrect that the more positive electronic energy of BMC2 than BMC1 prevents complex4 formation. In fact, the more positive value of BMC2 than BMC1 only indicates a preferred configuration for the pure solvent. When coordinated with Li ions, on the other hand, the twisted BMC molecules stabilize Li⁺ better than other forms, which means complex4 is should be the dominant coordination structure. The authors should correct the interpretation of the preferred solvation structure from their DFT calculation.
2. Appropriate discussions on the preferred Li⁺-BMC complex-4 in the MD simulation results need to be included. Since the Li⁺-O-C contribution in BMC is nearly half of the Li⁺-O-C contribution, it is not reasonable to conclude that Li⁺-BMC complex-4 is negligible.
3. Solubility can be affected by many factors. What is the underlying mechanism of efficient LiNO₃ dissolution in DME compared to DMC? Is the five-membered ring chelation structure critical for this? If so, it is suggested to provide a more detailed explanation of the role of this critical structure in the LiNO₃ dissolution beyond the empirical knowledge. Even so, the marginal solubility of LiNO₃ only approximatively reflects the limited population of Li⁺-BMC complex-2, but not Li⁺-BMC complex-4, which should possess a different thermodynamic driving force in LiNO₃ dissolution.
4. The updated flash point results need to be compared with the existing values by properly acknowledging the previous reports in their manuscript because the high thermal stability of BMC is not the novelty of this work.

Reviewer #2 (Remarks to the Author):

The revised manuscript has been improved significantly that can be accepted in current form.

Reviewer #3 (Remarks to the Author):

In the revised version, while the authors improved the manuscript significantly, they still cannot highlight the innovation of the research of this work.

The pure solvent electrolyte of 1.2m BMC only exhibit CE of 96%, which is far lower than the other single-solvent single-salt electrolytes published previously (Nat. Commun., 2023,14, 299; Nat. Commun, 2023, 14, 868; Nat. Commun., 2023, 14, 1081), which indicates that the BMC solvent is less compatible with Li metal and could not generate stable SEI layer for reversible plating/stripping.

The performance improvement of 1.2m BMC electrolyte mainly came from the additive of LiNO₃, however, the role of LiNO₃ additive in carbonate electrolyte was also widely investigated (10.1002/anie.201807034; 10.1002/adma.202007945; 10.1002/anie.202012005). Thus, one would question the impact of having BMC in the electrolyte.

In addition, the rate performance of full cell with 1.2m BMC electrolyte is rather poor, only 30 mah/g at 2C.

All in all, while this reviewer certainly appreciates the revisions, the article is not recommended for publication.

Response Letter to Reviewers

Thanks a lot for your evaluation of our manuscript and the valuable suggestions you provided. Your comments have been instrumental in guiding us to enhance the overall quality of our work. In response to your advice, we meticulously addressed each point in the response letter and implemented revisions in the revised manuscript and supplementary information. These modifications are highlighted in **green color** for your convenience. We trust that our detailed responses adequately address your concerns and hope that the revised versions meet your expectations. Once again, we are grateful for your efforts and constructive advice, which have significantly contributed to the improvement of our manuscript.

Response to Reviewer #1

Overall comment: The author partially resolved my concerns, and I do not recommend publishing the current manuscript unless the following issues can be settled.

Response: We really appreciate your positive feedback on our revisions. Your constructive suggestions would greatly contribute to the further enhancement of the quality of our manuscript. In accordance with your precious instructions, we have made a detailed response below and carefully revised the manuscript as well as the supplementary information, which are highlighted with green background for easy identification. We are truly thankful for your valuable comments and we also hope these revisions can well resolve your concerns.

Question 1: In Figure S2, the Li⁺-BMC complex-4 shows the strongest binding energy among the optimized structures. It is incorrect that the more positive electronic energy of BMC2 than BMC1 prevents complex4 formation. In fact, the more positive value of BMC2 than BMC1 only indicates a preferred configuration for the pure solvent. When coordinated with Li ions, on the other hand, the twisted BMC molecules stabilize Li⁺ better than other forms, which means complex4 is should be the dominant coordination structure. The authors should correct the interpretation of the preferred solvation structure from their DFT calculation.

Response: We sincerely appreciate your insightful comment and we are sorry for the inaccurate explanation that causes misunderstanding. According to your suggestion, we would like to make a detailed response with following two points:

(1) As you kindly point out, the more positive electronic energy of BMC-structure-2 indicates the preferred configuration of BMC-structure-1 in the pure solvent (**Fig. R1a, b**). When coordinated with Li⁺, the electronic energy of Li⁺-BMC complexes can also be used to show their relative stability among different binding forms (**Fig. R1c-1g**). Consistent with your viewpoint, Li⁺-BMC complex-4 (**Fig. R1f**) based on the twisted BMC-structure-2, which coordinated with Li⁺ through its carbonyl and ether oxygens, demonstrates a more negative electronic energy value than other forms. This indicates that Li⁺-BMC complex-4 is the most thermodynamically stable and likely represents the preferred coordination structure from DFT calculations.

(2) However, it is worth mentioning that the DFT results here have certain limitations as they solely consider scenarios where one Li⁺ interacts with a single solvent molecule. The coordination structure of Li⁺ in real electrolyte system is more complicated, including the interactions of Li⁺ to coordinated molecules, Li⁺ to anions, molecule-to-molecule as well as the molecular size and steric hindrance. Therefore, to further elucidate the dominant coordination configuration of BMC involved in realistic solution, more comprehensive and systematic molecular dynamics (MD) simulations have been conducted. According to our MD simulations, we have counted the number and

Fig. R1 (a, b) Different kinds of optimized BMC structures and their respective electronic energy. Optimized structures, binding energy and electronic energy of (c) Li⁺-BMC complex-1 (Li⁺ coordinates with carbonyl O atom), (d) Li⁺-BMC complex-2 (Li⁺ coordinates with one single bond O atom in the carbonate group and one ether O atom), (e) Li⁺-BMC complex-3 (Li⁺ coordinates with single bond O atoms in the carbonate group), (f) Li⁺-BMC complex-4 (Li⁺ coordinates with one carbonyl O atom and one ether O atom) and (g) Li⁺-BMC complex-5 (Li⁺ coordinates with one ether O atom). Purple balls stand for Li⁺, while gray, white and red balls stand for C, H and O atoms, respectively.

proportion of various Li⁺-BMC coordination configurations (**Table R1**) after a thorough summary of all Li⁺ primary solvation structures (the pink dashes represent the coordination lines, with a maximum length of 2.5 Å) (**Table R2**). For convenience, the coordinated BMC molecule configurations were categorized and specified as follows: BMC₁ (coordinated to Li⁺ via one

carbonyl O atom, corresponding to Li⁺-BMC complex-1 in DFT results), BMC₂ (coordinated to Li⁺ via one single bond O atom in the carbonate group and one ether O atom, corresponding to Li⁺-BMC complex-2), BMC₃ (coordinated to Li⁺ via the single bond O atom in the carbonate group, corresponding to Li⁺-BMC complex-3), BMC₄ (coordinated to Li⁺ via one carbonyl O atom and one ether O atom, corresponding to Li⁺-BMC complex-4), and BMC₅ (coordinated to Li⁺ via one ether O atom, corresponding to Li⁺-BMC complex-5). Accordingly, the Li⁺ primary solvation structures from MD results were systematically classified based on the type and number of coordinated species. For example, the Li⁺-2(BMC₁)-1(BMC₄)-1(FSI⁻) indicates that this kind of Li⁺ primary solvation structure contains two BMC₁, one BMC₄ and one FSI⁻.

Table R1 Statistical results of various Li⁺-BMC coordination configurations. The results are based on the summarized Li⁺ primary solvation structures shown in Table R2. (Li⁺-BMC₁, Li⁺-BMC₂, Li⁺-BMC₃, Li⁺-BMC₄ and Li⁺-BMC₅ can correspond to Li⁺-BMC complex-1, Li⁺-BMC complex-2, Li⁺-BMC complex-3, Li⁺-BMC complex-4 and Li⁺-BMC complex-5 shown in DFT results, respectively.)

Coordination configurations of Li ⁺ -BMC	Counts	Proportions (%)	
Li ⁺ -BMC ₁		62	70.4
Li ⁺ -BMC ₂		9	10.2
Li ⁺ -BMC ₃		5	5.7
Li ⁺ -BMC ₄		7	8.0
Li ⁺ -BMC ₅		5	5.7

As presented in **Table R1**, it can be detected that Li⁺-BMC₁ exhibits a remarkable proportion of as high as 70.4% of all solvated BMC molecules, representing the dominant coordination configuration. As a comparison, Li⁺-BMC₄ only displays a proportion of 8.0%, even less than that of Li⁺-BMC₂ (10.2%). These findings are in line with the Raman (**Supplementary Fig. 14**) and RDF (**Fig. 3c**) results in the manuscript, both of which indicate that the carbonyl oxygen serves as the primary solvation site for BMC in the real electrolyte system.

Although Li⁺-BMC₄ and Li⁺-BMC₂ exhibit more negative energy and thermodynamic stability in DFT when considering one solvent molecule to Li⁺, the partially surrounded chelation structure formed from their two solvating sites would hinder more coordinated molecules to participate in the

Li⁺ primary solvation structure of real electrolyte system. In contrast, the coordinating mode of Li⁺-BMC₁ only requires sacrificing one solvating site without partial surrounding, which results in a smaller steric hindrance effect. Therefore, MD simulations can yield more compelling solvation configuration through considering these complex factors including steric hindrance effect, whose statistical outcomes are also consistent with experimental results. Overall, based on these observations, it is reasonable to conclude that Li⁺-BMC₁ (Li⁺-BMC complex-1 in DFT results) represents the dominant coordination configuration where carbonyl oxygen solely coordinates to Li⁺.

In accordance with your suggestion, the previous inappropriate interpretation has been revised. The corresponding Fig. R1c-1g has been supplemented to replace the previous version of Supplementary Fig. 2 in the revised supplementary information. Statistical results of various Li⁺-BMC coordination configurations from MD simulations shown in Table R1 also have been provided as Supplementary Table 2 in the revised manuscript and supplementary information to better elucidate the dominant coordination configuration in the BMC-based electrolyte. All revisions have been highlighted in green background.

Table R2 Summary results of all Li⁺ primary solvation structures (the pink dashes represent the coordination lines, with a maximum length of 2.5 Å) extracted from MD snapshots of 1.2M BMC electrolyte. All Li⁺ solvation structures are classified based on the type and number of coordinated molecules. Specifically, the coordinated BMC molecules are categorized as BMC₁, BMC₂, BMC₃, BMC₄ and BMC₅ according to their coordination configurations with Li⁺. Purple balls stand for Li⁺, while gray, white, red, blue, yellow and cyan balls stand for C, H, O, N, S and F atoms respectively.

Compositions of Li ⁺ solvation structures	Statistical results
Li ⁺ -4(BMC ₁)	
$\text{Li}^+-3(\text{BMC}_1)-1(\text{BMC}_3)$

$\text{Li}^+-3(\text{BMC}_1)-1(\text{FSI})$

$\text{Li}^+-3(\text{BMC}_1)$

$\text{Li}^+-2(\text{BMC}_1)-1(\text{BMC}_2)$

$\text{Li}^+-2(\text{BMC}_1)-1(\text{BMC}_5)$

$\text{Li}^+-2(\text{BMC}_1)-1(\text{BMC}_4)-1(\text{FSI})$

Li^+ -2(BMC_1)-1(BMC_5)-
1(FSI)

Li^+ -2(BMC_1)-2(FSI)

Li^+ -1(BMC_1)-2(BMC_4)

Li^+ -1(BMC_1)-1(BMC_2)

Li^+ -1(BMC_1)-1(BMC_5)

Li⁺-1(BMC₁)-1(BMC₃)

Li⁺-1(BMC₁)

Li⁺-1(BMC₁)-1(BMC₃)-
1(BMC₄)-1(FSI)

Li⁺-1(BMC₁)-1(BMC₂)-
1(FSI)

Li⁺-1(BMC₁)-1(FSI)

Li⁺-1(BMC₁)-1(BMC₃)-
2(FSI)

$\text{Li}^+-1(\text{BMC}_1)-2(\text{FSI})$

$\text{Li}^+-1(\text{BMC}_1)-3(\text{FSI})$

$\text{Li}^+-1(\text{BMC}_2)-1(\text{BMC}_4)-1(\text{FSI})$

$\text{Li}^+-1(\text{BMC}_3)-1(\text{BMC}_5)-1(\text{FSI})$

$\text{Li}^+-1(\text{BMC}_2)-1(\text{FSI})$

$\text{Li}^+-1(\text{BMC}_4)-3(\text{FSI})$

Updates in the Revised Manuscript:

(1) *Line 3 on Page 11 in the revised manuscript*

The description “Similar situation can also be detected in 1.2m BMC, whose Li⁺ PSSs are found to have the highest number of FSI⁻ as expected (Fig. 3c and Supplementary Figs. 12 and 13). The simulation results further demonstrate that ethers-like five-membered ring chelation structure does exist while the weaker combinations between Li⁺ and carbonyl oxygen account for the majority, realizing the weaker solvation ability of BMC.” has been revised as “Similar situation can also be detected in 1.2m BMC, whose Li⁺ PSSs are found to have the highest number of FSI⁻ as expected (Fig. 3c and Supplementary Figs. 12 and 13). The simulation results also validate the presence of a diverse range of Li⁺-BMC coordination configurations, which have also been shown in DFT calculations. It is evident that the coordination number of Li⁺-C-O_{BMC} is only half that of Li⁺-O=C_{BMC}, implying that carbonyl oxygens serve as the main coordinating sites. Further statistical analysis of Li⁺-BMC coordination configurations in MD simulations confirms that the dominant coordination configuration is represented by Li⁺-BMC complex-1 (Supplementary Table 2), where the carbonyl oxygen coordinates solely with Li⁺ and exhibits relatively weak binding energy.”

(2) *Note for Supplementary Fig. 2 in the revised supplementary information*

The description “Note: There are several alternative major coordination configurations for Li⁺-BMC complex, where carbonyl oxygen solely coordinates to Li⁺ with relatively weak solvation energy.” has been revised as “Note: There are several alternative coordination configurations for Li⁺-BMC complex when considering the diverse potential coordinating sites (carbonyl oxygen and ether oxygen). Five possible coordination configurations and corresponding energy values are summarized in Supplementary Fig. 2: Li⁺-BMC complex-1 (Li⁺-BMC₁) which coordinates to Li⁺ with carbonyl oxygen (binding energy: -18.99 kJ mol⁻¹, Supplementary Fig. 2a), Li⁺-BMC complex-2 (Li⁺-BMC₂) whose five-membered ring chelation structure coordinates to Li⁺ through two oxygen atoms (binding energy: -23.82 kJ mol⁻¹, Supplementary Fig. 2b), Li⁺-BMC complex-3 (Li⁺-BMC₃) which coordinates to Li⁺ with the single bond oxygens in the carbonate group (binding energy: -

7.48 kJ mol⁻¹, Supplementary Fig. 2c), Li⁺-BMC complex-4 (Li⁺-BMC₄) whose chelation structure coordinates to Li⁺ through carbonyl and ether oxygens (binding energy: -34.48 kJ mol⁻¹, Supplementary Fig. 2d), and Li⁺-BMC complex-5 (Li⁺-BMC₅) which coordinates to Li⁺ with one ether oxygen atom (binding energy: -15.81 kJ mol⁻¹, Supplementary Fig. 2e). According to these DFT calculations, Li⁺-BMC complex-3 and Li⁺-BMC complex-5 exhibiting relatively more positive energy value suggests the less thermodynamic stability of their coordination configurations. As a comparison, Li⁺-BMC complex-2 and Li⁺-BMC complex-4 with strong chelation structures demonstrate a more negative energy value than other forms, indicating that they are more stable and likely represent the preferred coordination configurations. However, it is worth mentioning that the DFT results have certain limitations as they solely consider scenarios where one Li⁺ interacts with a single solvent molecule, which is different from the coordination structure of Li⁺ in real electrolyte system involving complicated interactions and steric hindrance. This implied that the partially surrounded chelation structure formed from their two solvating sites in Li⁺-BMC complex-2 and Li⁺-BMC complex-4 would hinder more coordinated molecules to participate in the Li⁺ primary solvation structure of real electrolyte system. Considering these complex factors, MD simulations, as conducted below, can output a more compelling distribution of solvation configuration, where Li⁺-BMC complex-1 represents the dominant coordination configuration where carbonyl oxygen solely coordinates to Li⁺. This conclusion will be further confirmed by experimental results later, including the marginal solubility of LiNO₃ and Raman spectra.”

(3) *Note for Supplementary Table 2 in the revised supplementary information*

The description “Note: After summarizing all Li⁺ primary solvation structures in MD simulations, followed by meticulously counting the number of various Li⁺-BMC coordination configurations, their respective proportions are shown in Supplementary Table 2. The dominant solvation configuration is represented by Li⁺-BMC₁ (coordinated to Li⁺ via one carbonyl O atom), constituting a remarkable 70.4% of all solvated BMC molecules. Subsequently, the chelation configurations, Li⁺-BMC₂ (coordinated to Li⁺ via one single bond O atom in the carbonate group and one ether O atom) and Li⁺-BMC₄ (coordinated to Li⁺ via one carbonyl O atom and one ether O atom) account for 10.2% and 8%, respectively. As for Li⁺-BMC₃ (coordinated to Li⁺ via one single bond O atom in the carbonate group) and Li⁺-BMC₅ (coordinated to Li⁺ via one ether O atom), they contribute equally with a proportion of 5.7%. Li⁺-BMC₁ representing the dominant coordination configuration is in line with the Raman results that indicate the carbonyl oxygen serves as the primary solvation site for BMC, while the relatively low content of Li⁺-BMC₂ and Li⁺-BMC₄ with strong chelation configurations is consistent with the limited solubility of LiNO₃ in BMC and only slight blue shift of ether C-O-C stretching vibrations observed in Raman spectra after the introduction of LiFSI in BMC.” has been added below Supplementary Table 2 in the revised supplementary information highlighted with green background.

Question 2: Appropriate discussions on the preferred Li⁺-BMC complex-4 in the MD simulation results need to be included. Since the Li⁺-O-C contribution in BMC is nearly half of the Li⁺-O-C contribution, it is not reasonable to conclude that Li⁺-BMC complex-4 is negligible.

Response: Great thanks for your precious suggestion. We concur with your perspective that Li⁺-BMC complex-4 is not negligible and necessitate appropriate discussions in the MD simulation results, which can be clarified with the following three points:

(1) As discussed in Response to your **Question 1**, although Li⁺-BMC complex-4 (Li⁺-BMC₄) is the preferred configuration from the DFT calculations, the proportion of Li⁺-BMC₄ is only about 8.0% in the real electrolyte system according to the MD simulation results, lower than that of Li⁺-BMC₁ (70.4%) (**Table R1**). Corresponding discussions about the coordination configuration have been included in above Response to **Question 1** and revised manuscript/supplementary information.

Table R1 Statistical results of various Li⁺-BMC coordination configurations. The results are based on the summarized Li⁺ primary solvation structures shown in Table R2. (Li⁺-BMC₁, Li⁺-BMC₂, Li⁺-BMC₃, Li⁺-BMC₄ and Li⁺-BMC₅ can correspond to Li⁺-BMC complex-1, Li⁺-BMC complex-2, Li⁺-BMC complex-3, Li⁺-BMC complex-4 and Li⁺-BMC complex-5 shown in DFT results, respectively.)

Coordination configurations of Li ⁺ -BMC	Counts	Proportion (%)
Li ⁺ -BMC ₁ 	62	70.4
Li ⁺ -BMC ₂ 	9	10.2
Li ⁺ -BMC ₃ 	5	5.7
Li ⁺ -BMC ₄ 	7	8.0
Li ⁺ -BMC ₅ 	5	5.7

(2) Moreover, it is worth mentioning again that the coordination number of Li⁺-O-C shown in the RDF results of **Fig. 3c** involves the contributions from Li⁺-BMC₂, Li⁺-BMC₃, Li⁺-BMC₄, and Li⁺-BMC₅. The total coordination number of Li⁺-O-C is only half that of Li⁺-O=C, implying that carbonyl oxygens serve as the main coordinating sites, which is well consistent with the conclusion that Li⁺-BMC₁ is the dominant coordination configuration.

(3) Furthermore, we are really sorry for the misleading word 'negligible' used to describe the blue

shift of ether C-O-C stretching vibrations in Raman spectra (**Supplementary Fig. 14c**). In fact, we did not deny the existence of coordination configurations of Li^+ -BMC₂, Li^+ -BMC₃, Li^+ -BMC₄, and Li^+ -BMC₅. Instead, we cherish their existence (especially for Li^+ -BMC₂ and Li^+ -BMC₄) and value their contributions to BMC's unique properties, such as the ability to slightly dissolve LiNO_3 . The newly captured snapshots of Li^+ solvation structures (with pink dashes representing the coordination lines) shown in **Table R2** have been selectively extracted to generate **Fig. R2** to better demonstrate the presence of varied Li^+ -BMC coordination configurations.

Fig. R2 Some representative Li^+ solvation structures extracted from the MD simulation of 1.2m BMC electrolyte. Purple balls stand for Li^+ , while gray, white, red, blue, yellow and cyan balls stand for C, H, O, N, S and F atoms respectively.

In order to avoid misleading the readers, the term 'negligible' used to describe the blue shift of ether C-O-C stretching vibrations in Raman spectra (**Supplementary Fig. 14c**) has been substituted with 'slight' in the revised supplementary information. Statistical **Table R1** obtained from MD simulations and corresponding discussions also have been offered as **Supplementary Table 2** in the

revised manuscript and supplementary information to more detailly clarify which Li⁺-BMC configuration is the dominant coordination configuration in the BMC-based electrolyte. Moreover, Fig. R2 has replaced the previous version of Supplementary Fig. 12b to better demonstrate the presence of varied Li⁺-BMC coordination configurations in the electrolyte. All revisions have been highlighted with green background.

Updates in the Revised Manuscript:

(1) *Line 3 on Page 11 in the revised manuscript*

The description “Similar situation can also be detected in 1.2m BMC, whose Li⁺ PSSs are found to have the highest number of FSI⁻ as expected (Fig. 3c and Supplementary Figs. 12 and 13). The simulation results further demonstrate that ethers-like five-membered ring chelation structure does exist while the weaker combinations between Li⁺ and carbonyl oxygen account for the majority, realizing the weaker solvation ability of BMC.” has been revised as “Similar situation can also be detected in 1.2m BMC, whose Li⁺ PSSs are found to have the highest number of FSI⁻ as expected (Fig. 3c and Supplementary Figs. 12 and 13). The simulation results also validate the presence of a diverse range of Li⁺-BMC coordination configurations, which have also been shown in DFT calculations. It is evident that the coordination number of Li⁺-C-O_BMC is only half that of Li⁺-O=C_BMC, implying that carbonyl oxygens serve as the main coordinating sites. Further statistical analysis of Li⁺-BMC coordination configurations in MD simulations confirms that the dominant coordination configuration is represented by Li⁺-BMC complex-1 (Supplementary Table 2), where the carbonyl oxygen coordinates solely with Li⁺ and exhibits relatively weak binding energy.”

(2) *Note for Supplementary Fig. 14 in the revised supplementary information*

The description “Note:As for the case with BMC, negligible shifts are observed for the carbonate C-O and ether C-O-C stretching vibrations after the introduction of LiFSI (Supplementary Fig. 14c),.....” has been revised as “Note:As for the case with BMC, only slight shifts are observed for the carbonate C-O and ether C-O-C stretching vibrations after the introduction of LiFSI (Supplementary Fig. 14c),.....”

(3) *Note for Supplementary Table 2 in the revised supplementary information*

The description “Note: After summarizing all Li⁺ primary solvation structures in MD simulations, followed by meticulously counting the number of various Li⁺-BMC coordination configurations, their respective proportions are shown in Supplementary Table 2. The dominant solvation configuration is represented by Li⁺-BMC₁ (coordinated to Li⁺ via one carbonyl O atom), constituting a remarkable 70.4% of all solvated BMC molecules. Subsequently, the chelation configurations, Li⁺-BMC₂ (coordinated to Li⁺ via one single bond O atom in the carbonate group and one ether O atom) and Li⁺-BMC₄ (coordinated to Li⁺ via one carbonyl O atom and one ether O atom) account for 10.2% and 8%, respectively. As for Li⁺-BMC₃ (coordinated to Li⁺ via one single bond O atom

in the carbonate group) and Li⁺-BMC₅ (coordinated to Li⁺ via one ether O atom), they contribute equally with a proportion of 5.7%. Li⁺-BMC₁ representing the dominant coordination configuration is in line with the Raman results that indicate the carbonyl oxygen serves as the primary solvation site for BMC, while the relatively low content of Li⁺-BMC₂ and Li⁺-BMC₄ with strong chelation configurations is consistent with the limited solubility of LiNO₃ in BMC and only slight blue shift of ether C-O-C stretching vibrations observed in Raman spectra after the introduction of LiFSI in BMC.’ has been added below Supplementary Table 2 in the revised supplementary information highlighted with green background.

Question 3: Solubility can be affected by many factors. What is the underlying mechanism of efficient LiNO₃ dissolution in DME compared to DMC? Is the five-membered ring chelation structure critical for this? If so, it is suggested to provide a more detailed explanation of the role of this critical structure in the LiNO₃ dissolution beyond the empirical knowledge. Even so, the marginal solubility of LiNO₃ only approximately reflects the limited population of Li⁺-BMC complex-2, but not Li⁺-BMC complex-4, which should possess a different thermodynamic driving force in LiNO₃ dissolution.

Response: Great thanks for your valuable advice and insightful reminder. Exactly as you pointed out, the solubility of salt can be affected by many factors, where some empirical ones include the donor number (DN) value, dielectric constant and polarity of the solvent. According to your suggestion, we firstly summarized the reported DN values (**Table R3**), dielectric constants (**Table R4**) and dipole moments (**Table R5**) of some solvents and the solubility of LiNO₃ in these solvents to show their correlation, aiming to reveal which is the most critical parameter affecting the dissolution of LiNO₃.

It can be detected from the tables that DME does exhibit advantages in all three parameters compared to DMC (DN values: 20 for DME vs. 15.2 for DMC; dielectric constants: 7.2 for DME vs. 3 for DMC; dipole moments: 1.71 for DME vs. 0.93 for DMC), which can be used to explain its ability to dissolve LiNO₃. Unfortunately, the relationship between LiNO₃-solubility and DN values or dielectric constant or dipole moment seems ruleless based on the statistical analysis from more solvent species. Therefore, we eager to follow your insightful reminder to make a deep understanding about the underlying mechanism of the LiNO₃ dissolution in different solvents.

Table R3 Donor number (DN) values for various solvents. The values are from the previous report^[1].

Solvents	DN values (descending order)	LiNO ₃ -soluble? (yes or no)
Dimethyl sulfoxide (DMSO)	29.8	Yes ^[2]
1,2-dimethoxyethane (DME)	20	Yes

Gamma-butyrolactone (GBL)	18	Yes ^[3]
Ethyl methyl carbonate (EMC)	17.2	No
Tetraglyme (G4)	16.6	Yes ^[4]
Diethyl carbonate (DEC)	16	No
Dimethyl carbonate (DMC)	15.2	No
Propylene carbonate (PC)	15.1	No ^[3]
Sulfolane (SL)	14.8	Yes ^[5]
Acetonitrile (AN)	14.1	No
Triglyme (G3)	14	Yes ^[6]

Table R4 Dielectric constants for various solvents. The values are from the previous report^[1].

Solvents	Dielectric constants (descending order)	LiNO ₃ -soluble? (yes or no)
PC	64.6	No
DMSO	46.4	Yes
SL	42	Yes
GBL	39	Yes
AN	36	No
G4	7.7	Yes
G3	7.5	Yes
DME	7.2	Yes
EMC	3	No
DMC	3	No
DEC	2.8	No

Table R5 Dipole moments for various solvents. The values are from the previous report^[1].

Solvents	Dipole moments (descending order)	LiNO ₃ -soluble? (yes or no)
PC	4.94	No
SL	4.68	Yes
GBL	4.27	Yes
DMSO	4.1	Yes
AN	3.44	No
G3	2.16	Yes
DME	1.71	Yes
DEC	1.07	No
DMC	0.93	No
G4	—	Yes
EMC	—	No

Generally speaking, the dissolution of salt in the solvent is primarily determined by the competition between cation-anion interaction in solid salt and ion-solvent interaction in solution^[7]. Moreover, a high DN value and strong polarity tend to result in stronger binding energy between Li⁺ and solvent^[8,9]. Therefore, we wonder whether the ability of a solvent to dissolve LiNO₃ has any relationship with its binding energy to Li⁺. Inspired by this, we have performed DFT calculations to assess the binding energy between various solvents and Li⁺ (**Fig. R3**). The correlation between binding energy and the solubility of LiNO₃ has been listed in **Table R6**. Moreover, considering the boundary location of propylene carbonate (PC) solvent, we have conducted the experiment to show the difficult dissolution of LiNO₃ in PC, which is similar to acetonitrile (AN) solvent (**Fig. R4**). It is rather delighted to observe a correlation, wherein solvents displaying a relatively stronger binding energy with Li⁺ demonstrate the ability to dissolve LiNO₃. This correlation appears to be more reliable than the three parameters mentioned above. This is especially true for the much stronger binding energy of Li⁺-DME (-43.06 kJ mol⁻¹) compared to Li⁺-DMC (-19.24 kJ mol⁻¹), which well supports the efficient LiNO₃ dissolution in DME.

Fig. R3 Optimized structures and binding energy of various complexes. Purple balls stand for Li⁺, while gray, white, red, blue and yellow balls stand for C, H, O, N and S atoms respectively.

Table R6 Binding energy for various solvents with Li⁺.

Solvents	Binding energy with Li ⁺ (kJ mol ⁻¹ , ascending order)	LiNO ₃ -soluble? (yes or no)
G3	-69.42	Yes
G4	-62.09	Yes
DME	-43.06	Yes
DMSO	-39.06	Yes
SL	-25.51	Yes
GBL	-24.37	Yes

PC	-22.43	No
DEC	-20.49	No
AN	-20.04	No
EMC	-19.63	No
DMC	-19.24	No

PC + 0.75 wt.% LiNO₃

AN + 0.75 wt.% LiNO₃

Fig. R4 Dissolution of LiNO₃ in PC and AN solvents. The solutions are stirred with a magnetic stirrer for 48 h.

Based on this analysis, we can therefore have a better understanding about the ability of BMC solvent to dissolve LiNO₃. According to discussions about the coordination configuration from DFT and MD results, the chelation configuration of Li⁺-BMC₂ (-23.82 kJ mol⁻¹) with double coordinating sites demonstrates a comparative binding energy with Li⁺-GBL (-24.37 kJ mol⁻¹, the solubility of LiNO₃ is about 3 wt%)^[3] and another chelation configuration of Li⁺-BMC₄ also with double coordinating sites shows an even more negative binding energy of -34.48 kJ mol⁻¹. These configurations can efficiently help dissolve LiNO₃ in BMC. Unfortunately, the limited proportions of Li⁺-BMC₂ (10.2%) and Li⁺-BMC₄ (8.0%) result in the marginal LiNO₃ solubility of 0.75 wt%, which is much lower than that in DME (5 wt%) with similar chelation configuration and double coordinating sites. As a comparison, the dominant coordination configuration, Li⁺-BMC₁, only displays a binding energy of -18.99 kJ mol⁻¹ (even weaker than that of Li⁺-DMC), which might not be able to contribute much to the dissolution of LiNO₃.

We are really grateful for your knowledgeable comments, which has led us to a better recognition about the underlying mechanism of salt dissolution in solvents. With these analyses, we have also gained a deeper understanding about the solvation structures of BMC-based electrolyte.

To prevent any potential misinterpretation that only the five-membered ring chelation structure of Li⁺-BMC₂ promotes the dissolution of LiNO₃, we have replaced the term 'clamp-like' used to describe the configuration of Li⁺-BMC₄ with 'chelation' in the revised manuscript. Additionally, according to your recommendation, a more detailed explanation on the solubility of LiNO₃ in BMC has been provided below **Supplementary Fig. 3** in the revised supplementary information. All revisions have been highlighted with green background.

References

- [1] Chen, J. et al. Design of Localized High-Concentration Electrolytes via Donor Number. *ACS Energy Lett.* **8**, 1723–1734 (2023).
- [2] Liu, S. et al. An inorganic-rich solid electrolyte interphase for advanced lithium-metal batteries in carbonate electrolytes. *Angew. Chem. Int. Ed.* **60**, 3661–3671 (2021).
- [3] Jie, Y. et al. Enabling high-voltage lithium metal batteries by manipulating solvation structure in ester electrolyte. *Angew. Chem. Int. Ed.* **59**, 3505–3510 (2020).
- [4] Chen, T. et al. Stable High-Temperature Lithium-Metal Batteries Enabled by Strong Multiple Ion–Dipole Interactions. *Angew. Chem. Int. Ed.* **61**, e202207645 (2022).
- [5] Piao, N. et al. Lithium metal batteries enabled by synergetic additives in commercial carbonate electrolytes. *ACS Energy Lett.* **6**, 1839–1848 (2021).
- [6] Xu, X., Yue, X., Chen, Y. & Liang, Z. Li Plating Regulation on Fast-Charging Graphite Anodes by a Triglyme-LiNO₃ Synergistic Electrolyte Additive. *Angew. Chem. Int. Ed.* **62**, e202306963 (2023).
- [7] Yao, N. et al. An atomic insight into the chemical origin and variation of the dielectric constant in liquid electrolytes. *Angew. Chem. Int. Ed.* **60**, 21473–21478 (2021).
- [8] Yang, H. et al. Dissolution–precipitation dynamics in ester electrolyte for high-stability lithium metal batteries. *ACS Energy Lett.* **6**, 1413–1421 (2021).
- [9] Fan, X. et al. All-temperature batteries enabled by fluorinated electrolytes with non-polar solvents. *Nat Energy* **4**, 882–890 (2019).

Updates in the Revised Manuscript:

(1) *Line 9 on Page 8 in the revised manuscript*

The description “Some of them demonstrate more negative binding energy, such as Li⁺-BMC complex-2 (similar five-membered ring chelation structure to Li⁺-DME) and Li⁺-BMC complex-4 (chelation structure coordinating to Li⁺ with both of carbonyl and ether oxygens). These chelation configurations potentially facilitate the dissolution of LiNO₃ in BMC (Supplementary Fig. 3).” has been revised as “**Some of them demonstrate more negative binding energy, such as Li⁺-BMC complex-2 (similar five-membered ring chelation structure to Li⁺-DME) and Li⁺-BMC complex-4 (chelation structure coordinating to Li⁺ with both of carbonyl and ether oxygens). These two chelation configurations with stronger interaction to Li⁺ potentially facilitate the dissolution of LiNO₃ in BMC (Supplementary Fig. 3)**”

(2) Note for Supplementary Fig. 3 in the revised supplementary information

The description “Note: LiNO₃ can hardly be dissolved in DMC-based electrolyte, which is consistent with its widely-reported low solubility in carbonate electrolytes (< 10⁻⁵ g mL⁻¹, corresponding to < 0.1 wt %)^{1,2}. In comparison, BMC-based electrolyte with LiNO₃ is a transparent solution, indicating that it can dissolve more LiNO₃ than in DMC. However, the upper limit of the LiNO₃-solubility is approximately 0.75 wt %, which is still much lower than that of the commonly-used ether solvents (typically 5 wt % in DME and 1,3-dioxolane)³.” has been revised as “**Note: The dissolution of salt in a solvent is primarily determined by the competition between cation-anion interaction in solid salt and ion-solvent interaction in solution¹. Therefore, it is generally considered that a stronger interaction between Li⁺ and solvent would facilitate the dissolution of salt. LiNO₃ is found hardly dissolved in DMC-based electrolyte (binding energy of Li⁺-DMC: -19.24 kJ mol⁻¹), which is consistent with its widely-reported low solubility in carbonate electrolytes (< 10⁻⁵ g mL⁻¹, corresponding to < 0.1 wt %)^{2,3}. The commonly-used DME-based electrolyte (binding energy of Li⁺-DME: -43.06 kJ mol⁻¹) exhibited a much higher solubility of LiNO₃ (typically 5 wt %)⁴. As for BMC, the binding energies of Li⁺-BMC complex-1, Li⁺-BMC complex-3 and Li⁺-BMC complex-5 are even weaker than that of Li⁺-DMC, whereas the binding energies of other two chelation configurations (Li⁺-BMC complex-2: -23.82 kJ mol⁻¹ and Li⁺-BMC complex-4: -34.48 kJ mol⁻¹) fall in between Li⁺-DMC and Li⁺-DME. Consequently, the capability of BMC to dissolve LiNO₃ might be attributed to the Li⁺-BMC complex-2 and Li⁺-BMC complex-4 chelation configurations with double coordinating sites. However, the upper limit of the LiNO₃-solubility in BMC is approximately 0.75 wt %, which is still much lower than that in DME. Such phenomenon implies that Li⁺-BMC complex-2 and Li⁺-BMC complex-4 might not be the dominant coordination configurations in BMC-based electrolyte, which will be further verified in the MD simulation and Raman spectra later.”**

Question 4: The updated flash point results need to be compared with the existing values by properly acknowledging the previous reports in their manuscript because the high thermal stability of BMC is not the novelty of this work.

Response: Great thanks for your kind suggestion. In accordance with your recommendation, corresponding description has been added in the revised manuscript to properly acknowledge the previous report (*Energy Environ. Sci.*, 2023, 16, 2924). All revisions have been highlighted in green background.

Updates in the Revised Manuscript:

(1) Line 16 on Page 9 in the revised manuscript

The description “Furthermore, its flash point was observed to reach as high as 117 °C, offering

strong confirmation of the superior characteristics of BMC solvent compared to numerous other linear ethers and carbonates (Fig. 2d)^{6,21,28}.” has been revised as “Furthermore, its flash point was observed to reach as high as 117 °C, which is close to the tested value (121 °C) on the recently published report focusing on the safety performance of BMC²². The superiority over numerous other chain ethers and carbonate solvents in terms of both boiling and flash points significantly highlights the safety characteristics of BMC (Fig. 2d)^{6,21,28}.”

(2) *Line 21 on Page 20 in the revised manuscript*

The description “The obtained result is similar to the very recent report that specifically focuses on the safety of LIBs²².” has been revised as “The obtained result is similar to the recent report that comprehensively delves into the pivotal role of BMC in enhancing the safety of LIBs²².”

Once again, we would like to express our sincere gratitude for your precious time and great efforts in reviewing our manuscript. Your insightful advice and expert guidance have been instrumental in shaping our revisions and responses in both the manuscript and supplementary information. These enhancements are crucial for further improving the overall quality of our work.

Response to Reviewer #2

Overall comment: The revised manuscript has been improved significantly that can be accepted in current form.

Response: We are delighted to receive your recognition of our work and sincerely appreciate your valuable suggestions, which had significantly enhanced the overall quality of our work.

Response to Reviewer #3

Overall comment: In the revised version, while the authors improved the manuscript significantly, they still cannot highlight the innovation of the research of this work.

Response: We sincerely appreciate the reviewer's recognition of our efforts to improve the manuscript. In order to show the innovation of the research, we would like to make a summary and list the key-points as follows: **(1) This work is the first report about the linear BMC with intramolecularly hybridizing ethers and carbonates as a single solvent for lithium metal batteries (LMBs). (2) This work initiates a fundamental understanding on the hybridization of the electron-donating ether group and the electron-withdrawing carbonate group and a comprehensive investigation of BMC for hyperactive Li metal anode and high-voltage LMBs. (3) Even with a typical concentration in the single solvent, the optimized BMC-based electrolyte fulfills a versatile function with a high-voltage tolerance (4.4 V), impressive Li plating/stripping Coulombic efficiency (99.4%), satisfactory cycling and safety performance of practical LMBs.** On the basis of this work, more following-up researches can be expected to design novel solvents for LMBs via intermolecular hybridization of functional groups including ethers and carbonates so that they can learn from each other's merits to offset their own weakness.

Moreover, we would like to thank you for your reminder regarding the performance concerns you raised below. According to your insightful comments, we have prepared a comprehensive point-by-point response to address these questions. Corresponding revisions have also been incorporated into the manuscript and supplementary information, highlighted with a green background for easy identification. We truly appreciate your advices and hope these detailed responses can effectively address your concerns.

Question 1: The pure solvent electrolyte of 1.2m BMC only exhibit CE of 96%, which is far lower than the other single-solvent single-salt electrolytes published previously (Nat. Commun., 2023,14, 299; Nat. Commun, 2023, 14, 868; Nat. Commun., 2023, 14, 1081), which indicates that the BMC solvent is less compatible with Li metal and could not generate stable SEI layer for reversible plating/stripping.

Response: Thanks a lot for your comments. We have carefully collected related information about solvent molecules and detailed electrolyte formulations, and found that all these researches focused on ether-based electrolytes for LMBs and achieved their CEs with a Li salt concentration of approximately 2M (mol L⁻¹). To make a fair comparison, we have also formulated a 2M LiFSI-BMC electrolyte to evaluate its Li plating/stripping CE (**Fig. R5**). It can be detected that increasing the Li salt concentration leads to a significant enhancement in the CE for Li plating/stripping, with an improvement from the initial value of about 96% (for 1.2m (mol kg⁻¹) BMC electrolyte) to a high CE reaching to 98.8% in the single-solvent single-salt formulation. In addition, the CE for BMC-

based electrolytes are superior over ether-based counterpart (73.2% for 1.2m DME), carbonate-based counterpart (9.9% for 1.2m DMC) and their physical blending electrolyte (11.5% for 1.2m DME/DMC), well indicating the compatibility of BMC with Li metal arising from the molecular-scale hybridization of electron-donating ether group and electron-withdrawing carbonate group.

Fig. R5 Li plating/stripping CE tests performed at 0.5 mA cm^{-2} and 1 mAh cm^{-2} in 2M LiFSI-BMC electrolyte.

Noteworthy, increasing the concentration of Li salt can indeed achieve higher CE, but it will also result in increased viscosity and decreased ionic conductivity. Therefore, we herein adopted the additives strategy and enhanced the Li plating/stripping CE from ~96% to 99.4% for 1.2m BMC+, which approaches these reported ether-based electrolytes. To make a clear comparison, the detailed electrolytes have been further provided in **Table R7**. It can be observed that BMC solvent exhibits much higher boiling point ($238 \text{ }^{\circ}\text{C}$) and flash point ($117 \text{ }^{\circ}\text{C}$) compared to these reported ether solvents, showing an advantage in terms of thermal stability and safety. Besides the comparable CE and high boiling/flash point, BMC also exhibits considerable tolerance towards high voltage (4.4 V), obviously demonstrating comprehensive properties required for high-energy and high-safety LMBs.

Table R7 Information for other single-solvent single-salt electrolytes recently published.

Ref.	Solvents	Boiling / flash points ($^{\circ}\text{C}$)	Formulations	Li anode CEs (%)
	TFDMP			
Nat. Commun. , 2023,14, 299		92 / $14.1 \pm 23.2^*$	2M LiFSI-TFDMP	99.6
	DPE			
Nat. Commun. , 2023, 14, 868		90 / -18	1.8M LiFSI-DPE	99.42

Nat. Commun. , 2023, 14, 1081		127 / 14.3 ± 16.3**	2M LiFSI-BFE	99.75
This work		238 / 117	1.2m BMC 2M LiFSI-BMC 1.2m BMC+	~96 98.8 99.4

* The flash point of TFDMP solvent is predicted value that is obtained from SciFinder:

<https://scifinder-n.cas.org/searchDetail/substance/65acc5559a727d6a8558cb40/substanceDetails>;

** The flash point of BFE solvent is predicted value that is obtained from SciFinder:

<https://scifinder-n.cas.org/searchDetail/substance/65acc6399a727d6a8558d266/substanceDetails>.

The CE values obtained with those reported ether solvents mentioned by the reviewer had been added in Fig. 4b and Supplementary Table 4 to compare with that obtained by our BMC-based electrolyte in previous manuscript and supplementary information. For your convenience, the captions of the corresponding figure and table have been highlighted with green background.

Question 2: The performance improvement of 1.2m BMC electrolyte mainly came from the additive of LiNO₃, however, the role of LiNO₃ additive in carbonate electrolyte was also widely investigated (10.1002/anie.201807034; 10.1002/adma.202007945; 10.1002/anie.202012005). Thus, one would question the impact of having BMC in the electrolyte.

Response: Great thanks for your question. To relieve your concern, we would like to make a detailed response with following three points:

(1) In order to exhibit the improvement arising from solvent, we here make a comparison between BMC-based electrolyte and other electrolytes without any additives. The Li plating/stripping CE can reach 96% for the 1.2m BMC electrolyte. As a comparison, the CE for ether-based counterpart (1.2 m DME) is only 73.2%, and even worse for the carbonate-based counterpart (9.9% for 1.2 m DMC) or their physical blending electrolyte (11.5% for 1.2 m DME/DMC). The superior CE well indicates the benefits of BMC solvent, which can be attributed to the molecular-scale hybridization of electron-donating ether group and electron-withdrawing carbonate group.

(2) As discussed in the response to your **Question 1**, increasing the Li salt concentration can realize a higher CE (98.8% for 2M BMC electrolyte). However, it will also lead to increased viscosity and decreased ionic conductivity. Therefore, we would like to adopt the additives strategy by incorporating 0.75 wt.% LiNO₃ and 1 wt.% LiDFBOP into 1.2m BMC (denoted as 1.2 m BMC+), and obtained an enhanced Li plating/stripping CE of 99.4% with the aid of additives. To make a fair comparison, similar formulations have also been applied to ether-based electrolyte and carbonate-based electrolyte. However, both of them still show inferior CE (98.9% for 1.2m DME+ and 97.5% for LB010+) compared to 1.2m BMC+ in spite of their improved CE. This can display the

advantages of BMC-based electrolyte over others even though all electrolytes introduce functional additives.

(3) Thank you for your kind reminder regarding LiNO₃ additives and the inclusion of these valuable references, which helps highlight the advantage of the ability to dissolve LiNO₃ in BMC solvent itself. As you pointed out, the application of LiNO₃ as an additive in carbonate electrolytes has been investigated and proved efficient. However, its poor solubility in many carbonate solvents has long restrained its application in carbonate electrolytes^[10]. Therefore, researches pay efforts to maintain LiNO₃ in carbonate solvents by implanting LiNO₃ particles into porous separators or coating layers to release trace amount of LiNO₃ into the electrolyte, or introduce LiNO₃ solubilizers into carbonate electrolytes to improve the solubility of LiNO₃^[10-14]. The valuable references adopting the latter strategy via the incorporation of an additional LiNO₃-solubilizer have been listed as **Table R8** in details. As a comparison, our designed BMC solvent can form some unique Li⁺-BMC coordination configurations (Li⁺-BMC₂ and Li⁺-BMC₄ in **Fig. R1**, corresponding detailed discussions have been included in above Response to Question 1 and Question 3 raised by **Reviewer #1**, and revised manuscript/supplementary information), which makes BMC capable of dissolving a certain amount of LiNO₃ by its own. This is one of the multiple advantages of BMC that support better performance in this work. Overall, BMC plays a crucial role in achieving considerable electrochemical performance.

Table R8 Detailed information for published LiNO₃-containing carbonate electrolytes mentioned by the reviewer.

Ref.	Formulations	LiNO ₃ -solubilizer
[10]	1M LiPF ₆ in EC/DEC (1:1 by vol.) + 0.2 wt.% CuF ₂ + 1 wt.% LiNO ₃	CuF ₂
[13]	1M LiPF ₆ in FEC/DME (3:7 by vol.) + 0.65M LiNO ₃	DME
[14]	0.8M LiPF ₆ in FEC/DMC (1:4 by vol.) + 5 wt.% (4M LiNO ₃ in DMSO)	DMSO
This work	1.2m LiFSI in BMC + 0.75 wt.% LiNO₃ + 1 wt.% LiDFBOP	/

For your convenience, **Fig. 3h** and **Fig. 4a** together with corresponding descriptions which can demonstrate the superior Li anode CE of BMC over routine ethers and carbonates, as well as emphasize the impact of having BMC in the electrolyte have been highlighted with green background in revised manuscript and supplementary information.

References

[10] Yan, C. et al. Lithium nitrate solvation chemistry in carbonate electrolyte sustains high-voltage

- lithium metal batteries. *Angew. Chem. Int. Ed.* **43**, 14055–14059 (2018).
- [11] Liu, Y. et al. Solubility-mediated sustained release enabling nitrate additive in carbonate electrolytes for stable lithium metal anode. *Nat. Commun.* **9**, 3656 (2018).
- [12] Shi, Q., Zhong, Y., Wu, M., Wang, H. & Wang, H. High-capacity rechargeable batteries based on deeply cyclable lithium metal anodes. *Proc. Natl. Acad. Sci. USA* **115**, 5676–5680 (2018).
- [13] Wang, X. et al. Hybrid electrolyte with dual-anion-aggregated solvation sheath for stabilizing high-voltage lithium-metal batteries. *Adv. Mater.* **33**, 2007945 (2021).
- [14] Liu, S. et al. An inorganic-rich solid electrolyte interphase for advanced lithium-metal batteries in carbonate electrolytes. *Angew. Chem. Int. Ed.* **60**, 3661–3671 (2021).

Question 3: In addition, the rate performance of full cell with 1.2m BMC electrolyte is rather poor, only 30 mah/g at 2C.

Response: Thank you very much for pointing out this concern and we can understand that the rate performance is not that good. It is important to note that the rate performance of the full cell was conducted under very harsh conditions, where the areal capacity of NCM811 cathode is up to 4.8 mAh cm⁻², even surpassing some batteries available in the market. When operated at a rate of 2 C, the applied current density reaches 9.6 mA cm⁻² for the high-mass-loading cathode and the Li anode, which will inevitably induce significant overpotential and cause the loss of capacity. Reducing the areal capacity of NCM811 cathode can efficiently enhance the rate capability of the cell. As depicted in **Fig. R6**, the cell can deliver a high capacity of around 150 mAh g⁻¹ at a rate of 2 C when the areal capacity of NCM811 cathode is decreased to 1.5 mAh cm⁻². Even under a high rate of 3 C, the cell operated in 1.2m BMC+ can still output a high capacity of around 120 mAh g⁻¹.

Fig. R6 (a) Rate capability of 100 μm Li||1.5 mAh cm⁻² NCM811 cell operated in 1.2m BMC+ with the voltage range of 3-4.3 V, and (b) corresponding selected charge/discharge curves.

Besides, given the stringent demands of practical applications requiring high areal capacity, incorporating co-solvents to enhance the ionic conductivity of BMC-based electrolytes will be a more feasible approach. Take adding FEC co-solvent as an example, the 1.2m BMC/FEC (1/1 by

weight, without additives) successfully enables 100 μm Li||4.8 mAh cm^{-2} NCM811 full cell to provide a capacity of more than 100 mAh g^{-1} at 2 C (**Fig. R7**). Similar strategies have also been shown to be effective by the previous literature (*Energy Environ. Sci.*, 2023, 16, 2924). **Table R9** summarizes the performance and improvement of rate capability via above two methods in order to better relieve your concern. Therefore, the poor rate performance obtained in 1.2m BMC electrolyte can be compensated via proper co-solvents, which indicates that BMC is still a promising solvent for high-safety and high-energy LMBs.

Fig. R7 (a) Rate capability of 100 μm Li||4.8 mAh cm^{-2} NCM811 full cell operated in lean 1.2m BMC/FEC (1/1 by weight) (7.3 mL Ah^{-1}) with the voltage range of 3-4.3 V, and (b) corresponding selected charge/discharge curves.

Table R9 Comparison for rate performance of cells operated with different areal capacity of NCM811 cathode and different electrolyte formulations.

Rate	Capacity (mAh g^{-1})		
	4.8 mAh cm^{-2} NCM811 1.2m BMC+	1.5 mAh cm^{-2} NCM811 1.2m BMC+	4.8 mAh cm^{-2} NCM811 1.2m BMC/FEC (1/1 by wt.)
0.5 C	152.7	187.4	168.5
1 C	119.0	173.6	149.9
2 C	30.0	150.4	103.1

In order to better demonstrate that the poor rate performance of 100 μm Li||4.8 mAh cm^{-2} NCM811 full cell with BMC-based electrolyte can be compensated via proper co-solvents, the rate performance of 100 μm Li||4.8 mAh cm^{-2} NCM811 full cell operated in 1.2m BMC/FEC electrolyte (**Fig. R7**) has been offered as Supplementary Fig. 45 in the revised manuscript and supplementary information. The corresponding modifications have been highlighted with green background.

Updates in the Revised Manuscript:

Note for Supplementary Fig. 45 in the revised supplementary information

The description “Upon the introduction of FEC as a co-solvent, the 1.2m BMC/FEC (1/1 by weight, without additives) successfully enables 100 μm Li||4.8 mAh cm^{-2} NCM811 full cell to provide a capacity of more than 100 mAh g^{-1} at 2 C, which is superior to 1.2m BMC+” has been added below Supplementary Fig. 45 in the revised supplementary information highlighted with green background.

Question 4: All in all, while this reviewer certainly appreciates the revisions, the article is not recommended for publication.

Response: We really appreciate your recognition of our revisions and offer a chance to better improve our manuscript. According to your suggestions, we have made more revisions and supplementary data to relieve your concerns. We hope these revisions are satisfactory and acceptable to meet your expectations.

Once again, we would like to express our gratitude to you for your valuable comments and constructive suggestions, which have significantly contributed to improving our manuscript.

REVIEWERS' COMMENTS

Reviewer #1 (Remarks to the Author):

The authors have resolved my concerns, and I recommend publishing the revised manuscript.

Reviewer #3 (Remarks to the Author):

In this revised version, the authors provided an additional set of data including the addition of co-solvents to BMC to improve the rate performance. The reviewer also notes the increased CE upon increasing the salt concentration. Taken all together, the manuscript is now suitable for publication.

Response to Reviewer #1

Overall comment: The authors have resolved my concerns, and I recommend publishing the revised manuscript.

Response: We really appreciate your recognition of our revised manuscript. We would like to express our sincere gratitude for the precious time and tremendous efforts you have dedicated to reviewing our manuscript. Your insightful advice and expert guidance have played a pivotal role in shaping our revisions and responses, both in the main manuscript and supplementary information. These enhancements are of utmost importance in further enhancing the overall quality of our work.

Response to Reviewer #3

Overall comment: In this revised version, the authors provided an additional set of data including the addition of co-solvents to BMC to improve the rate performance. The reviewer also notes the increased CE upon increasing the salt concentration. Taken all together, the manuscript is now suitable for publication.

Response: We are sincerely grateful for your acknowledgement of the revisions we made. We would like to extend our heartfelt appreciation to you for taking the time and effort to provide us with such valuable comments and constructive suggestions which have played a pivotal role in enhancing the quality of our manuscript.